# Essential omega-3 fatty acids tune microglial phagocytosis of synaptic elements in the mouse developing brain

C. Madore[1,2,14], Q. Leyrolle[1,3,14], L. Morel[1,14], M. Rossitto[1,14], A. D. Greenhalgh[1], J. C. Delpech[1], M. Martinat [1], C. Bosch-Bouju[1], J. Bourel[1], B. Rani [4], C. Lacabanne[1], A. Thomazeau [1], K. E. Hopperton[5], S. Beccari [6], A. Sere[1], A. Aubert[1], V. De Smedt-Peyrusse[1], C. Lecours[7], K. Bisht[7], L. Fourgeaud[8], S. Gregoire[9], L. Bretillon [9], N. Acar[9], N. J. Grant[10], J. Badaut [11], P. Gressens[3,12], A. Sierra [6], O. Butovsky [2,13], M. E. Tremblay [7], R. P. Bazinet[5], C. Joffre[1], A. Nadjar [1✉] & S. Layé[1✉]

Omega-3 fatty acids (n-3 PUFAs) are essential for the functional maturation of the brain. Westernization of dietary habits in both developed and developing countries is accompanied by a progressive reduction in dietary intake of n-3 PUFAs. Low maternal intake of n-3 PUFAs has been linked to neurodevelopmental diseases in Humans. However, the n-3 PUFAs deficiency-mediated mechanisms affecting the development of the central nervous system are poorly understood. Active microglial engulfment of synapses regulates brain development. Impaired synaptic pruning is associated with several neurodevelopmental disorders. Here, we identify a molecular mechanism for detrimental effects of low maternal n-3 PUFA intake on hippocampal development in mice. Our results show that maternal dietary n-3 PUFA deficiency increases microglia-mediated phagocytosis of synaptic elements in the rodent developing hippocampus, partly through the activation of 12/15-lipoxygenase (LOX)/ 12-HETE signaling, altering neuronal morphology and affecting cognitive performance of the offspring. These findings provide a mechanistic insight into neurodevelopmental defects caused by maternal n-3 PUFAs dietary deficiency.

---

[1] Univ. Bordeaux, INRAE, Bordeaux INP, NutriNeuro, UMR 1286, F-33000 Bordeaux, France. [2] Ann Romney Center for Neurologic Diseases, Department of Neurology, Brigham and Women´s Hospital, Harvard Medical School, Boston, MA, USA. [3] NeuroDiderot, Inserm, Université de Paris Diderot, F-75019 Paris, France. [4] Department of Health Sciences, University of Florence, Florence, Italy. [5] Department of Nutritional Sciences, University of Toronto, Toronto, ON M5S 3E2, Canada. [6] Achucarro Basque Center for Neuroscience, University of the Basque Country and Ikerbasque Foundation, 48940 Leioa, Spain. [7] Neurosciences Axis, CRCHU de Québec-Université Laval, Québec City, QC, Canada. [8] Molecular Neurobiology Laboratory, The Salk Institute for Biological Studies, La Jolla, CA 92037, USA. [9] Centre des Sciences du Goût et de l'Alimentation, AgroSup Dijon, CNRS, INRAE, Univ. Bourgogne Franche-Comté, F-21000 Dijon, France. [10] CNRS UPR3212, Institut des Neurosciences Cellulaires et Intégratives, Strasbourg, France. [11] CNRS UMR5287, University of Bordeaux, Bordeaux, France. [12] Centre for the Developing Brain, Department of Division of Imaging Sciences and Biomedical Engineering, King's College London, King's Health Partners, St. Thomas' Hospital, London SE1 7EH, UK. [13] Evergrande Center for Immunologic Diseases, Brigham and Women's Hospital, Harvard Medical School, Boston, MA, USA. [14]These authors contributed equally: C. Madore, Q. Leyrolle, L. Morel, M. Rossitto. ✉email: agnes.nadjar@u-bordeaux.fr; sophie.laye@inrae.fr

L ipids are one of the main constituents of the central nervous system (CNS). Among these, arachidonic acid (AA, 20:4n-6) and docosahexaenoic acid (DHA, 22:6n-3) are the principal forms of the long chain (LC) polyunsaturated fatty acids, omega-6 and omega-3 (referred to as n-6 or n-3 PUFAs) of the grey matter. These two LC PUFAs are found predominantly in the form of phospholipids and constitute the building blocks of brain cell membranes[1]. AA and DHA are either biosynthesized from their respective dietary precursors, linoleic acid (LA, 18:2n-6), and α-linolenic acid (ALA, 18:3n-3), or directly sourced from the diet (mainly meat and dairy products for AA, fatty fish for DHA)[2]. In addition, AA and DHA are crucial for cell signaling through a variety of bioactive mediators. These mediators include oxylipins derived from cyclooxygenases (COX), lipoxygenases (LOX), and cytochrome P450 (CYP) pathways[1] that are involved in the regulation of inflammation and phagocytic activity of immune cells[3–6]. An overall increase in the dietary LA/ALA ratio and a reduction in LC n-3 PUFA intake, as found in the Western diet, leads to reduced DHA and increased AA levels in the brain[7].

DHA is essential for the functional maturation of brain structures[8]. Low n-3 PUFA consumption globally has raised concerns about its potential detrimental effects on the neurodevelopment of human infants[9] and the incidence of neurodevelopmental diseases, such as Autism Spectrum Disorder (ASD) and schizophrenia[10].

To understand the mechanisms by which n-3/n-6 PUFA imbalance affects CNS development, we investigated the impact of maternal dietary n-3 PUFA deficiency on offspring's microglia, the resident immune cells involved in CNS development and homeostasis[11]. Microglia are efficient phagocytes of synaptic material and apoptotic cells, which are key processes in the developing brain[12]. Specifically, once neuronal circuits are established, microglia contribute to the refinement of synaptic connections by engulfing synaptic elements in early post-natal life, in a fractalkine- and complement cascade-dependent manner[13–15]. This refinement is important for behavioral adaptation to the environment[16].

We have previously reported that a maternal dietary n-3 PUFA deficiency alters the microglial phenotype and reduces their motility in the developing hippocampus[17]. Therefore, we investigated the effect of maternal (i.e. gestation and lactation) n-3 PUFA deficiency on microglial function and its consequences on hippocampal development. Using multidisciplinary approaches, we reveal that maternal exposure to an n-3 PUFA deficient diet impairs offspring's microglial homeostatic signature and phagocytic activity, underpinning excessive synaptic pruning and subsequent behavioral abnormalities. N-3 PUFA deficiency drives excessive microglial phagocytosis of synaptic elements through the oxylipin 12/15-LOX/12-HETE pathway and is a key mechanism contributing to microglia-mediated synaptic remodeling in the developing brain. These results highlight the impact of bioactive fatty acid mediators on brain development and identify imbalanced nutrition as a potent environmental risk factor for neurodevelopmental disorders.

## Results

**Maternal dietary n-3 PUFA deficiency alters the morphology of hippocampal neurons and hippocampal-mediated spatial working memory.** The hippocampus plays an essential role in learning and memory, with the neurons of the CA1 region required for spatial working memory. Therefore, we assessed the effects of maternal dietary n-3 PUFA deficiency on neuronal morphology in the offspring hippocampus CA1 using Golgi-Cox staining at postnatal day (P)21. Dendritic spine density of Golgi-stained CA1 pyramidal neurons was significantly decreased in n-3 deficient mice compared to n-3 sufficient mice (Fig. 1a, b). This was associated with a decrease in the length of dendrites, whereas the complexity of dendritic arborization was unchanged (Fig. 1a, b). Western blot analyses on whole hippocampi showed a significant decrease in the expression of the post-synaptic scaffold proteins PSD-95 and Cofilin, but not of SAP102, in n-3 deficient vs. n-3 sufficient mice (Fig. 1c, d). AMPA and NMDA subunit expression levels were unaffected (Supplementary Fig. 1A, B). In the Y-maze task, a hippocampus-dependent test assessing spatial working memory, P21 n-3 deficient mice were unable to discriminate between the novel and familiar arms of the maze, unlike the n-3 sufficient mice (Fig. 1e). These data show that maternal n-3 PUFA deficiency disrupts early-life (P21) spine density, neuronal morphology and alters spatial working memory.

**Maternal dietary n-3 PUFA deficiency increases microglia-mediated synaptic loss.** As microglia phagocytose pre- and post-synaptic elements to shape neural networks[13,14], we used electron microscopy (EM) to analyze, at high spatial resolution, microglia–synapse interactions in the hippocampal CA1 region of n-3 deficient vs. n-3 sufficient mice (Fig. 2a, b). At P21, i.e. at the peak of hippocampal synaptic pruning[18], we found significantly more contacts between Iba1-positive microglial processes and the synaptic cleft and more dendritic spine inclusions within those processes in n-3 deficient mice (Fig. 2b). Microglial processes from n-3 deficient mice contained more inclusions within endosomes such as spines and other cellular elements, along with a reduced accumulation of debris in the extracellular space, thus suggesting an increase in phagocytic activity[16] (Fig. 2a, b). This was not due to changes in microglial process size, as diet did not affect process morphology (Supplementary Fig. 2A). In addition, confocal imaging and 3D reconstruction of microglia showed that PSD95-immunoreactive puncta within Iba1-positive microglia were more abundant in the hippocampus of n-3 deficient mice, corroborating the EM data (Fig. 2c, d). To provide further support to this increase in microglial phagocytosis, we stimulated ex vivo hippocampal microglia from n-3 deficient vs. n-3 sufficient P21 mice with pHrodo E.Coli bioparticles. The pHrodo dye-labelled *E. coli* becomes fluorescent upon entering acidic phagosomes, enabling the assessment of phagocytic activity of microglia. The results revealed that the percentage of phagocytic microglia is significantly higher in n-3 deficient mice across the time points studied (Fig. 2e, f and Supplementary Fig. 2B). Increased microglial phagocytosis in the hippocampus was not a compensation to a decrease in microglial density, as there were no differences in microglial density assessed by stereological counting in the different hippocampal sub-regions, including the CA1 (Supplementary Fig. 2C, D).

Enhanced microglial phagocytic activity was also not linked to an increase in cell death, as the volume of the hippocampus, number of neurons and astrocytes, number of apoptotic cells and expression of pro-apoptotic protein Bax and anti-apoptotic protein Bcl-2 were not different between diets (Supplementary Fig. 3A–F). Furthermore, it is unlikely that n-3 PUFA deficiency-induced increase in phagocytosis resulted from brain infiltration of peripherally-derived macrophages. This is consistent with the observation that maternal n-3 PUFA deficiency did not modulate the number of CD11b-high/CD45-high cells within the whole brain (Supplementary Fig. 3G) and that the blood-brain barrier was intact in both diet groups as revealed by the expression of Claudin-5 and GFAP as well as IgG extravasation (Supplementary Fig. 3H–M).

**Maternal dietary n-3 PUFA deficiency dysregulates microglial homeostasis.** During brain development, microglia display a unique transcriptomic signature that supports their physiological

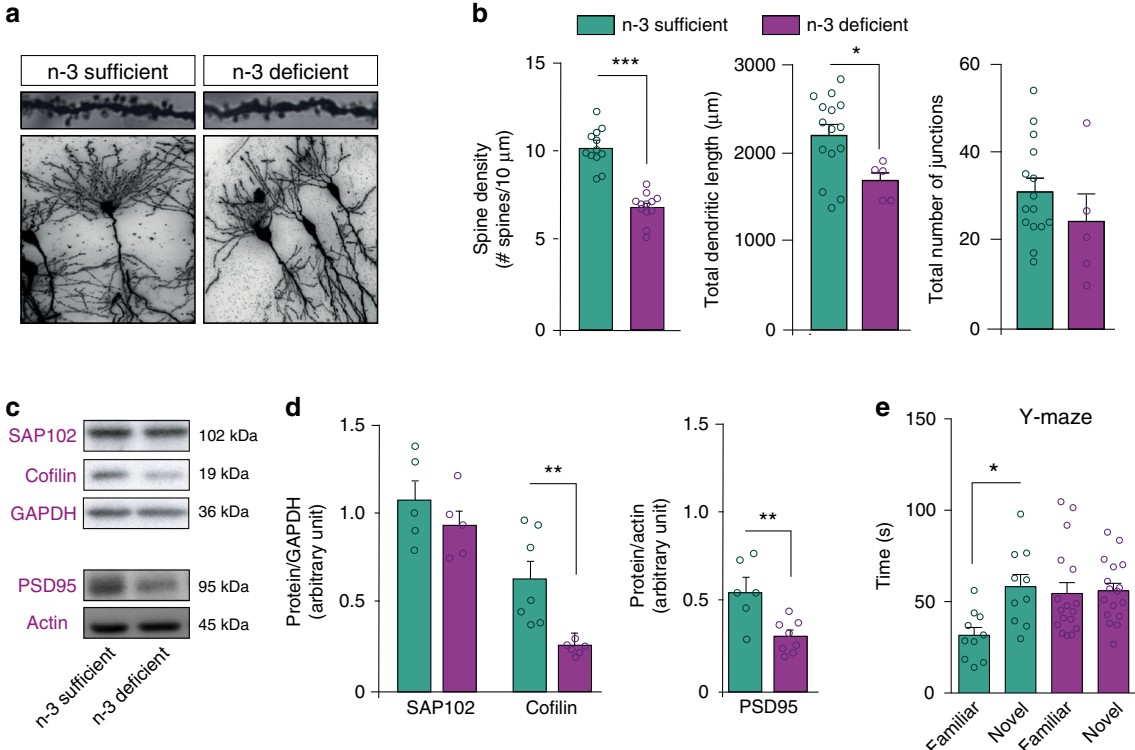

**Fig. 1 Maternal n-3 PUFA deficiency alters neuronal morphology and function. a** Representative images of tertiary apical dendrites (upper panel) and dendritic arborization (lower panel) of hippocampal CA1 pyramidal neurons from n-3 sufficient and n-3 deficient mice at P21 (Golgi staining). **b** Quantification of spine density, total dendritic length and total number of junctions in n-3 deficient and n-3 sufficient animals. For spine counting, $n = 12$ CA1 pyramidal neurons of the hippocampus, 15–45 segments per animal, 3 mice per group. Means ± SEM. Two-tailed unpaired Student's $t$-test, t = 6.396, ***$p < 0.0001$. For dendritic arborization and total dendritic length, $n = 5$ neurons from 3 mice for n-3 deficient mice and n = 15 neurons from 4 mice for n-3 sufficient mice. Two-tailed unpaired Student's $t$-test; Dendritic length, t = 2.501, *$p = 0.0223$; number of junctions: $t = 1.118$, $p = 0.28$. **c** Representative western blots. **d** Expression of scaffolding proteins in n-3 deficient mice relative to n-3 sufficient mice. Means ± SEM; $n = 5$–8 mice per group. Two-tailed unpaired Student's $t$-test, t = 0.9986, $p = 0.35$, SAP102; $t = 3.572$, **$p = 0.0044$, cofilin; $t = 3.529$, **$p = 0.0042$, PSD95. **e** Time spent in novel vs familiar arm in the Y maze task. Means ± SEM; $n = 10$–17 mice per group. Two-way ANOVA followed by Bonferroni post hoc test: diet effect, $F_{(1,50)} = 5.47$, $p = 0.071$; arm effect, $F_{(1,50)} = 10.08$, $p = 0.015$; interaction, $F_{(1,50)} = 8.08$, $p = 0.029$; familiar vs novel arm for n-3 diet mice: *$p < 0.05$. Source data are provided as a Source data file.

functions, including in synaptic refinement[19]. Therefore, we investigated whether and how n-3 PUFA deficiency modulates the homeostatic molecular signature of microglia isolated from n-3 deficient vs. n-3 sufficient mouse brains. Using a Nanostring-based mRNA chip containing 550 microglia-enriched genes[20], 154 of these genes were found to be significantly down-regulated and 41 genes up-regulated in microglia from n-3 deficient mice whole brains (Fig. 2g, j and Table 1). Among the 40 genes reported to be unique to homeostatic microglia by[20], 20 were significantly dysregulated, indicating that maternal n-3 PUFA deficiency alters homeostatic functions of microglia (Fig. 2g). We next sought to define the diet-specific microglial phenotype by assessing patterns of gene co-expression using unbiased network analysis software, Miru[21]. The utility of identifying gene expression patterns and transcriptional networks underpinning common functional pathways in microglia has been previously described[22]. Using a Markov clustering algorithm to non-subjectively subdivide our data into discrete sets of co-expressed genes, we found three major clusters of genes that were differentially regulated in n-3 deficient mice (Fig. 2i, j). The mean expression profiles of these three clusters showed that clusters 1 (microglial homeostatic signature) and 2 (fatty acid uptake, transport and metabolism) contained genes whose expression was relatively lower in the n-3 deficient group as compared to n-3 sufficient group. In contrast, cluster 3 (innate immune response and inflammation) contained genes with relatively greater

expression in n-3 deficient mice (Fig. 2j). These results show that an n-3 PUFA deficient diet dysregulates microglial homeostasis and increases microglial immune and inflammatory pathways during development.

**Fatty acid changes in microglia, not in synaptic elements, exacerbate phagocytosis after maternal dietary n-3 PUFA deficiency.** N-3 PUFA deficient diets cause fatty acid alterations in whole brain tissue[23], yet it is unclear to what extent individual cell types or cell compartments incorporate these changes. To assess this, we profiled the fatty acid composition of microglia and synaptosomes isolated from n-3 sufficient and n-3 deficient mice (Fig. 3a, b). We observed that maternal n-3 PUFA deficiency significantly altered the fatty acid profile of both microglia and synaptosomes, with an increase in n-6 fatty acids and a decrease in n-3 fatty acids (Fig. 3a, b). In particular, DHA was decreased in microglia from n-3 deficient mice (Fig. 3a), while C22:5n-6 (or DPA n-6), the n-6-derived structural equivalent of DHA, was increased as a marker of n-3 PUFA deficiency[24] (Fig. 3a). Total AA was not significantly modified by the diet in microglia (Fig. 3a), unlike total AA levels in the whole hippocampus (Supplementary Fig. 4). In synaptosomes, total AA and DPA n-6 levels were significantly increased while total DHA level was significantly decreased and DPA n-6 inversely increased, as observed in microglia (Fig. 3b). In synaptosomes, more fatty acids

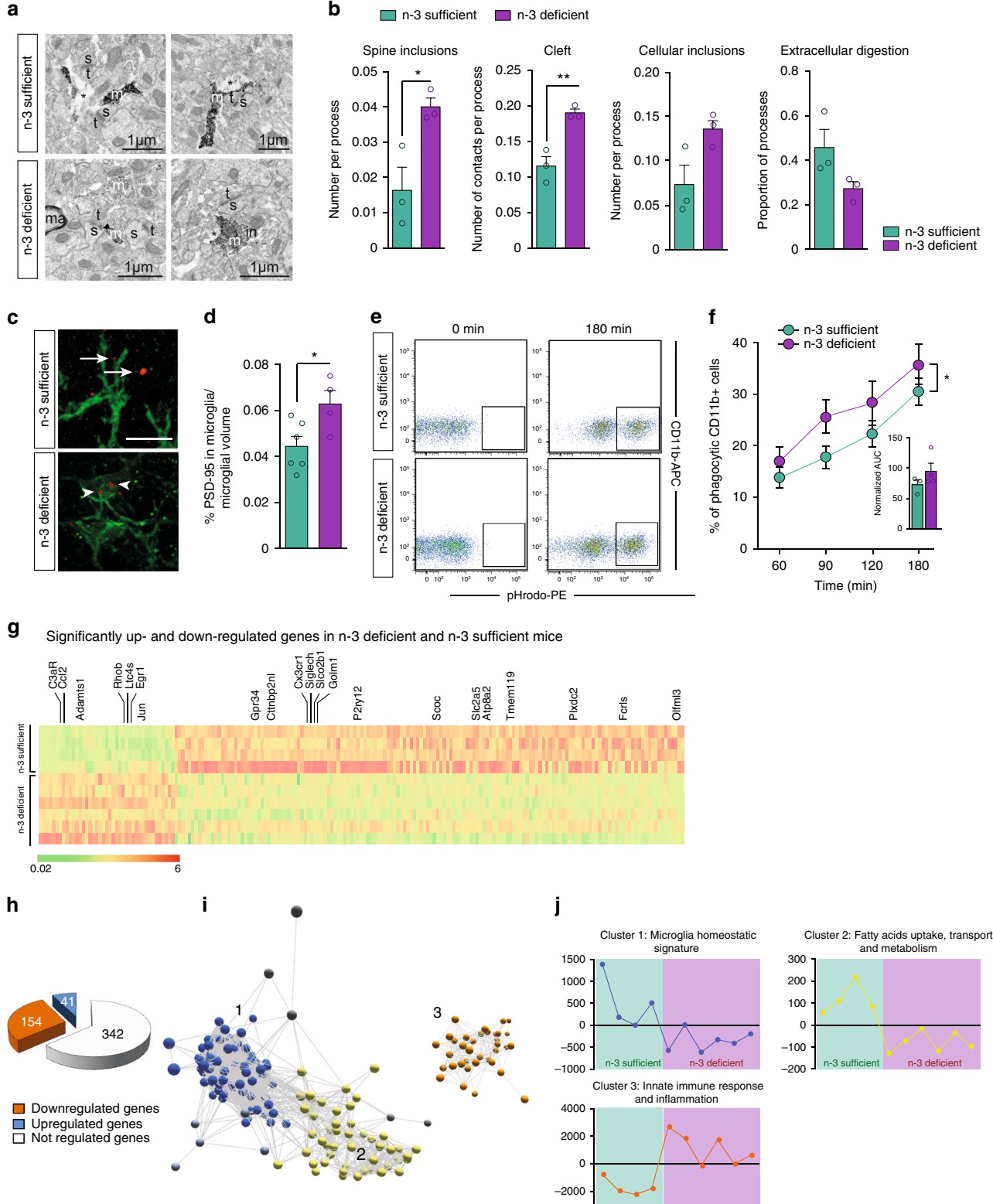

were significantly modified by the diet, such as the saturated fatty acid (SFA) stearic acid (18:0) and docosanoic acid (C22:0), and the monounsaturated fatty acids (MUFA) C16:1 n-7, C16:1 n-9, C18:1 n-9, and C20:1 n-9. Overall, our results show that n-3 and n-6 PUFA profiles of both microglia and synaptosomes are altered by low maternal n-3 PUFA intake, displaying similarities and discrepancies (Fig. 3a, b).

To directly evaluate the effect of microglial and synaptosome fatty acid alterations on microglial phagocytic activity, we developed an ex vivo assay in which we exposed freshly isolated n-3 sufficient or n-3 deficient microglia to n-3 sufficient or n-3 deficient pHrodo-labelled synaptosomes (Fig. 3c). We measured microglial phagocytic activity based on the level of cellular accumulation of pHrodo fluorescence (Fig. 3c). Phagocytosis was

**Fig. 2 Maternal n-3 PUFA deficiency increases microglial phagocytosis and gene expression profile. a, b** Representative images (**a**) and quantification (**b**) of EM data from the CA1 region of n-3 sufficient and n-3 deficient mice. s spine, m microglial processes, t terminals, * extracellular debris, ma myelinated axons. Arrowheads point to synaptic clefts. Scale bars $= 1 \mu m$. Two-tailed unpaired Student's t-test; $t = 5.24$, **$p = 0.0063$, cleft; $t = 3.366$, *$p = 0.0282$, spine inclusions; $t = 2.601$, $p = 0.06$, cellular inclusions; $t = 2.119$, $p = 0.1015$, extracellular digestion. **c** Three-dimensional reconstructions of PSD95-immunoreactivity (red) outside (arrowheads) or colocalized with (arrows) microglial cells (green). **d** Quantification of PSD95 and Iba1 immunoreactivity colocalization in the CA1 region of n-3 deficient and n-3 sufficient group. Means ± SEM; $n = 4–6$ mice per group. Two-tailed unpaired Student's t-test; $t = 2.583$, *$p = 0.032$. **e** Representative bivariate dot plots of isolated microglial cells gated on $CD11b^+/CD45^{low}$ expression from n-3 sufficient and n-3 deficient mice. **f** FACS analysis of phagocytic uptake of pHrodo-labeled *E. Coli* bioparticles by $CD11b^+$ microglial cells from both dietary groups after 180 min of incubation (expressed as the percent of cells that are $CD11b^+$ and PE positive). Means ± SEM; $n = 3–4$ per condition. Two-way ANOVA; diet effect: $F(1, 20) = 5.628$, *$p = 0.028$; time effect, $F(3,20) = 10.04$, $p = 0.0003$; Interaction, $F(3,20) = 0.1771$, $p = 0.91$. The area under the curve is presented as an independent graph in **c**. Two-tailed unpaired Student's t-test, $t = 1.289$, $p = 0.25$. **g** Heatmap of significantly up or downregulated genes in each dietary group. Each lane represents one animal ($n = 4$ or 6 per group). The 20 homeostatic microglia unique genes that were significantly affected by the diet are labelled. **h** Number of microglial genes that are up-regulated, down-regulated or not regulated by the diet. **i** A transcript-to-transcript correlation network graph of transcripts significantly differentially expressed by diet groups was generated in Miru (Pearson correlation threshold $r \geq 0.85$). Nodes represent transcripts (probe sets), and edges represent the degree of correlation in expression between them. The network graph was clustered using a Markov clustering algorithm, and transcripts were assigned a color according to cluster membership. **j** Mean expression profile of all transcripts within clusters 1, 2, and 3 where each point represents an animal and the average gene expression of all genes within that cluster for that animal. Source data are provided as a Source data file.

| Table 1 Composition of the diets (g/kg diet). | |
| --- | --- |
| **Ingredient** | **Amount** |
| Casein | 180 |
| Cornstarch | 460 |
| Sucrose | 230 |
| Cellulose | 20 |
| Fat[a] | 50 |
| Mineral mix[b] | 50 |
| Vitamin mix[c] | 10 |

[a]For detailed composition, see Table 2.
[b]Composition (g/kg): sucrose, 110.7; $CaCO_3$, 240; $K_2HPO_4$, 215; $CaHPO_4$, 215; $MgSO_4,7H_2O$, 100; NaCl, 60; MgO, 40; $FeSO_4,7H_2O$, 8; $ZnSO_4,7H_2O$, 7; $MnSO_4,H_2O$, 2; $CuSO_4,5H_2O$, 1; $Na_2SiO_7,3H_2O$, 0.5; $AlK(SO_4)_2,12H_2O$, 0.2; $K_2CrO_4$, 0.15; NaF, 0.1; $NiSO_4,6H_2O$, 0.1; $H_2BO_3$, 0.1; $CoSO_4,7H_2O$, 0.05; $KIO_3$, 0.04; $(NH_4)_6Mo_7O_{24},4H_2O$, 0.02; LiCl, 0.015; $Na_2SeO_3$, 0.015; $NH_4VO_3$, 0.01.
[c]Composition (g/kg): sucrose, 549.45; retinyl acetate, 1; cholecalciferol, 0.25; DL-α-tocopheryl acetate, 20; phylloquinone, 0.1; thiamin HCl, 1; riboflavin, 1; nicotinic acid, 5; calcium pantothenate, 2.5; pyridoxine HCl, 1; biotin, 1; folic acid, 0.2; cyanobalamin, 2.5; choline HCl, 200; DL-methionin, 200; p-aminobenzoic acid, 5; inositol, 10.

significantly increased in microglia from n-3 deficient mice, independent of the fatty acid profile of the synaptosome (Fig. 3c). These data show that both microglial and synaptosome fatty acid profiles are modified by maternal n-3 PUFA deficiency, but changes in microglial fatty acid composition drive the exacerbation of microglia-mediated synaptic loss.

**Maternal n-3 PUFA deficiency alters neuronal morphology and working spatial memory in a complement-dependent manner.** Recently, the classical complement cascade has been described as a key mediator of microglia synaptic refinement in the developing brain[25]. Moreover, our EM data revealed more spine inclusions within microglia and more contacts between these cells and the synaptic cleft (Fig. 2a, b). Thus, we explored the implication of the complement cascade in the exacerbation of microglia-mediated synaptic refinement observed in n-3 deficient mice.

The complement system acts as a recognition mechanism by which C1q, the initiating protein of the cascade, tags synapses to be removed. C1q-tagging leads to the cleavage of C3 and subsequent deposition of C3a and C3b, which activates CD11b and C3aR respectively on microglia and triggers synaptic elimination by microglial phagocytosis[25,26]. C1q immunostaining was significantly increased in both the hippocampus CA1 and DG

in n-3 deficient mice (Fig. 4a). Protein expression levels of CD11b (a subunit of CR3) was slightly, yet significantly increased in the DG in n-3 deficient mice (Fig. 4b). CD11b gene expression was also enhanced in the whole hippocampus in these mice (Supplementary Fig. 5). Our analyses also revealed a robust increase of C3aR immunostaining in the whole hippocampus of n-3 deficient mice, corroborating microglia transcriptomic data (Figs. 4c and 2g and Table 1). The C3aR-positive cells were uniformly distributed throughout the hippocampus, including the CA1. Expression levels of *Cx3cr1*, *Cx3cl1* and *Tgfb*, all involved in microglia-mediated synaptic pruning[13,27], were found to be increased in the whole hippocampus of n-3 deficient mice (Supplementary Fig. 5). Finally, we quantified the C3 protein on freshly extracted hippocampus synaptosomes from n-3 deficient and n-3 sufficient mice using an ELISA assay. We found higher amounts of C3 under n-3 PUFA deficiency (Fig. 4D). These findings indicate that low n-3 PUFA intake alters the developmental expression of complement cascade proteins both in microglia and at the synapse.

To test whether proper neuronal function could be restored by acting on the complement pathway in vivo, we antagonized CR3 activation in n-3 deficient mice. We first confirmed that a CR3 antagonist XVA-143, blocked n-6 PUFA (AA) induced increases in phagocytosis in vitro (Supplementary Fig. 6A–D). We then administered the CR3 antagonist, XVA-143[28] in vivo in the hippocampus four days before assessing the expression of the post-synaptic protein PSD95 at P21. Antagonizing CR3 prevented PSD-95 expression decrease in the hippocampus of n-3 deficient mice (Fig. 4e). Injection of XVA-143 also restored optimal memory abilities in n-3 deficient mice in the Y-maze task (Fig. 4f). These results show that complement activation increases microglia-mediated synaptic refinement when dietary n-3 PUFAs are low. Thus, a global increase in C3/CR3 interaction may account for the altered neuronal function observed in the hippocampus of n-3 deficient mice.

**Enhancement of microglia-mediated synaptic refinement by maternal PUFA deficiency is phosphatidyl serine recognition-independent.** Microglia possess a number of phagocytic receptors to eliminate neuronal elements, notably by binding to externalized phosphatidyl serine (PS) at the surface of neurons[29]. Since PUFAs are the main constituents of membrane phospholipids, we assessed whether microglial phagocytic activity was enhanced through PS recognition (Supplementary Fig. 7). First, we showed that cell-surface PS, stained with

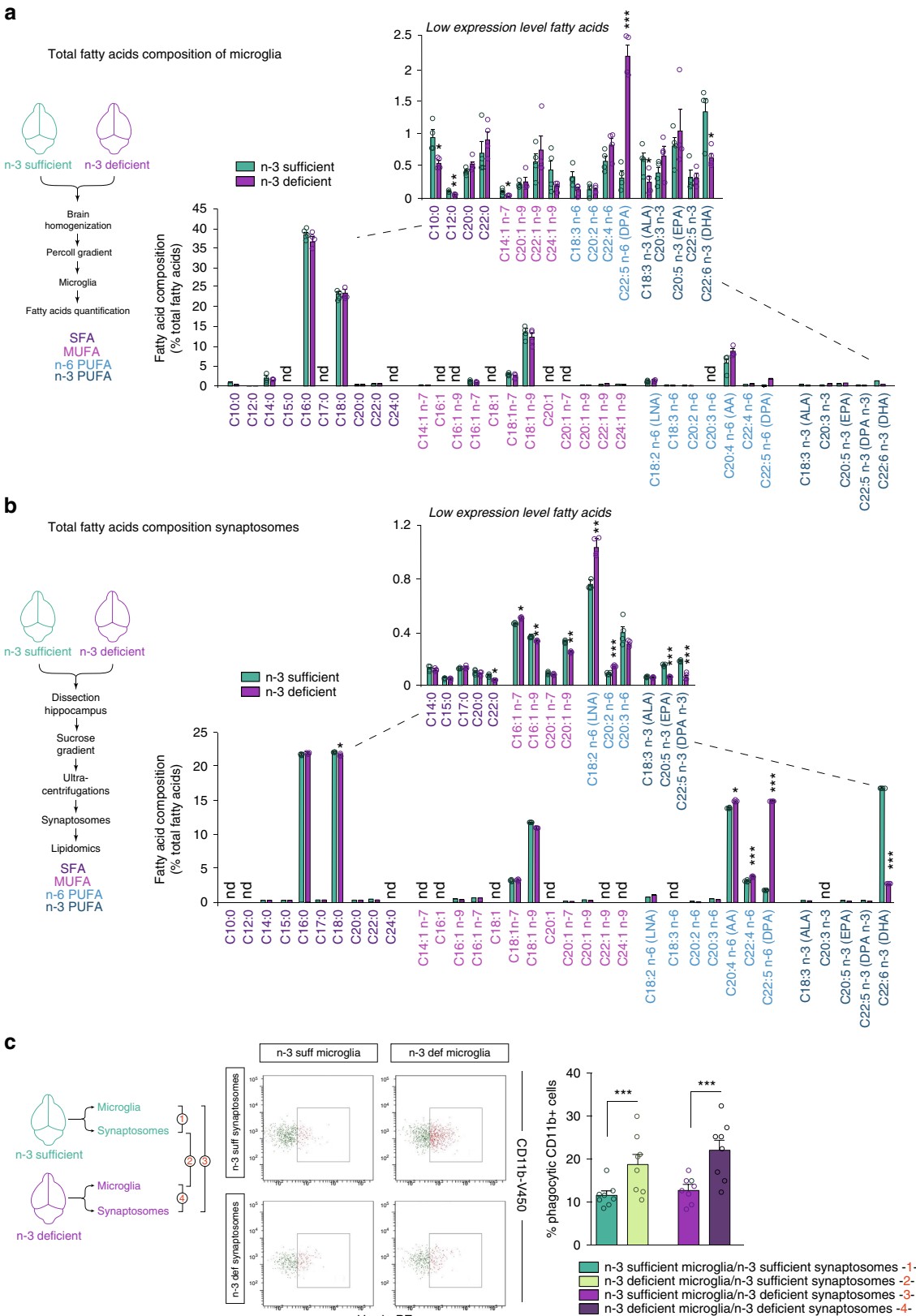

annexin-V, was unchanged between diets in the hippocampus (Supplementary Fig. 7A). We also quantified the expression levels of the microglial PS-binding protein milk fat globule EGF factor 8 (MFG-E8), which binds and activates a vitronectin receptor (VNR), TAM receptor tyrosine kinases (MER

and Axl), triggering receptor expressed on myeloid cells-2 (TREM2) and CD33[29,30]. Only MFG-E8 expression level was significantly increased by maternal n-3 PUFA dietary deficiency (Supplementary Fig. 7B–G). We administered a blocking peptide cRGD which inhibits the interaction between

**Fig. 3 Maternal dietary n-3 PUFA deficiency exacerbates microglial phagocytosis of synaptic elements by predominantly impacting microglia. a** Fatty acid composition of microglial cells sorted from n-3 sufficient and n-3 deficient mice. Means ± SEM; $n = 4$ mice per group. Insert: higher magnification of low expressed fatty acids. Two-tailed unpaired Student's $t$-test; $t = 3.37$, *$p = 0.015$, C10:0; $t = 4.852$, **$p = 0.0028$, C12:0; $t = 2.473$, *$p = 0.0482$, C14:1 n-7; $t = 9.646$, ***$p < 0.0001$, DPA n-6; $t = 2.558$, *$p = 0.043$, ALA; $t = 2.972$, *$p = 0.0249$, DHA; nd not detected. **b** Fatty acid composition of synaptosomes sorted from n-3 sufficient and n-3 deficient mice. Means ± SEM; $n = 4$ mice per group. Insert: higher magnification of low expressed fatty acids. Two-tailed unpaired Student's $t$-test; $t = 2.82$, *$p = 0.047$, C17:0; $t = 3.05$, *$p = 0.038$, C18:0; $t = 5.05$, **$p = 0.0072$ C18:1 n-9; $t = 2.99$, *$p = 0.04$, C18:2 n-6; $t = 26$, ***$p < 0.0001$, C20:2 n-6; $t = 3.57$, *$p = 0.023$, C22:4 n-6; $t = 12.68$, ***$p = 0.0002$, DPA n-6; $t = 4.49$, *$p = 0.011$, C20:5 n-3; $t = 7.12$, **$p = 0.0021$, DHA; nd not detected. **c** FACS analysis of phagocytic uptake of pHrodo-labeled synaptosomes (sorted from n-3 sufficient or n-3 deficient mice) by CD11b$^+$ microglial cells (sorted from n-3 sufficient or n-3 deficient mice). Analyses were performed 2 h post-synaptosomes application. On FACS plots: $x$-axis = (pHrodo intensity (PE intensity), $y$-axis = SSC. Means ± SEM; $n = 8$ per condition. Two-way ANOVA: microglial fatty acid status effect, $F_{(1,28)} = 19.77$, ***$p = 0.0001$; synaptosomes fatty acid status effect, $F_{(1,28)} = 1.63$, $p = 0.2123$; interaction, $F_{(1,28)} = 0.2826$, $p = 0.599$. Source data are provided as a Source data file.

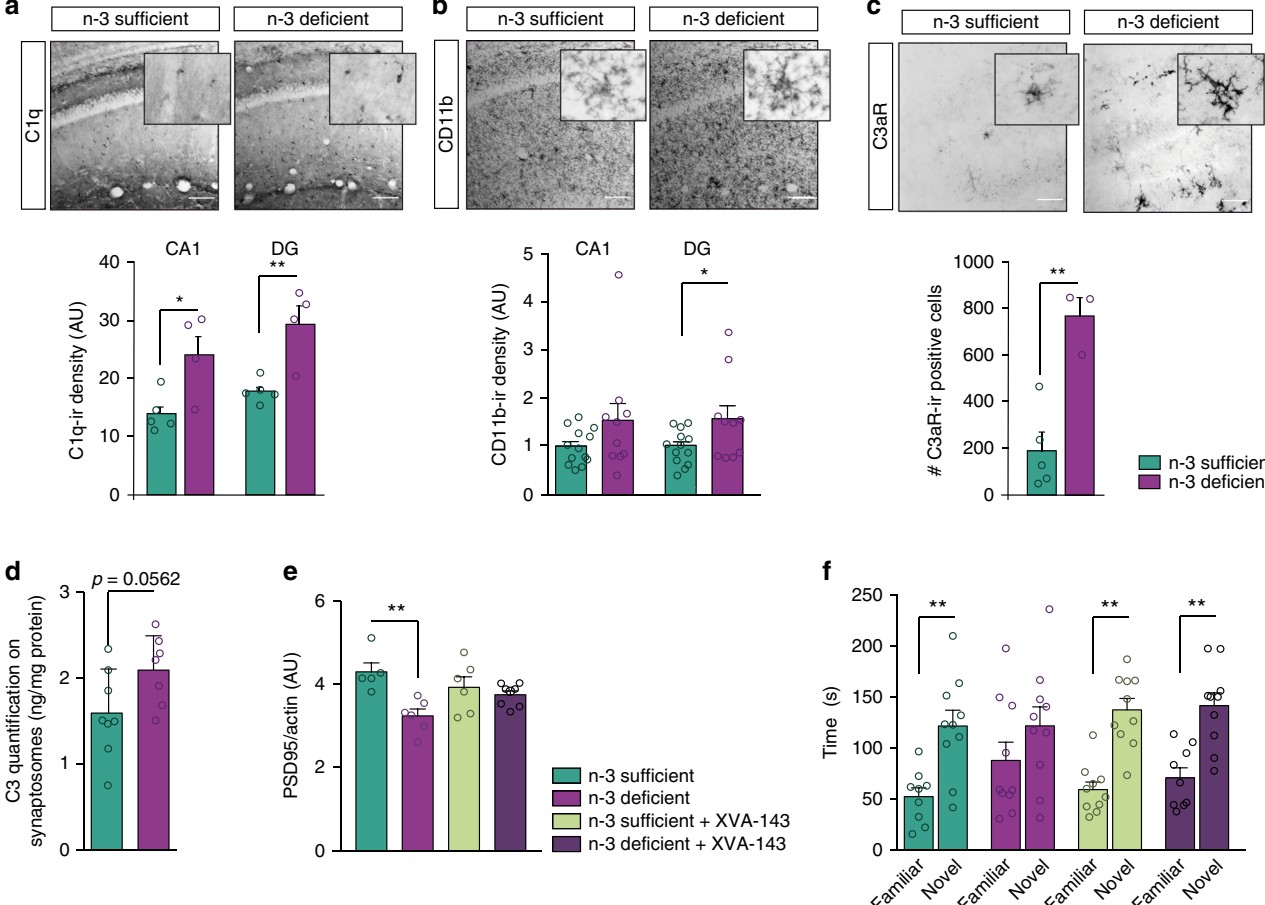

**Fig. 4 Maternal n-3 PUFA deficiency exacerbates microglia-mediated shaping of neuronal networks in a complement-dependent manner. a–c** Representative images and quantification of C1q, CD11b, and C3aR immunostaining in coronal sections of n-3 deficient and n-3 sufficient mice hippocampus. Scale bar = 100 μm. Means ± SEM; $n = 3$–13 mice per group. Two-tailed unpaired Student's $t$-test; $t = 1.553$, $p = 0.135$, CD11b CA1; $t = 2.114$, *$p = 0.466$, CD11b DG; $t = 0.5495$, *$p = 0.0223$ C1q CA1; $t = 3.938$, **$p = 0.0056$, C1q DG; $t = 4.91$, **$p = 0.0027$, C3aR, total hippocampus. **d** Protein quantification (ELISA) reveals that synaptosomes from n-3 deficient mice express more C3 than n-3 sufficient animals. Means ± SEM; $n = 7$–8 mice per group; Two-tailed unpaired Student's $t$-test; $t = 2.097$, $p = 0.0562$. **e** Quantification of PSD95 protein expression in n-3 sufficient vs n-3 deficient mice treated with XVA-143 or its vehicle. Means ± SEM; $n = 5$–8. Two-way ANOVA followed by Bonferroni post hoc test: diet effect, $F_{(1,21)} = 11.89$, $p = 0.0024$; treatment effect, $F_{(1,21)} = 0.1436$, $p = 0.708$; interaction, $F_{(1,21)} = 5.977$, $p = 0.0234$; n-3 sufficient vs n-3 deficient, **$p < 0.01$. **f** Time spent in novel vs familiar arm in the Y maze task in n-3 sufficient vs n-3 deficient mice treated with XVA-143 or its vehicle. Means ± SEM; $n = 9$–10 mice per group. Paired $t$-test: n-3 sufficient group, **$p = 0.0019$; n-3 deficient group, $p = 0.357$; n-3 sufficient + XVA-143 group, **$p = 0.0013$; n-3 deficient + XVA-143 group, **$p = 0.0099$. Source data are provided as a Source data file.

MFG-E8 and VNR vs a scrambled control (sc-cRGD), to n-3 deficient vs n-3 sufficient mice animals and quantified the expression of the post-synaptic protein PSD-95. Blocking MFG-E8 action on microglia did not prevent the reduction in PSD-95 expression in the hippocampus of n-3 deficient mice (Supplementary Fig. 7H). These results indicate that modulation of the microglial phagocytic capacity by early-life PUFAs is PS recognition-independent.

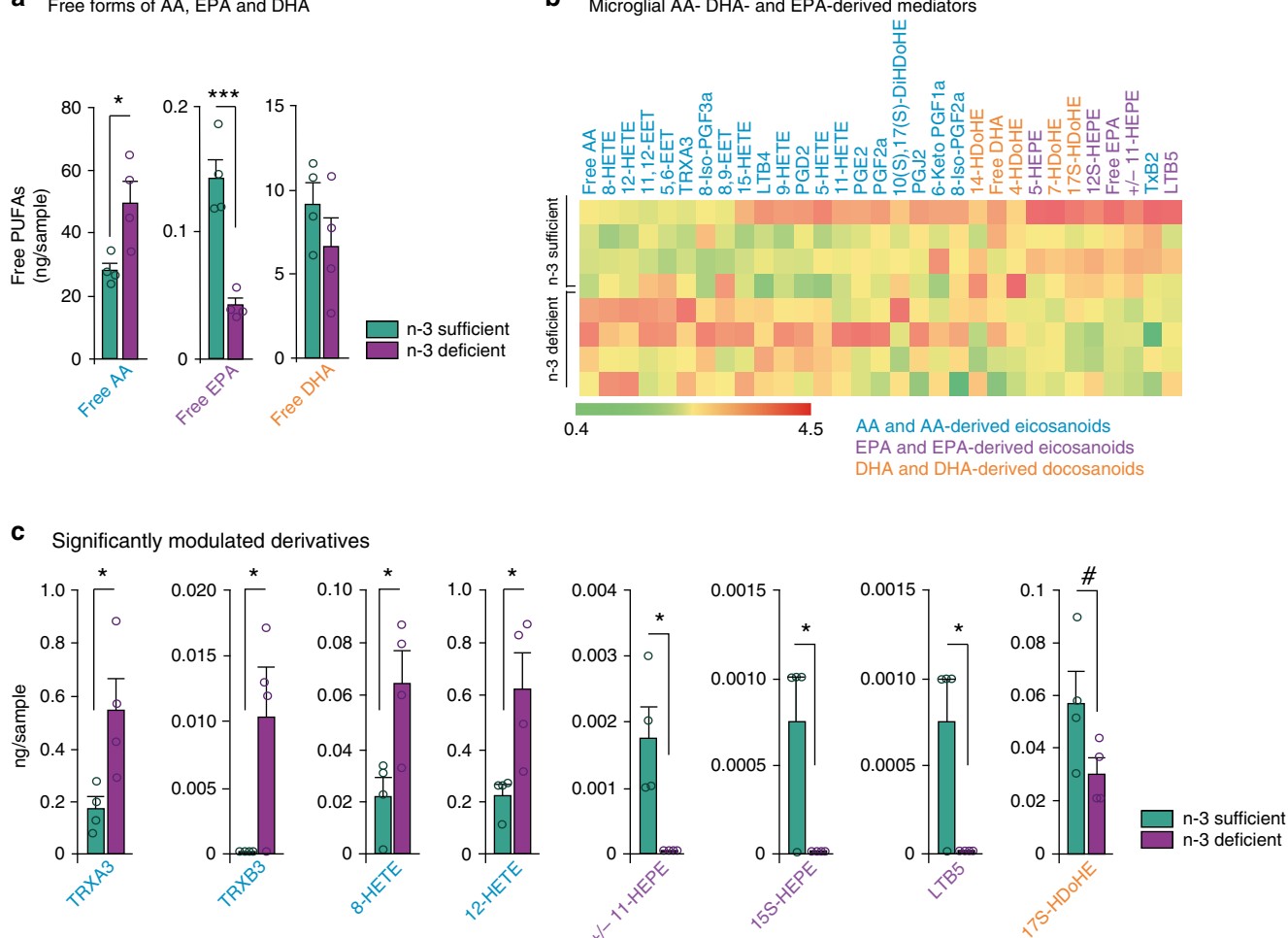

**Fig. 5 Maternal n-3 PUFA deficiency alters fatty acid profile in microglia. a** Quantification of free (unesterified) forms of AA, EPA, and DHA levels in microglia. Means ± SEM; $n = 4$ mice per group. Two-tailed unpaired Student's $t$-test; $t = 3$, *$p = 0.024$, free AA; $t = 6.12$, ***$p = 0.0009$, free EPA; $t = 1.149$, $p = 0.2942$, free DHA. **b** Heat map of all AA-, EPA-, and DHA-derived intracellular mediators expressed by microglia. **c** AA- EPA- and DHA-derived mediators that are significantly modulated by early-life n-3 PUFA deficient diet. Means ± SEM; $n = 4$ mice per group. Two-tailed unpaired Student's $t$-test; $t = 2.693$, *$p = 0.0359$, TRXA3; $t = 2.867$, *$p = 0.0286$, TRXB3; $t = 3.028$, *$p = 0.0231$, 8-HETE; $t = 2.886$, *$p = 0.0278$, 12-HETE; $t = 3.656$, *$p = 0.0106$, +/−11-HEPE; $t = 3$, *$p = 0.024$, 15S-HEPE; $t = 3$, *$p = 0.024$, LTB5; $t = 1.975$, $p = 0.0957$, 17S-HDoHE. Source data are provided as a Source data file.

**Maternal n-3 PUFA deficiency alters microglial PUFA metabolism in the offspring.** Our data suggest that n-3 PUFA deficiency specifically drives exacerbated microglial phagocytosis (Fig. 3). We, therefore, assessed the mechanisms by which microglial PUFA metabolism was driving their phagocytic activity. Microglia isolated from n-3 deficient mice had higher levels of free, unesterified, AA, and lower quantities of free EPA, as determined by high-performance liquid chromatography with tandem mass spectrometry (LC-MS-MS) (Fig. 5a). When unesterified from the cellular membrane, free PUFAs are rapidly converted to bioactive mediators (or oxylipins) with potential effects on microglial cells[1,31,32]. Interestingly, microglia from n-3 deficient animals presented greater levels of AA-derived bioactive mediators and lower amounts of DHA- and EPA-derived mediators (Fig. 5b). Four AA-derived oxylipins were significantly increased by maternal n-3 PUFA deficiency: trioxilins TRXA3 and TRXB3, 8-HETE, 12-HETE, while three EPA-derived species were significantly decreased: +/−11-HEPE, 15S-HEPE, and LTB5. The DHA-derived 17SHDoHE was also reduced ($p = 0.0957$) (Fig. 5c). Hence, we show the post-natal microglia oxylipin signature in the

context of low maternal n-3 PUFA intake and how it is modified by the dietary content in n-3 PUFAs.

**Activation of a 12/15-LOX/12-HETE signaling pathway increases microglial phagocytic activity under maternal n-3 PUFA deficiency.** Using pharmacological approaches, we continued to dissect the molecular mechanisms linking n-3 PUFA deficiency and microglial phagocytosis of dendritic spines by assessing the oxylipins identified above. We first tested the potential role of AA, EPA, and DHA free forms and their bioactive metabolites identified in Fig. 5 (except trioxilins TRXA3 and TRXB3, which are not commercially available) on microglial phagocytosis in vitro. We applied synaptosomes that were conjugated to a pH-sensitive dye (pHrodo), as a surrogate of synaptic refinement processes on cultured microglia (Fig. 6a). Twenty-four-hour application of AA on microglial cells significantly increased their phagocytic activity towards synaptosomes (Fig. 6b). These observations confirmed our data showing that AA increased phagocytosis of latex beads by primary microglial cells (Supplementary Fig. 6A, B). Conversely, incubation with

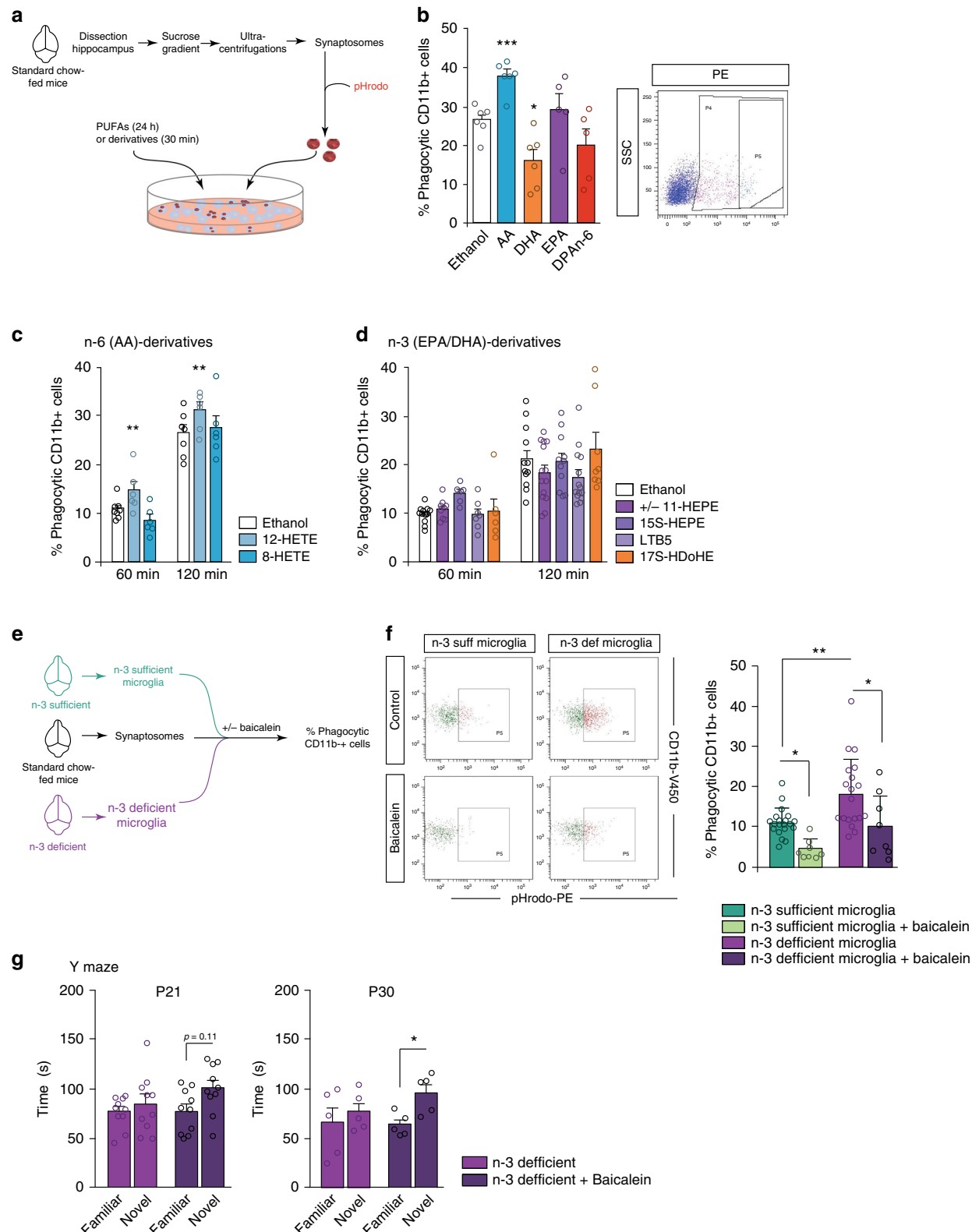

DHA significantly decreased phagocytosis (Fig. 6b), while EPA and DPA n-6 had no effect when compared to controls (Fig. 6b). Then, we applied PUFA bioactive mediators for 30 min before applying pHrodo synaptosomes and studied the kinetics of phagocytosis over the following 120 min. We show that the AA-derived 12-HETE significantly increased microglia phagocytosis of synaptosomes while 8-HETE had no effect (Fig. 6c). None of

the EPA- and DHA-derived bioactive mediators affected primary microglia phagocytic activity (Fig. 6d). For all conditions, the percentage of phagocytic cells is significantly higher at 120 min post-treatment (Fig. 6c, d).

AA is metabolized into 12-HETE via the 12/15-LOX enzyme and its activity is blocked by baicalein, which reduces 12-HETE production[33,34]. We then tested whether inhibiting 12/15-LOX

**Fig. 6 Maternal n-3 PUFA deficiency exacerbates microglial phagocytic activity towards synapses by activating the 12/15-LOX/12-HETE signaling pathway. a** Experimental setup. **b** FACS analysis of phagocytic uptake of pHrodo-labeled synaptosomes by CD11b$^+$ microglial cells in primary culture exposed to PUFAs. Means ± SEM; $n = 5$–6 per condition. Two-tailed unpaired Student's $t$-test; ***$p = 0.0005$, AA; *$p = 0.0118$, DHA; $p = 0.55$, EPA; $p = 0.153$; DPA n-6. **c**, **d** FACS analysis of phagocytic uptake of pHrodo-labeled synaptosomes by CD11b$^+$ microglial cells in primary culture exposed to n-6 AA-derived (**c**) or n-3 EPA- and DHA-derived (**d**) lipids. Means ± SEM; $n = 6$–14 per condition. AA derivatives: Two-way ANOVA: time effect, $F_{(1,33)} = 171.8$, ***$p < 0.0001$; treatment effect, $F_{(2,33)} = 5.538$, **$p = 0.0084$; interaction, $F_{(2,33)} = 0.686$, $p = 0.51$. EPA and DHA derivatives: Two-way ANOVA: time effect, $F_{(1,64)} = 57.25$ ***$p < 0.0001$; treatment effect, $F_{(2,64)} = 1.462$, $p = 0.239$; interaction, $F_{(2,64)} = 1.904$, $p = 0.157$. **e** Experimental setup. **f** FACS analysis of phagocytic uptake of pHrodo-labeled synaptosomes by freshly sorted n-3 deficient and n-3 sufficient CD11b$^+$ microglial cells, exposed to baicalein or its solvent. Analyses were performed 2 h post-synaptosomes application. Two-way ANOVA: microglia fatty acid status effect, $F_{(1,51)} = 12.6$ ***$p = 0.0008$; treatment effect, $F_{(1,51)} = 16.06$, $p = 0.0002$; interaction, $F_{(1,51)} = 0.1912$, $p = 0.6638$. **g** Time spent in novel vs familiar arm in the Y maze task in P21 and P30 n-3 deficient mice treated with baicalein or its vehicle. Means ± SEM; $n = 10$ (P21) or 5 (P30) mice per group. Paired $t$-test: P21: n-3 deficient group, $p = 0.5$; n-3 deficient + baicalein group, $p = 0.11$; P30: n-3 deficient group, $p = 0.57$; n-3 deficient + baicalein group, *$p = 0.029$. Source data are provided as a Source data file.

activity was able to restore normal phagocytic activity in n-3 PUFA deficient microglial cells. We used an ex vivo assay in which we exposed freshly sorted n-3 sufficient or n-3 deficient microglia to pHrodo synaptosomes with or without the 12/15-LOX inhibitor baicalein (Fig. 6e). We show that baicalein significantly reduced synaptosome phagocytosis by n-3 deficient microglia (Fig. 6f). Injection of baicalein persistently restored optimal memory abilities in n-3 deficient mice in the Y-maze task (Fig. 6g), without altering anxiety level (Supplementary Fig. 8A). Baicaelin had no effect in n-3 sufficient mice (Supplementary Fig. 8B). These results confirm that the 12/15-LOX/12-HETE pathway is active in n-3 deficient microglia to increase their phagocytic activity towards synaptic elements. In addition, baicalein improves memory deficits in n-3 PUFA deficient mice.

**The 12/15-LOX/12-HETE signaling pathway acts upstream of the complement cascade.** We finally assessed the relationship between 12/15-LOX/12-HETE signaling pathway and the expression of genes from the complement cascade under n-3 PUFA deficiency. We exposed freshly sorted n-3 sufficient or n-3 deficient microglia to the 12/15-LOX inhibitor baicalein or its vehicle and quantified the expression level of CD11b and C3aR mRNAs, 2 h later (Fig. 7a). We show that $c3ar$ gene expression was increased in n-3 deficient microglia while the expression of $cd11b$ was not significantly modulated by the diet. This is in line with our immunostaining experiments in which CD11b was moderately increased under n-3 PUFA deficiency, only in the DG, while C3aR was robustly increased in the whole hippocampus (Fig. 4b). Baicalein significantly reduced C3aR expression in n-3 deficient microglia, while it had no effect on CD11b (Fig. 7b, c). These results suggest that in non-stimulated n-3 deficient microglia, the 12/15-LOX/12-HETE signaling pathway is implicated in the maintenance of a high C3aR tone, while the activity of CD11b (as shown in Fig. 4e, f) is likely to depend on other mechanisms.

Once activated, the expression of most immune receptors is dampened as part of a global regulatory mechanism[35]. We thus examined the relationship between the 12/15-LOX/12-HETE signaling pathway and the complement cascade in the condition of stimulation of microglia with synaptosomes. We exposed freshly sorted cells to pHrodo synaptosomes in the presence of baicalein or its vehicle and quantified the expression level of $cd11b$ and $c3ar$ (Fig. 7d). The expression level of CD11b was significantly increased by baicalein in n-3 deficient microglia, whereas inhibiting the 12/15 LOX enzyme did not affect C3aR mRNA expression (Fig. 7e, f). This suggests that once microglia phagocyte synaptosomes, the 12/15-LOX/12-HETE signaling pathway inhibits CD11b expression while allowing phagocytosis of neuronal material (Fig. 6f). We finally studied the expression levels of BLT2 and GPR31, the two principal 12-HETE receptors

(Supplementary Fig. 9A). While GPR31 was expressed by the cells, BLT2 gene expression was almost not measurable in microglia in vitro. Hence, we then focused on the expression of GPR31 in vivo, on freshly isolated n-3 deficient and n-3 sufficient microglia, treated with vehicle or baicalein, in the absence or presence of synaptosomes. We could not find any significant modification of GPR31 mRNA expression level in any of the conditions tested (Supplementary Fig. 9B, C). Further experiments are needed to decipher whether 12 HETE effect on phagocytosis is mediated by GPR31.

Altogether, our data suggest that the 12/15-LOX/12-HETE signaling pathway modulates microglial phagocytosis under n-3 PUFA deficiency, partly by regulating the expression of the C3a and C3b receptors.

## Discussion

The phagocytic activity of microglia during development is crucial in shaping neuronal networks, and dysfunction of this activity contributes to neurodevelopmental disorders[15]. Here we show that low maternal n-3 PUFA intake-induced modification of microglia fatty acid composition is a potent driver leading to exacerbation of spine phagocytosis in the offspring. This process is complement-dependent, a mechanism that has been repeatedly reported to be the molecular bridge of microglia phagocytosis of spines[14,27,36]. Our data also reveal that n-3 PUFA deficiency alters the shaping of hippocampal neurons and spatial working memory in pups. Finally, we identified a previously unknown function of 12-HETE, a LOX metabolite of AA, in the regulation of microglial phagocytosis. This microglial 12/15-LOX/12-HETE pathway is also likely to contribute to alterations in spine density and connectivity observed in synaptopathies and neurodevelopmental disorders[37]. Considering that n-3 PUFAs also determine the offspring's microglia homeostatic molecular signature and fatty acid profile, our study suggests that the maternal nutritional environment is essential for proper microglial activity in the developing brain. These findings further support the importance of dietary n-3 PUFAs in the neurodevelopmental trajectory, pointing to their role in microglia-spine interactions.

Previous clinical and epidemiological studies revealed a relationship between dietary n-3 PUFA content, brain development, and the prevalence of neurodevelopmental disorders[38]. The n-3 PUFA index, which reports the level of n-3 PUFAs (EPA + DHA) in erythrocytes and that is used as a biomarker for cardiovascular diseases, is now also considered a biomarker for several neurodevelopment diseases such as ASD[39], attention deficit hyperactivity disorder (ADHD)[40] and schizophrenia[41]. The lack of n-3 PUFA during pregnancy and early post-natal life could participate in the etiology of these diseases. Since these essential fatty acids are necessary for healthy brain development, low dietary supply or impairment of their metabolism is suggested to be

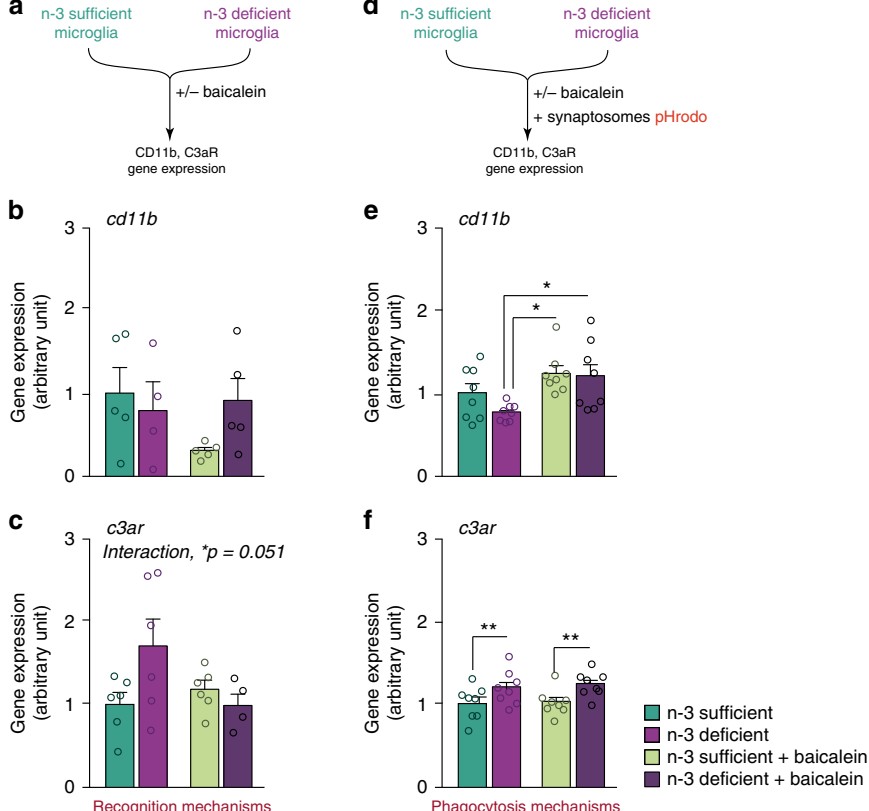

**Fig. 7 The 12/15-LOX/12-HETE signaling pathway controls gene expression of the complement pathway. a** Experimental setup. **b, c** Quantification of *cd11b* and *c3ar* mRNA expression in freshly sorted n-3 deficient and n-3 sufficient microglia, treated with baicalein or its vehicle. Means ± SEM; $n = 4–6$ per group. Two-way ANOVA: *cd11b*: diet effect, $F_{(1,15)} = 0.57$, $p = 0.46$; treatment effect, $F_{(1,15)} = 1.37$, $p = 0.26$; interaction, $F_{(1,15)} = 2.48$, $p = 0.14$; *c3ar*: diet effect, $F_{(1,18)} = 1.36$, $p = 0.26$; treatment effect, $F_{(1,18)} = 1.52$, $p = 0.23$; interaction, $F_{(1,18)} = 4.36$, *$p = 0.051$. **d** Experimental setup. **e, f** Quantification of *cd11b* and *c3ar* mRNA expression in freshly sorted n-3 deficient and n-3 sufficient microglia, exposed to synaptosomes for 2 h and treated with baicalein or its vehicle. Means ± SEM; $n = 8$ per group. Two-way ANOVA followed by Bonferroni post hoc test: *cd11b*: diet effect, $F_{(1,28)} = 0.16$, $p = 0.19$; treatment effect, $F_{(1,28)} = 0.92$, **$p = 0.0028$; interaction, $F_{(1,28)} = 0.072$, $p = 0.36$; n-3 deficient + vehicle vs n-3 deficient+ baicalein, *$p = 0.014$; n-3 deficient + vehicle vs n-3 deficient+ baicalein, *$p = 0.029$; *c3ar*: diet effect, $F_{(1,28)} = 0.35$, **$p = 0.0034$; treatment effect, $F_{(1,28)} = 0.0043$, $p = 0.72$; interaction, $F_{(1,28)} = 0.0019$, $p = 0.81$. Source data are provided as a Source data file.

involved in the etiology of several psychiatric diseases, including neurodevelopmental disorders[1]. Indeed, decreased DHA levels have been reported in several brain regions of patients diagnosed with such disorders[42]. Low n-3 PUFA index is also associated with reduced brain size or connectivity, as well as to reduced cortical functional connectivity during an attentional task[43]. Of special interest is the limbic system as the hippocampus activity in Humans has been reported to be altered in subjects with low n-3 PUFA status and consistently altered in animal models of n-3 PUFA deficiency[44,45]. Hippocampal alterations are reported in several neurodevelopmental disorders, including ASD, that could account for a neurodevelopmental compensatory mechanism to preserve cognitive domains[46]. Preterm infants display lower brain DHA concentrations as compared to term infants on the same ALA fortified diet[47]. Very low birth weight is a high-risk factor for both n-3 PUFA deficiency and neurodevelopmental disorders/ cognitive impairments[48]. These children display improved recognition memory at 6-months of age when supplemented with LC n-3 PUFA[49] while the brain structure was unchanged later in life (8 years old)[50]. Whether the pathophysiology of cognitive alterations, a common feature of neurodevelopmental disorders, may be linked to inadequate bioavailability of n-3 PUFA in the hippocampus is of high interest in part due to the increasing knowledge on the role of nutrition in psychiatric disorders and in particular ASD, ADHD and schizophrenia.

Overall, this knowledge suggests the use of specific dietary supplementation with n-3 PUFA as an environmental modifiable factor. Indeed, dietary supplementation or increase intake of food rich in n-3 PUFA could counteract the adverse effect of low n-3 PUFA intake. Our data shed light on an oxylipin-dependent mechanism underlying these clinical observations.

Our data are in line with previous studies showing that the 12/ 15-LOX/12-HETE pathway is altered in several pathological conditions in macrophage/microglia and especially in inflammatory conditions. Human macrophages incubated with 12-HETE showed induced gene expression after lipopolysaccharide (LPS) application[51]. Indeed, 12-HETE in macrophages was shown to activate extracellular signal-regulated kinase 1/2 (ERK1/ 2) and nuclear factor kappa-light-chain-enhancer of activated B cells (NFκB) via GPR31[52]. In humans, it was found that the expression of 12/15-LOX was increased in CD68-positive activated microglia[53]. Moreover, 12/15-LOX and Gpr31 receptor gene expression is increased in microglia along with aging[22] and in DAM/MGnD[54]. Here, no change in *gpr31* gene expression was observed in microglia treated with 12-HETE in vitro or exposed to a n-3 PUFA deficient diet in vivo. However, downstream analysis of GPR31 activation after 12-HETE application or expression should be tested.

Previous studies conducted in rodents and primates showed that early-life dietary n-3 PUFA deficiency impairs learning and

memory in adulthood[55,56]. Here, we show that perinatal n-3 PUFA deficiency-induced memory deficits occur as soon as weaning. This corroborates observations in humans that prenatal and cord blood n-3 PUFA levels are positively correlated with later-life cognitive abilities in infants[57]. In our study, cognitive alterations were associated with a decrease in dendritic length, whereas the complexity of dendritic arborization was unchanged, supporting the previous reports[58]. A significant decrease of post-synaptic scaffold proteins PSD-95 and Cofilin in the hippo-campus of n-3 deficient mice at weaning, suggests an early alteration of synaptic plasticity(Madore et al., 2014). This is in line with a previous study showing that loss of PSD-95 is inversely correlated with DHA dietary supply in a mouse model of Alz-heimer's disease[59] suggesting that PSD-95 expression depends on DHA levels in a broader context than just neurodevelopment. N-3 PUFA deficiency-mediated dendritic arborization alteration has been associated with functional defects of neuronal networks in the adult hippocampus and cortex[60,61] and atypical whole-brain functional community structure[62], a surrogate of cognitive defi-cits[63]. Finally, our finding of abnormal hippocampal neuronal spine density in n-3 PUFA deficient mice at weaning corroborates data previously found[58].

Lipid changes driving metabolic alterations in microglia and their impact on phagocytosis in the developing brain are under-studied. Knowing that microglia are now regarded as key con-trollers of synaptic architecture due to their phagocytic activity[13,14,16], we show that specific fatty acid mediators drive microglia phagocytic activity toward synapses. Our findings of a decrease in n-3 PUFAs and increase in n-6 PUFAs in the whole hippocampus, microglia, and synaptosomes of n-3 deficient mice is consistent with a large body of literature[24,64]. It is well estab-lished that in response to n-3 PUFA deficiency, brain DHA decreases and is replaced by DPA n-6[24,64]. We have previously shown that microglia from n-3 deficient dams have higher levels of DPA n-6[65]. While the mechanisms of this replacement at the level of the microglia are not clear, in whole brain, the uptake of DPA n-6 from the plasma free pool increases dramatically[66].

Oxylipins are involved in the regulation of inflammation and phagocytic activity of immune cells[1]. Their production is largely influenced by n-3 PUFA dietary supply, including in the brain[32]. Using LC-MS-MS, we examined their role in microglial phago-cytic activity during development. We show that n-3 PUFA deficient microglia have a unique oxylipin profile, linked to altered free forms of AA, EPA, and DHA. AA and its LOX-derived mediator 12-HETE promote microglia phagocytosis of synaptosomes. While there is no data available in the literature on microglia, this pathway has already been described as essential in the clearance of apoptotic cells by peripheral macrophages during inflammation[67]. Therefore, we predict that the 12/15-LOX enzyme is involved in AA-mediated increase in microglia pha-gocytosis in vitro. Conversely, DHA (but not its LOX derivative, 17S-HDOHE) decreased microglial phagocytosis in vitro. Pre-vious work reported that DHA inhibits microglia phagocytic activity of myelin both in vivo and in vitro[68] and normalizes inflammation-induced microglial phagocytic activity of spines in vitro[69]. Hence, it is likely that DHA dampens microglial phagocytosis via a repressive effect on synaptosome phagocytosis and/or a decrease in pro-inflammatory factors expression, in accordance with previous work showing its anti-inflammatory activity both in vitro and in vivo[70–73]. Additionally, recent work found that DHA decreases microglia phagocytic activity of myelin debris in the spinal cord after injury through mir-124[68], a non-coding nucleotide involved in synaptic plasticity. In macrophages, mir-124 has been shown to regulate actin-related protein 2/3 (ARP2/3), which is involved in the formation of the phagocytic cup[74]. This is in line with results showing that DHA reduced LPS-induced phagocytic activity of microglia, through a reduction of filopodia[75].

12-HETE is produced by the 12/15 LOX enzyme and conver-sion of AA by 12-HETE is enhanced in n-3 PUFA deficient microglia in the context of brain injury, autoimmune and neu-rodegenerative diseases[53]. 12/15 LOX is also expressed in the developing human brain by CD68-positive microglia and oligo-dendrocytes[53]. We show that blocking activity of 12/15-LOX by baicalein[34] attenuates the excessive n-3 PUFA deficient microglia phagocytic activity, which is coherent with the role of this enzyme in mediating phagocytic activity of apoptotic cells by macro-phages[67]. Hence, we describe a cell-specific fatty acid profile of microglia and provide a comprehensive description of the pro-cesses involved in n-3 PUFA deficiency-induced microglial alterations, including 12/15-LOX/12-HETE pathway as the molecular trigger enhancing the phagocytic activity of microglia toward spines.

We further deciphered the relationship between the 12/15-LOX/12-HETE pathway and the complement system. Our data reveal that the phagocytic capacity of microglia is increased only when the cells are n-3 deficient, even though C3 is overexpressed on n-3 deficient synaptosomes. These data suggest that both opsonization of synapses and recognition of C3 by its microglial receptors are required to trigger phagocytosis. This confirms a plethora of reports on the complement-dependent phagocytosis in the immune system. It is already well described that there is little to no ingestion of opsonized elements without phagocyte activation[76]. Conversely, CR3-mediated phagocytosis is depen-dent on receptor conformational changes, lateral diffusion, and clustering, processes that require activating stimuli[77]. Hence, binding of C3 to the complement receptors can be considered as the recognition ("find-me") mechanism of spines to be removed by microglia, as previously shown in other experimental contexts[14,78], while activation of the 12/15-LOX/12-HETE path-way is the triggering signal for phagocytosis by microglia. This is coherent with the observation that the complement cascade classically interacts with inflammatory pathways to control the activity of phagocytes[79].

Interestingly, C3a, that binds to microglial C3aR, is usually considered as a chemoattractant signal, attracting immune cells including microglia toward the structures to be cleared, while C3b, that binds to CR3 (CD11b + CD18) opsonizes the target to be removed[79]. Our data show that the 12/15-LOX/12-HETE pathway controls the basal expression of C3aR in n-3 deficient microglia, suggesting that it may control the capacity of microglia to reach the target. However, the 12/15-LOX/12-HETE pathway does not regulate the gene expression of CD11b, which is in line with the observations that CD11b activation is more dependent on conformational changes and redistribution in the cellular membrane rather than on transcriptional regulation[77]. As for CD11b activity, our data suggest that the 12/15-LOX/12-HETE pathway is involved in the homeostatic down-regulation of CD11b expression once the opsonized elements have been recognized by microglia. Overall, our data suggest that the 12/15-LOX/12-HETE pathway is activated upstream of the complement cascade, controlling the gene expression level and activity of this latter. We also found that 12/15-LOX/12-HETE pathway mod-ulates microglial phagocytic activity as baicalein reduces synap-tosomes engulfment. More experiments should be conducted to decipher the interplay of 12/15-LOX/12-HETE and the comple-ment pathways in the phagocytic activity of n-3 deficient microglia.

Deficits in microglia-mediated synaptic refinement lead to dysfunctional neuronal networks and behavioral abnormalities resembling some aspects of neurodevelopmental disorders[15,80,81]. Here, we show that n-3 PUFA deficient microglia display an

altered homeostatic molecular profile at weaning with features similar to neurodegenerative microglia, previously reported as more phagocytic[54]. We further identified the complement system, and not PS exposure, as the molecular mechanism driving spine phagocytic activity upon n-3 PUFA deficiency. This is consistent with previous work showing that in the hippocampus, PS-dependent and complement-independent trogocytosis of pre-synaptic elements remodels neuronal network during normal brain development[82,83] while complement-dependent phagocytosis of synapses is promoted in models of brain diseases presenting neuronal network abnormalities, including neurodevelopmental disorders[13,14,38]. Of note, neither PS recognition nor the complement cascade seem to be involved in synaptic refinement in n-3 sufficient mice. This suggests that (1) other mechanisms of microglia-mediated synaptic refinement are in place in the hippocampus of these animals or (2) microglia-mediated synaptic refinement is not prominent in the hippocampus of these animals. Microglia use a plethora of recognition mechanisms for phagocytosis of neuronal elements/dead bodies[33,84] but few studies have addressed the microglia-spine interaction in the developing hippocampus. On the other hand, several mechanisms, including microglia-independent processes such as neuronal macroautophagy or phagocytosis by astrocytes, control synaptic pruning in the developing brain, though the proportion of synaptic pruning relying on microglia in the developing hippocampus is not known[85]. Our data suggest that synaptic pruning mechanisms in the developing hippocampus of n-3 sufficient mice may favor microglia-independent processes.

Overall, our data also describe a role for the 12/15-LOX/12-HETE oxylipin pathway as a crucial metabolic signaling axis in the regulation of n-3 PUFA deficient microglia activity. We also delineate that maternal dietary n-3 PUFA deficiency drives microglia in the offspring into a complement-dependent phagocytic phenotype, aberrantly reducing synaptic elements, which drive hippocampal synaptic network dysfunction and spatial working memory deficit. Our findings not only support the necessity of adequate n-3 PUFA status during brain development[86], but also reveal that maternal lipid nutrition, by modulating microglial fatty acid metabolism and sensing, can be an important determinant of neurodevelopmental synaptopathies such as ASD. Regarding the clinical implications, these data highlight the relevance of diagnosing DHA/EPA deficits in expectant mothers and in early-life. We provide knowledge on highly targetable cellular and molecular pathways linking maternal nutrition and neurodevelopmental disorders. Indeed, specific dietary strategies can enhance PUFA status in microglia[65], being promising avenues for the prevention and treatment of neurodevelopmental disorders.

## Methods

Animal experiments were carried out according to the Quality Reference System of INRA and approved by the local ethical committee for care and use of animals (#APAFIS 4198and 15533). Male and female CD1 mice 8-week old were purchased from Janvier Labs (Le Gesnest St Isle, France) and were SPF-housed in temperature and humidity-controlled cages on a 12 h light/dark cycle with food and water ad libitum. The use of CD1 mice, an outbred strain, implies intrinsic inter-individual differences which we consider as an asset, considering the translational perspective of our research.

**Diet**. After mating, CD1 females were single housed and randomly assigned to either a n-3 deficient or a n-3 sufficient diet, both containing 5% fat, in the form of sunflower oil (rich in LA, the "n-3 deficient diet") or a mixture of different oils of which rapeseed oil (rich in ALA, the "n-3 sufficient diet") throughout gestation and lactation[20] (Tables 2 and 3). The composition of the standard chow diet is closer to the n-3 deficient diet in terms of LA, total n-6 PUFA, total PUFAs, while it is more similar to the n-3 sufficient diet in terms of LA/ALA ratio and ALA. Hence, we used the n-3 sufficient diet as our control as it more consistently contains higher levels of n-3 PUFAs and lower levels of n-6 PUFAs than the n-3 deficient diet. The content of proteins, carbohydrates, and lipids varies as well (3.1% of lipids in the

**Table 2 Fatty acid composition of the diets (% wt of total fatty acids).**

| Diets | Deficient | Sufficient | Standard chow (A04) |
|---|---|---|---|
| 16:0 | 7.3 | 22.6 | 20.3 |
| 18:0 | 4.1 | 3.3 | 2.2 |
| Other saturated FAs | 1.6 | 1.8 | 1.8 |
| Total saturated FAs | 13.0 | 27.7 | 24.3 |
| 18:1n-9 | 28.1 | 57.9 | 19.3 |
| 18:1n-7 | 0.9 | 1.5 | 1.5 |
| Other monounsaturated FAs | 0.2 | 0.4 | 3.9 |
| Total monounsaturated FAs | 29.4 | 60.0 | 24.7 |
| 18:2n-6 (LA) | 57.4 | 10.6 | 45.9 |
| Other n-6 PUFAs | n.d. | n.d. | 0.3 |
| Total n-6 PUFAs | 57.4 | 10.7 | 46.2 |
| 18:3n-3 (ALA) | 0.2 | 1.6 | 3.3 |
| 20:5 n-3 | n.d. | n.d. | 0.6 |
| 22:5 n-3 | n.d. | n.d. | 0.1 |
| 22:6 n-3 | n.d. | n.d. | 0.8 |
| Total n-3 PUFAs | 0.2 | 1.6 | 4.8 |
| Total PUFAs | 57.6 | 12.3 | 51.0 |
| LA/ALA | 287 | 6.6 | 13.9 |

FAs fatty acids, PUFAs polyunsaturated fatty acids, LA linoleic acid, ALA α-linolenic acid, n.d. not detected (under the limit for the detection by gas chromatography, <0.05%).

standard chow vs 5% in the n-3 sufficient diet). All experiments were performed on the offspring from n-3 deficient and n-3 sufficient dams at post-natal day (P) 21. Pups from different dams were used for each measurement to avoid any cage effect. Each mother had 2–3 litters and was put back on a standard diet for a minimum of one month between each litter.

**Drugs**. Cyclo(Arg-Gly-Asp-D-Phe-Val) (cRGD; Bachem, H-2574) and its control, scrambled, peptide c(RADfV) (or sc-cRGD; Bachem, H-4088) were dissolved in acetic acid (1N) and further diluted into aCSF to a final concentration of 1 mM. A volume of 1 μl of cRGD or sc-cRGD was injected in the CA1 region of the hippocampus, bilaterally, at P14. Animals were euthanized at P21 and processed for Western blot analysis.

XVA-143 (kindly provided by Hoffmann-La Roche) was dissolved in aCSF at 5 mM. In all, 1 μl was injected bilaterally in the CA1 region of the hippocampus at P17. Animals were used at P21 for behavioral and biochemical analyses. For primary microglial cell cultures, XVA-143 was applied 30 min before beads incubation at a final concentration of 1 μM.

Baicalein (Sigma-Aldrich, #465119) was applied at a final concentration of 20 μM in 0,2% DMSO for ex vivo experiments. For behavioral assessment, baicalein was injected intraperitoneally at a concentration of 20 mg/kg.

**Cortex fatty acid composition**. Fatty acid composition was studied in the cortex (not hippocampus) of animals as: (1) cortex and hippocampus fatty acid composition vary similarly under the diets used in this study; (2) to reduce the number of animals. Cortical fatty acids were extracted according to the method of Folch, fatty acids were transmethylated according to the method of Morrison and Smith and fatty acid methyl esters (FAMEs) were analyzed on a FOCUS GC gas chromatograph (Thermo Electron Corporation) equipped with a split injector and a flame ionization detector.

**Microglia isolation and sorting**. Brains were homogenized in Hanks' Balanced Salt Solution (HBSS), pH 7.4 passing through a 70 μm nylon cell strainer. Homogenates were centrifuged at 600 g for 6 min. Supernatants were removed and cell pellets were re-suspended in 70% isotonic Percoll (GE-Healthcare, Aulnay sous Bois, France). Single cell suspensions were prepared and centrifuged over a 37%/70% discontinuous Percoll gradient (GE Healthcare) at 2000g for 20 min, mononuclear cells were isolated from the interface. For each brain extraction ~3 × 10^5 cells were isolated. Cells were washed and in order to sort resident microglia from recruited myeloid cells, we used a monoclonal antibody that recognizes FCRLS, which is expressed on microglia, but not on infiltrating myeloid cells (Butovsky et al.[20]). Isolated cells were stained with rat anti-FCRLS [clone 4G11, 3 mg ml1, validated in Butovsky et al.[20]] and then followed by a secondary detection using an anti-rat IgG PE (Biolegend).

Cells were sorted using a FACS Aria 5-Blue 2-Violet 2-Red laser configuration (BD Biosciences).

**Table 3 List of all genes that are significantly modulated by maternal n-3 PUFA deficiency.**

| Up-regulated genes | Ratio n-3 deficient/ n-3 sufficient | | | | |
|---|---|---|---|---|---|
| Ccl3 | 50,2 | Slco2b1 | 0,55 | Ttr | 0,35 |
| Ccl4 | 18,9 | Golm1 | 0,55 | BMP6 | 0,35 |
| Rasgef1b | 9,5 | A430107D22Rik | 0,55 | NOX1 | 0,34 |
| Cd83 | 5,4 | D830030K20Rik | 0,54 | Nav2 | 0,34 |
| Atf3 | 5,3 | Vat1l | 0,54 | Mmp12 | 0,34 |
| Ccrl2 | 4,7 | LOC100038847 | 0,54 | Hist1h2ac | 0,34 |
| C3ar1 | 4,4 | Ifi202b | 0,54 | Siglec1 | 0,33 |
| Ccl2 | 4,1 | Rgmb | 0,54 | Plxna1 | 0,32 |
| Ppp1r15a | 3,5 | Ccr5 | 0,54 | Fcrls | 0,32 |
| Nlrp3 | 3,0 | ADAM17 | 0,54 | Slco4a1 | 0,31 |
| Apoe | 3,0 | Ppp1r9a | 0,53 | Cbr3 | 0,31 |
| Rgs1 | 2,9 | Rnf180 | 0,53 | Slc46a1 | 0,31 |
| Trim47 | 2,8 | FcgrIIIa (CD16a) | 0,53 | Upk1b | 0,29 |
| Adamts1 | 2,8 | P2ry12 | 0,53 | Khdrbs3 | 0,29 |
| Dab2 | 2,7 | CD200R | 0,52 | Tnfrsf17 | 0,29 |
| Klhl38 | 2,6 | Gpr165 | 0,52 | TREM1 | 0,28 |
| 1810011O10Rik | 2,6 | Fgd2 | 0,52 | Il7r | 0,28 |
| CEBPB | 2,5 | Trem2 | 0,51 | Ak1 | 0,28 |
| Myc | 2,4 | P4ha1 | 0,51 | BMP7 | 0,28 |
| Cd14 | 2,4 | Gtf2h2 | 0,51 | WNT5A | 0,27 |
| Fos | 2,3 | Zfp691 | 0,51 | Kitl | 0,24 |
| Il1a | 2,3 | Tmem144 | 0,51 | Chi3l1 | 0,22 |
| Rgl1 | 2,2 | Plxna4 | 0,51 | Cfb | 0,21 |
| Fosb | 2,2 | Itgax | 0,51 | Arg1 | 0,20 |
| Manba | 2,2 | Bend6 | 0,51 | Olfml3 | 0,18 |
| Npl | 2,1 | Csf3r | 0,51 | Gal3st4 | 0,17 |
| cathepsin | 2,0 | SEMA3C | 0,50 | | |
| Rhob | 2,0 | Bco2 | 0,50 | | |
| Ltc4s | 2,0 | MPO (myeloperoxidase) | 0,50 | | |
| Egr1 | 2,0 | ENSMUSG00000079376 | 0,50 | | |
| Fth1 | 1,9 | Erf | 0,50 | | |
| Abca1 | 1,9 | Tmc7 | 0,49 | | |
| Jun | 1,9 | 4933406P04Rik | 0,49 | | |
| Socs3 | 1,8 | Mmp2 | 0,48 | | |
| Junb | 1,7 | Tmem204 | 0,48 | | |
| SREBP1 | 1,7 | Tlr3 | 0,48 | | |
| Tlr2 | 1,6 | Spnb4 | 0,47 | | |
| Il1rl2 | 1,5 | Scoc | 0,47 | | |
| Ctsl | 1,5 | B930046C15Rik | 0,47 | | |
| 9030625A04Rik | 1,4 | Asph | 0,47 | | |
| SRA1 | 1,4 | WNT7A | 0,47 | | |
| | | Pycard | 0,46 | | |
| Down-regulated genes | | Sall1 | 0,46 | | |
| Icam1 | 0,82 | Pros1 | 0,46 | | |
| HIST1H2AB | 0,76 | Garnl3 | 0,46 | | |
| Ryk | 0,75 | Fabp5 | 0,45 | | |
| Grm1 | 0,75 | MR | 0,45 | | |
| Rtn1 | 0,75 | Camk2n1 | 0,44 | | |
| Tfeb | 0,74 | B4galt4 | 0,44 | | |
| Map3k7 | 0,72 | Slc2a5 | 0,44 | | |
| Slc24a3 | 0,71 | Il21r | 0,43 | | |
| Qdpr | 0,70 | IL34 | 0,43 | | |
| ADAM10 | 0,68 | Atp8a2 | 0,43 | | |
| Cxxc5 | 0,67 | Jam2 | 0,43 | | |
| Ckb | 0,67 | Myo1b | 0,42 | | |
| Ttc28 | 0,66 | Gpr56 | 0,42 | | |
| Tmem100 | 0,66 | Ebf3 | 0,42 | | |
| Nfia | 0,65 | Olfml2b | 0,42 | | |
| Tm9sf4 | 0,65 | Fgfr1 | 0,42 | | |
| Snn | 0,64 | Tmem119 | 0,42 | | |
| Abi3 | 0,64 | Scamp5 | 0,42 | | |
| Epn2 | 0,64 | Fads1 | 0,41 | | |
| Sema4d | 0,63 | Psd | 0,41 | | |
| Map2k1 | 0,62 | Tmeff1 | 0,41 | | |

## Table 3 (continued)

| Up-regulated genes | Ratio n-3 deficient/ n-3 sufficient | | |
|---|---|---|---|
| Zfpm1 | 0,62 | Ptprm | 0,41 |
| Rbbp9 | 0,62 | Tlr5 | 0,40 |
| Adora3 | 0,62 | Adamts16 | 0,40 |
| Ang | 0,61 | Eng | 0,40 |
| Gpr34 | 0,61 | CD36 | 0,39 |
| Rtn4rl1 | 0,61 | Itga9 | 0,39 |
| Bin1 | 0,60 | Gpr40 | 0,39 |
| Acp2 | 0,60 | Tspan18 | 0,39 |
| Epb4.1l2 | 0,60 | Ecscr | 0,39 |
| Cttnbp2nl | 0,60 | Cntn1 | 0,39 |
| Itgb5 | 0,59 | WNT2 | 0,39 |
| Sgce | 0,58 | Gp9 | 0,38 |
| D18Ertd653e | 0,58 | Arhgap22 | 0,37 |
| Gm10790 | 0,57 | Pon3 | 0,37 |
| Tjp1 | 0,57 | Plxdc2 | 0,37 |
| Spsb1 | 0,57 | Tanc2 | 0,37 |
| C3 | 0,57 | Rab6b | 0,36 |
| Lair1 | 0,57 | Kcnd1 | 0,36 |
| Tppp | 0,57 | Csmd3 | 0,36 |
| IL6ST | 0,56 | NRG1 | 0,36 |
| Cx3cr1 | 0,56 | Capn3 | 0,35 |
| Siglech | 0,56 | | |

Only genes with $P < 0.05$ are considered significant.

**Cell culture and treatments**. For in vitro phagocytic assays, primary mixed glial cell cultures were prepared from the cortices of P0–1 CD1 mice. After dissection in 0.1 M PBS with 6% glucose and 2% penicillin–streptomycin (Gibco, Cergy Pontoise, France) and removal of the meninges, the cortices were chopped and subsequently mechanically dissociated. The suspension was diluted in low glucose DMEM (31885, Gibco) supplemented with 10% fetal bovine serum (Gibco) and 1% penicillin–streptomycin. Microglia were isolated from primary mixed glial cultures on DIV14 using a reciprocating shaker (45 min at RT) and repeated rinsing with their medium using a 10 mL pipette. Media was subsequently removed, microglia pelleted via centrifugation ($504\,g \times 10$ min) and following resuspension maintained in DMEM at a concentration of $4 \times 10^5$ cells/mL in six-well culture plates. Cells were then treated for 24 h in serum-free medium containing 50 μmol/L fatty acid-free bovine serum albumin (BSA) added either with DHA, AA, DPAn-6 or EPA 30 μmol/L in 0.1% ethanol or 0.1% ethanol. For oxylipins, 30 min prior to the phagocytosis assay, culture medium was replaced by serum-free DMEM containing 100 nM of the given lipid in 0.03% ethanol, or 0,03% ethanol.

**Phagocytic assay**. Ex vivo assays: the phagocytic capacity of microglia was determined by the level of pHrodo™ Red fluorescence accumulated in the cells. First, we used pHrodo-conjugated *E.Coli* bioparticles. A suspension of $2.5 \times 10^5$ sorted microglia cells in 100 μL of RPMI/1% BSA was incubated with bioparticles in an incubator at 37 °C and 5% of $CO_2$ for each time point. Cells were then pelleted by centrifugation at $1000 \times g$ for 10 min at 4 °C, resuspended in 200 μL of RPMI/1 %BSA and stained with CD11b-APC [clone M1/70, eBioscience] and rat CD45-PerCP-Cy™5.5 [clone 30-F11, BDBiosciences] antibodies for quantification by flow cytometry. We also developed a model using pHrodo-conjugated synaptosomes as a substrate for the phagocytosis. Synaptosomes were prepared from P21 mice hippocampus, and labeled with pHrodo Red, succinimidyl ester (Thermo Fisher Scientific, P36600) according to manufacturer instructions. Synaptosomes were resuspended at a concentration of 10 mg/ml in DMEM. Microglia were plated at a density of $5.10^5$ cells/well in 24-well plates and received 150 μg of synaptosomes for 120 min in an incubator at 37 °C and 5% $CO_2$. When needed, baicalein was applied at a final concentration of 20 μM in 0,2% DMSO 90 min prior to the beginning of the assay (i.e. the addition of synaptosomes). Cells were collected by gentle trypsin treatment, rinsed and resuspended in PBS/1%BSA buffer to be stained with CD11b-V450 [clone M1/70, BDBiosciences], CD45 PerCP Cy5.5 [Clone 30-F11, BDBiosciences], Ly6G-APC [clone 1A8, BDBiosciences] and Ly6C-APC-Cy™7 [clone AL-21, BDBiosciences] both for negative exclusion and analysis by cytometry. After selection of the CD11b+/CD45low, cells were gated on PE channel for quantification of pHrodo fluorescence.

Experiments in Figs. 3c and 6f have been performed on the same batches in order to spare animals. Hence, data used for Fig. 3c control groups are a randomly chosen subset of Fig. 6f's data.

In vitro, two different substrates where used: FCS-coated Yelloworange fluorescent carboxylated microspheres (Fluoresbrite® YO Carboxylate Microspheres 3.00 μm, Polysciences Europe GmbH, #19393-5) or the pHrodo-conjugated synaptosomes as described above. To quantify their phagocytic index, primary microglial cell cultures were incubated with a suspension of microspheres at a concentration of $1.1 \times 10^7$ microspheres/ml for 30 min at 37 °C, intensively washed, and finally fixed with 4% paraformaldehyde. Cells were stained with anti-Iba1 antibodies. A blinded experimenter counted beads per cell using NIS Elements AR 3.26 software. To measure phagocytic activity by cytometry, cells were incubated for 60 or 120 min with 150 μg of synaptosomes per well. Medium was removed and cell collected following trypsin treatment, pelleted, rinsed in PBS/1% BSA, and stained with CD11b-FITC [clone M1/70, BDBiosciences] antibody for cytometry analysis.

FACS data were acquired using an LSR Fortessa 2-Blue 6-Violet 3-Red 5-YelGr laser configuration (BD Biosciences). Diva 8.0 (BD Biosciences) and FlowJo 10.5 (FlowJo, LLC) were used for data analyses.

**Microglia fatty acid analysis**. Fatty acid analyses requiring lots of material, we pooled microglia from 4 brains for $n = 1$. Based on our experience, we sort 400,000 cells out of the brain of P21 mice. Most measurements were made on around 1.6 million microglia. It is noted in ng/sample in the legend.

*Fatty acids*: Extraction and quantification of total fatty acids by GC-FID: Microglia from 3 to 4 brains per sample were lysed in 100% methanol and extracted via from the Folch method (2:1:0.8 chloroform: methanol: 0.88% KCl) and analyzed by a Varian-430 gas chromatograph (Varian, Lake Forest CA).

*Lipid mediators*: Isolation and quantification of brain eicosanoids and docosanoids by LC-MS-MS: Pooled microglia extracted from 3 to 4 brains per sample were lysed in 15% methanol and stored at −80 °C prior to analysis. One ng of internal standard mixture was added to each sample, and lipid mediators were extracted, analyzed by LC-MS-MS and quantified as previously described[87].

**Synaptosomes preparation and fatty acid quantification**. Synaptosomes were prepared from P21 mice hippocampus[88]. The two hippocampi of each animal were dissected and placed in 1 mL of ice-cold iso-osmolar buffer. The tissues were homogenized with 2 cm³ rotating potter with 12 strokes at 900 rpm. The homogenate was centrifuged at $1000 \times g$ for 5 min at 4 °C, the supernatant was collected in a new tube and centrifuged at $12,500 \times g$ for 8 min at 4 °C. The supernatant was removed and the pellet resuspended with 1 mL of ice-cold iso-osmolar buffer. The discontinuous sucrose gradients were prepared in ultracentrifuge tubes with ice-cold 1.2 M sucrose solution and 0.8 M sucrose solution. The resuspended pellet was placed on top of the sucrose gradient. The centrifugation at $50,000 \times g$ for 1 h at 4 °C provided a visible layer at the interface of sucrose gradients. This fraction

(synaptosomes) was collected with a syringe. Synaptosomes were then rinsed with 100 mM pH 8.5 sodium bicarbonate buffer and stored for further experiments.

Fatty acid quantification was run as already explained for the cortex fatty acid composition (see above).

**RNA isolation and Nanostring RNA counting.** Total RNA was extracted from isolated microglia using mirVanaTM miRNA isolation kit (Ambion) according to the manufacturer's protocol. We performed nCounter multiplexed target profiling of 542 microglial transcripts (MG550). MG550 encompasses 400 unique and enriched microglial genes we have identified previously[20] and additional 150 inflammation-, inflammasome- and phagocytosis-related genes. In all, 100 ng of total RNA per sample were used in all described nCounter analyses according to the manufacturer's protocol[20].

**Network analysis.** To investigate gene coexpression relationships between groups, a pairwise transcript-to-transcript matrix was calculated in the software tool Miru (Kajeka, UK) from the set of differentially expressed transcripts using a Pearson correlation threshold $r = 0.85$. A network graph was generated where nodes represent individual probe sets (transcripts/genes), and edges between them correlation of expression pattern with Pearson correlation coefficients above the selected threshold. The graph was clustered into discrete 6 groups of transcripts sharing similar expression profiles using the Markov clustering algorithm (inflation 2.2).

**Electron microscopy.** Mice were anesthetized with sodium pentobarbital (80 mg/kg, i.p.) and perfused with 0.2% glutaraldehyde in 4% PFA. Transverse sections of the brain (50 μm thick) were cut in PBS (50 mM at pH 7.4) using a vibratome and stored at −20 °C in cryoprotectant solution (30% glycerol and 30% ethylene glycol in PBS).

Sections were washed in PBS and quenched with 0.3% $H_2O_2$ in PBS for 5 min and then with 0.1% NaBH₄ for 30 min at room temperature (RT), washed in Tris–buffered saline (TBS; 50 mM at pH 7.4) and processed freely-floating for immunoperoxidase staining. Sections were pre-incubated for 1 h at RT in a blocking solution of TBS containing 10% fetal bovine serum, 3% bovine serum albumin, and 0.01% Triton X100, before overnight incubation at 4 °C in rabbit anti-Iba1 antibody (1:1000; Wako Pure Chemical Industries) and rinsed in TBS. After incubation for 1.5 h at RT in goat anti-rabbit IgGs conjugated to biotin (1:200 in blocking solution; Jackson Immunoresearch) and for 1 h with ABC Vectastain mix (1:100 in TBS; Vector Laboratories), the labeling was revealed using diaminobenzidine (DAB; 0.05%) and hydrogen peroxide (0.015%) in TBS. After immunostaining, sections were post-fixed flat in 1% osmium tetroxide and dehydrated in ascending concentrations of ethanol. They were treated with propylene oxide, impregnated in Durcupan (EMS) overnight at RT, mounted between ACLAR embedding films (EMS), and cured at 55 °C for 72 h. Areas of CA1 stratum radiatum were excised from the embedding films and re-embedded at the tip of resin blocks. Ultrathin (65–80 nm) sections were cut with an ultramicrotome (Leica Ultracut UC7), collected on bare square-mesh grids, and examined at 80 kV with a FEI Tecnai Spirit G2 transmission electron microscope.

Pictures were randomly taken at 9300X in the CA1 stratum radiatum of each animal, for a total surface of ~2000 μm² of neuropil captured per animal, using an ORCA-HR digital camera (10 MP; Hamamatsu). Cellular profiles were identified according to criteria previously defined[89]. For quantitative analysis, each captured Iba1-positive microglial process was analyzed. A phagocytic index was compiled by summing up the vacuoles and endosomes containing cellular materials such as membranes, axon terminals with 40-nm synaptic vesicles and dendritic spines with a postsynaptic density, on a microglial process basis[16]. Images were analyzed using the AMT Image Capture Engine Software 601.384.

**Immunohistochemistry.** Mice were deeply anesthetized with isoflurane and transcardially perfused with PBS followed by 4% PFA. Brain was removed, post-fixed in PFA overnight at 4 °C and cryoprotected in 30% sucrose at 4 °C. Immunohistochemistry experiments were performed on free-floating coronal 30 μm cryostat slices. The following antibodies were used: 1:1000 rabbit anti-Iba1 (Wako, #019-19741), 1:100 mouse anti-PSD95 (www.anticorps-en-ligne.fr, ABIN1304920), 1:500 rat anti-CD11b (AbD Serotec, #MCA711), 1:500 rabbit anti-C1q (Abcam, #ab182451), 1:20 mouse anti-C3aR (Hycult biotech, #HM1123), 1:1000 rabbit anti-GFAP (Dako, Z03334), 1:1000 mouse anti-NeuN (Millipore, #MAB377),1:200 rabbit anti-Annexin V (Abcam, #ab14196), 1:500 mouse anti-claudin 5 (Life Technologies: Invitrogen), 1:500 IRDye 800 conjugated affinity purified goat-anti-mouse IgG (Rockland, Gilbertsville, PA). Primary antibodies were visualized with appropriate secondary antibodies conjugated with Alexa fluorophores (Invitrogen) and counterstained with DAPI or with biotin. When biotinylated, secondary antibodies were revealed using the streptavidin-biotin-immunoperoxidase technique, giving a black precipitate.

**Image analysis.** Densitometry: Individual images were analyzed with Fiji or Image J software (Image J, open source), using the following procedure: (1) user-defined thresholding value applied to each image, (2) calculation of area of staining from background for each protein of interest. Final values are represented as a surface

area in pixel values. Control sections for all studies in which primary or secondary antibodies were omitted resulted in negative staining (not shown).

Stereological analysis: Iba1- and GFAP-immunoreactive cell numbers and volume of the hippocampus were thoroughly determined in the hippocampus with the unbiased stereological sampling method based on optical dissector stereological probe.

Apoptotic cells counting: The number of apoptotic cells (cells with pycnotic/karryorhectic morphology) was estimated using unbiased stereology methods, and is reported as cells/mm3.

**Quantitative real-time PCR.** Total RNA was extracted from hippocampi using TRIzol (Invitrogen, Life TechnologiesTM). RNA purity and concentration were determined using a Nanodrop spectrophotometer (Nanodrop technologies, Wilmington, DE). In all, 2 μg of RNA was reverse transcribed to synthesize cDNA using Superscript III (Invitrogen, Life TechnologiesTM) and random primer according to the manufacturer's protocol. Quantitative PCR were performed on 384-well plates using epMotion 5070 (Eppendorf). 10 μl of cDNA diluted 1:5 (20 ng/μl) were amplified by real-time PCR. Primer references: C1qa Mm00432142_m1; C3 Mm00437838_m1; TREM2 Mm00451744_m1; CD11b Mm01271262_m1 (for in vitro experiments); CD33 Mm00491152_m1; CX3CR1 Mm00438354_m1; CX3CL1 Mm00436454_m1; TGFb Mm03024053_m1; beta-2 microglobulin Mm00437762_m1 (housekeeping gene) (Life Technologie). For in vivo quantification of CD11b mRNA expression, we used SYBR Green technology. Ten μl of cDNA diluted 1/60 (1.66 ng/μL) were amplified by real-time PCR. Primers sequences: CD11b Forward AATGATGCTTACCTGGGTTAT GCT/Reverse TGA TAC CGA GGT GCTCCTAAAAC; Housekeeping gene beta-2 microglobulin: Forward CTGATACATACGCCTGCAGAGTTAA/Reverse GATC ACATGTCTCGATCCCAGTAG. For all experiments, the difference between target and housekeeping gene Ct values (ΔCt) was calculated to normalize for differences in the amount of total nucleic acid added to each reaction and in the efficiency of the RT step. The expression of target gene (linear value) normalized to the housekeeping gene was determined by $2^{-(\Delta Ct)}$.

**Droplet digital (dd)PCR.** Total RNA was extracted using the RNeasy® micro Kit (Qiagen, Germany) according to the manufacturer's protocol. The integrity of the RNA was checked by capillary electrophoresis using the RNA 6000 Pico Labchip kit and the Bioanalyser 2100 (Agilent Technologies, Massy, France), and quantity was estimated using a DS.11 (DeNovix, USA). The RNA integrity number (RIN) was above 7,5. cDNA was synthesized from 6 ng of total RNA using Qscript XLT cDNA supermix (QuantaBio). Quantitative PCR was used for the choice of the reference gene and perfomed using a LightCycler® 480 Real-Time PCR System (Roche, Meylan, France). Quantitative PCR reactions were done using transcript-specific primers, cDNA, and the LightCycler 480 SYBR Green I Master (Roche) in a final volume of 10 μl. For the determination of the reference gene, the Genorm method was used. The elongation factor 1-alpha 1 (Eef1a1) and peptidylprolyl isomerase A (Ppia) genes were used as reference genes.

ddPCR was used for the quantification of the interest genes. PCRs were prepared with cDNA and the required QX200 ddPCR EvaGreen Supermix (Bio-Rad) with a final concentration of 150 nM for each transcript-specific primer to a final volume of 20 μl. Each reaction was loaded into a sample well of an 8-well disposable cartridge (Bio-Rad) followed by 70 μl of droplet generator oil (Bio-Rad), which was added to the oil wells of the cartridge. Droplets were formed in the QX200 droplet generator (Bio-Rad). Droplets were then transferred to a 96-well PCR plate, heatsealed with foil in a PX1 PCR Plate Sealer Bio-Rad and amplified with an Eppendorf Nexus Gradient master cycler (95 °C primary denaturation/activation for 5 min, followed by 45 cycles of 95 °C for 30 sec and 61 °C for 1 min, followed by 4 °C 5 mn and a final 90 °C heat treatment for 5 min). PCRs were analyzed with the QX200 droplet reader (Bio-Rad) and data analysis was performed with QuantaSoft software (version 1.7; Bio-Rad). qPCR primer sequences: Eeflal (GenBank NM_010106): Forward TGAACCATCCAGGCCAAATC, Reverse: GCATGCTATGTGGGCTGTGT; Ppia (GenBank NM_008907): Forward CAAATGCTGGACCAAACACAA, Reverse: GCCATCCAGCCATTCAGTCT; C3ar1 (GenBank NM_009779): Forward TCCCATCTCTCCCTACTTTGCA, Reverse: TGTTTTAGGCACACCATGGTAAA; Itgam (GenBank NM_001082960): Forward CTCATCACTGCTGGCCTATACAA, Reverse: GCAGCTTCATTCATCATGTCCTT; Gpr31b (GenBank NM_001013832): Forward GCTGCAGTGTCCAGCAAGC, Reverse: TGTACTGTGCAGGCAGGTGAG.

**Golgi staining.** Brains were processed for Golgi-Cox staining using the FD Rapid Golgi Staining kit (FD Neurotechnologies, Inc). Brains were left in the staining solution for 12 days, frozen in isopentane solution and kept at −80 °C until sectioning and coloration. The brains were cryostat-cut at a thickness of 100 μm. Sections were mounted on gelatin-coated slides and analyzed using a motorized Leica DM5000 microscope at ×63 magnification. The images were acquired using a CCD Coolsnap camera and Metamorph software. For analysis, we randomly selected pyramidal neurons from the CA1 region of the dorsal hippocampus that were fully penetrated by the Golgi coloration and distinguishable from other neurons (n-3 deficient mice had less usable neurons than n-3 sufficient mice). Spine density in these neurons was determined by counting the number of spines on at least 3 basal and 3 apical dendritic segments of 10 μm in length. Segments

from dendrites situated as far from the cell body as possible, with no overlap with other dendrites were randomly selected. Primary dendrites were never used for analysis as their thickness hampers detection of spines. Spine density was calculated per 10 μm and averaged across the different segments in the same neuron. Images were processed with Mercator Pro 7.9.11.

**Western blotting**. Brains were carefully placed on a glass plate over dry ice to collect hippocampi that were immediately frozen and stored at − 80 °C until use. Samples were homogenized in lysis buffer plus anti-phosphatase solution (Tris/HCl 20 mM pH 7.4 with EDTA 1 mM, MgCl2 5 mM, dithiothreitol 1 mM, Na orthovanadate 2 mM, protease inhibitors cocktail 1X and Na fluoride 1 mM). Homogenates were centrifuged 10 min at $2200 \times g$ to remove nuclei. Supernatants were stored at −80 °C. Protein contents were determined by Bio-Rad protein assay according to the manufacturer's protocol (Bio-Rad) and then heated to 100 °C for 5 min in Laemmli sample buffer (2% sodium dodecyl sulfate and 5% dithiothreitol).

Equal quantities of proteins (20 μg/well) were electrophoresed onto an 8% or 12% polyacrylamide gel with a 4% stacking gel. Proteins were blotted on PVDF membranes (Immobilon, Millipore, Paris, France). Membranes were saturated by incubation with 5% milk in TBS and Tween 0.1% for 1 h. Blots were probed overnight at 4 °C with 1:1,000 rabbit anti-PSD95 (Cell signaling Technology, 3450 S), 1:500 rabbit anti-Bax (Santa cruz, SC-493), 1:500 rabbit anti-Bcl2 (Santa Cruz Biotechnology, SC-492), 1:1,000 mouse anti-Mer (R&D systems, AF591), 1:1,000 mouse anti-Axl (R&D systems, AF854), 1:1,000 rabbit anti-ERK1/2 (Cell Signaling Technology, 137F5), 1:500 goat anti-MFG-E8 (R&D systems, AF2805), 1:1000 rabbit anti-GluA1 (Santa Cruz Biotechnology, SC-28799), 1:1000 rabbit anti-GluA2 (Santa Cruz Biotechnology, SC-7611), 1:1000 goat anti-GluN2A (Santa Cruz Biotechnology, SC-1468), 1:1000 goat anti-GluN2B (Santa Cruz Biotechnology, SC-1469), 1:1000 rabbit anti-GluN1 (Santa Cruz Biotechnology, SC-31556), 1:5000 rabbit anti-SAP102 (Synaptic System, 124213), 1:300 mouse anti-cofilin (Abcam, Ab-54532), 1:5000 rabbit anti-actin (Cell Signaling Technology, 4967), 1:5000 rabbit anti-GAPDH (Cell Signaling Technology, 2118S). After washing in TBS-Tween, membranes were incubated with the secondary antibody coupled to Horse Radish Peroxidase (HRP, Southern Biotechnology Associates, Birmingham, AL, USA) diluted in TBS-Tween supplemented with 3% milk, for 2 h at room temperature. Membranes were washed and the complex was detected with an ECL kit (ElectroChemoLuminescence, Amersham, Orsay, France). Optical density capture of the signal obtained was performed with the Syngene Chemigenius2 apparatus (Synoptics, Cambridge, UK). Intensity of the signal was quantified using GeneTools software (Synoptics, Cambridge, UK).

**C3 ELISA assay**. The level of C3 protein located on synaptosomes was measured using the Mouse C3 (Complement Factor 3) ELISA Kit (Genway Biotech GWB-7555C7). Synaptosomes were prepared using the hippocampi of 2 mice per sample. After extraction, synaptosomes were lysed using sonication in TB. Quantifications were performed according to manufacturer instructions, in duplicates and by loading 5 μg of proteins extract per well, and using the following software Wallac 1420 instrument Workout 2.5 (PerkinElmer).

**Y maze task**. P21-old mice were handled daily and weighed before and during behavioral experiments. Sessions were recorded with a ceiling-mounted video camera and analyzed using Smart software (Panlab, Barcelona, Spain). The Y-maze was used to assess spatial working memory[90]. Each arm was 34 cm long, 8 cm wide, and 14 cm high. The floor of the maze was covered with corn cob litter, which was mixed between each trial to remove olfactory cues. Visual cues were placed in the testing room and kept constant during the whole test. In the first trial, one arm was closed with a guillotine door and mice were allowed to visit two arms of the maze for 5 min. After a 30-min to 3 h inter-trial interval (ITI), mice were placed back in the start arm and allowed free access to the three arms for 5 min. Start and closed arms were randomly assigned for each mouse. When required, baicalein was injected intraperitoneally at a concentration of 20 mg/kg, at P14, P17, and 30 min prior the first trial (at P21 or P30 depending on the experiments). Data are presented as the time spent exploring the novel and the familiar arms during the second trial.

**Statistical analyses**. For most experiments, experimental groups were compared using Student $t$-test or non-parametric Mann-Whitney test (when equality of variance or normality failed). We used the Grubbs' test, also called the ESD method (extreme studentized deviate), to determine whether the most extreme value in the list is a significant outlier from the rest. Paired t-tests were used for Y-maze experiments (familiar vs novel arm for each experimental group). A two-way ANOVA was used for XVA-143, cRGD, and baicalein experiments (except Y-maze), ex vivo and in vitro phagocytic activity. All data were expressed as means ± standard error of the mean (SEM). A $p < 0.05$ was considered as statistically significant. Statistics were performed using GraphPad Prism 7.0.

**Reporting summary**. Further information on research design is available in the Nature Research Reporting Summary linked to this article.

## Data availability

Source data are provided with this paper. All data supporting the findings are provided within this paper and its supplementary information. Fatty acids derivatives data are available on Metabolights. Our study is identified as MTBLS1952. Transcriptomic data are available in the GEO database. Accession number: GSE158181.

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

## Acknowledgements
Funding for this research was provided by the Institut National pour la Recherche Agronomique (INRAE), the Bordeaux Univ, the Foundation for Medical Research (FRM, DEQ20170336724 to S.L.), the French Foundation (FDF, #00070700) and the Fédération pour la Recherche sur le Cerveau (FRC Connect). C.M. was funded by the French Ministry for Research and by the Fondation pour la Recherche Medicale (ARF201909009101), CBB was supported by the French National Agency for Research (ANR-12-BSV4-0025-04, to S.L.) and Agreenskills (MSCA cofund-INRAE), J.C.D. was supported by the Region Nouvelle Aquitaine (2011.1303003, to S.L.), Q.L. was supported by the Region Ile de France (PICRI, the Ceberal Palsy Foundation #13020605, to A.N.) and the FRM, AT was supported by the French National Agency for Research (ANR-2010-BLAN-141403, to S.L.), ADG was supported by Agreenskills (MSCA cofund-INRAE) and Marie-Curie European Grant, C.L. was supported by Idex grant, M.M. was funded by FRM (DEQ20170336724, to S.L.) and M.R. was supported by the Region Nouvelle-Aquitaine (2017-1R30237-00013179, to S.L.). A.S. was supported by grants from the Spanish Ministry of Economy and Competitiveness with FEDER funds to AS (BFU2015-66689, RYC-2013-12817), a BBVA Foundation Grant for Researchers and Cultural Creators to A.S., and a Basque Government grant (PI_2016_1_0011) to AS. PG was supported by Inserm, Université de Paris, ERA-NET Neuron (Micromet). This work was partly funded by NeurATRIS ANR-11-INBS-0011 of the French Investissements d'Avenir Program run by the ANR. MET, C.L., and K.B. were supported by grants from NARSAD and the CIHR awarded to MET. We would like to thank C.Tridon, S. Delbary and B. Péré for taking care of the mice, Rémi Van der Vynckt for his assistance in the Western blot experiment, and Celine Lucas for technical assistance. This work benefited from the facilities and expertise of the imaging platform Imag'In (www.incia.u-bordeaux1.fr), which is supported by CNRS and Region Aquitaine and from the Bordeaux Imaging Center. We thank Atika Zouine and Vincent Pitard for their technical assistance at the Flow cytometry facility, CNRS UMS 3427, INSERM US 005, Univ. Bordeaux, F-33000 Bordeaux, France. Analysis of eicosanoids and docosanoids was performed at the Analytical Facility for Bioactive Molecules (AFBM) of the Centre for the Study of Complex Childhood Diseases (CSCCD) at the Hospital for Sick Children, Toronto, Ontario. CSCCD was supported by the Canadian Foundation for Innovation (CFI). We thank the Biochemistry and Biophysics Platform of the Bordeaux Neurocampus at the Bordeaux University funded by the LABEX BRAIN (ANR-10-LABX-43) and Yann Rufin for the Western Blot analysis.

## Author contributions
The data reported in this study can be found in the supplementary materials. C.M., Q.L., L.M., M.R., M.M., B.R., C.L., C.B.B., J.B., A.T., J.C.D., A.Sere., A.A., and V.D.P. performed most experiments. K.E.H. performed and R.P.B. oversaw the fatty acid analyses of microglia. S.B. and A.Sierra. performed and A.S. oversaw the apoptosis experiments. C.L., K.B., and M.E.T. performed and M.E.T. oversaw the E.M. experiments. S.G., N.A., and L.B. performed and analyzed fatty acid experiments on whole hippocampus. J.B. performed and analyzed BBB experiments. C.M. performed and O.B. oversaw transcriptomic analyses. A.D.G. analyzed transcriptomic data. N.J.G. performed FACS phagocytosis experiments. L.F. performed and analyzed TAM receptors Western blots. P.G. contributed to the design of the experiments. C.J. performed whole structure fatty acid composition experiments and conducted the data analyses. S.L. and A.N. equally supervised the entire project and wrote the paper. All authors proof-read the manuscript.

## Competing interests
The authors declare no competing interests.
