## [Peer Review File · Nature Communications]

Reviewers' Comments:

Reviewer #1:

Remarks to the Author:

This manuscript presents a potential molecular mechanism for detrimental effects of low maternal omega-3 fatty acid intake on hippocampal development. The authors report that low maternal dietary intake of omega-3 fatty acids increases microglial phagocytosis of synaptic elements in developing hippocampus through the activation of 12/15-LOX/12-HETE signaling, affecting neuronal morphology and cognitive function in offspring. While it may be a novel mechanism for potential n-3 deficiency-related suboptimal neurodevelopment, there are many questions to be addressed.

1. Figure 4H. Why did PSD-95 expression not increase with cRGD compared to sc-cRGD for the n-3 sufficient group? Given that 'pharmacological inhibition of the interaction between MFG-E8 and its microglial vitronectin receptor using a blocking peptide (cRGD) increased PSD-95 in standard diet-fed animals (Supplemental Fig. 5)', similar effects would be anticipated, even if 'the modulation of the microglial phagocytic capacity by early-life PUFA is PS recognition-independent.' Is there large difference in the brain fatty acid profile and gene/protein expression between n-3 sufficient and standard-diet-fed animals? The fatty acid profile and gene expression data from all three diet groups should also be clearly presented, particularly because the biological difference is supposedly driven by the lipid composition.
2. The DHA content seems very low in synaptosomes. What is the total fatty acid composition in the cortex or hippocampus? Supplemental Figure 4 should be expressed as weight percentage rather than the ratio.
3. The authors showed that antagonizing CR3 by administering XVA-143 directly in the hippocampus four days before assessing protein expression at P21 prevented PSD-95 expression decrease in the hippocampus of n-3 deficient mice and restored optimal memory ability in n-3 deficient mice (Fig. 5H,I). How does the treatment compensate for the loss of synapses that already occurred prior to the drug application? Presumably, the XVA-143 treatment does not make more synapses but prevents further pruning, and therefore it is strange to see the full restoration on P21. One explanation might be that phagocytic synaptic pruning occurs primarily during that 4-day period. In such case, don't we expect to observe increased PSD-95 also with the n-3 sufficient group after XVA-143 treatment?
4. Is the AA-induced increases in phagocytosis (Supplemental Figure 7) mediated through 12-/15-LOX activation?
5. If global increase in complement signaling such as C3/CR3 interaction accounts for the exacerbation of microglia-mediated synaptic refinement in n-3 deficient mice primarily through alteration of the developmental expression of complement cascade genes/proteins, what might be the reason that microglial phagocytic activity was not different between n-3 sufficient and deficient synaptosomes (Fig. 3C)? Low n-3 PUFA intake was shown to alter the C3 protein in the synapse (Fig. 5G).
6. What could be the mechanism for the reduced microglial phagocytosis after DHA treatment (Fig. 7B)?
7. According to recent findings, it is becoming clear that 12/15-LOX and its metabolites have both pro- and anti-inflammatory effects. What could be the mechanism by which 12-HETE regulates microglial phagocytosis? Is the 12-HETE receptor or further metabolism involved? What is the relationship between 12-HETE signaling and gene expression and/or complement signaling? Is 12-HETE signaling upstream of gene alteration caused by n-3 fatty acid deficiency? Is 12-HETE or a 12-HETE metabolite necessary for the phagocytic activity of microglia? Inhibiting 12/15-LOX using baicalein decreased phagocytosis even for n-3 sufficient microglia (Fig. 7F). Was 12-HETE or its metabolite production inhibited in these samples? What is the basal level of 12-HETE and its metabolites? Is it possible that baicalein has off-target effects? ALOX15 KO and transgenics as well as GPR31/12-HETER KO can help address some of these questions.
8. Considering that phagocytic activity of n-3 sufficient microglia was significantly affected by baicalein, what is the optimum phagocytic synaptic pruning for healthy neurodevelopment? Does baicalein affect in vivo expression of genes including PSD-95 and behavioral outcome? If increased

12-/15-LOX signaling mediates the impaired microglial synapse refinement caused by n-3 deficiency, what is the proper level of 12-/15-LOX metabolites in the brain for optimal neurodevelopment? In such case, how much n-3 represents deficiency or sufficiency for optimal microglial activation?

9. Total AA level did not change but free AA increased in n-3 deficient microglia. Does the observed free AA increase represent in vivo situation? How reliable is the measurement of free AA after all those processes to prepare microglia and synaptosomes? It is well-recognized that postmortem free AA level changes unless animals are killed in a manner that enzyme activities are instantaneously stopped. Is phospholipase expression/activity different in the n-3 deficient microglia?

10. The principal 12/15-LOX metabolites of AA in the brain are 12(S)-HETE and 15(S)-HETE. It appears that 15-HETE also is upregulated (Fig. 6B). Did 15-HETE produce any effects?

11. Line 87-88. Which data reveals that n-3 PUFA deficiency impairs neuronal circuit formation?

Reviewer #2:

Remarks to the Author:

C. Madore et al. describe in the article "Essential omega-3 fatty acids tune microglial phagocytosis of synaptic elements in the developing brain" a molecular mechanism for detrimental effects of low maternal n-3 PUFA intake on hippocampal development. C. Madore et al. show in their results part that that maternal dietary n-3 PUFA deficiency mice show increased microglial phagocytosis of synaptic elements in the developing hippocampus, through the activation of 12/15- lipoxygenase (LOX)/12-HETE signalling. They additionally state that the neuronal morphology and cognition is affected in the postnatal offspring. The authors discuss their results under consideration of changed lipid composition in microglia and synaptosomes, immunostaining, EM and their findings in proteins of the complement system. However, a detailed discussion of the own results with the literature is missing. Instead the authors extensively discuss their results in a bigger picture and a possible general context. Also, a detailed discussion of the own findings with reported findings in other animal models is too short and must be extended. The discussion of the biosynthesis of C22:5n6 is missing and should be attached to manuscript. Generally, data about the parent animals are missing in the manuscript. A compare of results found in parents and infants of the same group would be highly valuable. Two major aspects are missing in the manuscript: 1) The results found in the offspring mice are not compared with the parent generation. To show an impact of the n-3-PUFA deficient mice it is important to compare the fatty acid pattern in microglia and synaptosome. Here an impact of n-3-PUFA deficient could be proved over the first generation. 2) Regrettably the alteration found in the offspring of n-3-PUFA deficient are not compared with control animals. Since both n-3 PUFA sufficient and deficient mice were caged with different diets, an effect of the diet independent from the parent generation can not be shown. Generally, the data set is really weak and I recommend to increase the number of experiments. A more detailed description of the performed experiments and animal caging is necessary. Unfortunately, I have to recommend the rejection of this manuscript but encourage the authors to resubmit their work with a more sound data set.

Major:

Figure 1B) The authors show that Dendritic spine density of Golgi stained CA1 pyramidal neurons was significantly decreased in n-3 deficient mice compared to n-3 sufficient mice. (line 102-104) Please explain the difference of the number of data points between the N-3 sufficient and the n3-deficient in total numbers of junction. Here is a massive difference from 15 vs. 5. This fact might lead to misleading results and wrong significance. Also the graphic for the total dendritic length is confusing. Please increase the number of data points for the n-3 deficient.

Figure 2B /line 119-121. The author report the result: "At P21, i.e. at the peak of hippocampal synaptic pruning, we found significantly more contacts between Iba1-positive microglial processes and the synaptic cleft and more dendritic spine inclusions within those processes in n-3 deficient mice." In this graphic only three data points with a wide variation are reported. Because of the wide distribution, single data points determine the average. Without the outlier no significant

results are shown. Please increase the number of data points and perform more experiments for a statistically significant result.

Figure 2 F. line 130-132 The author report the result: "The results revealed that the percentage of phagocytic microglia is significantly higher in n-3 deficient mice across the time points studied (Figure 2E-F)." The significant results are obtained from a compare 4 vs. 3, whereby the n3-deficient animals are (n=4). Single low value in the sufficient group and additional, fourth high value in the deficient group determine the average. Without outlier probably now significant difference. Please exclude result or perform more experiments for significant statement.

Figure 2 G) (line 151-156) The author report the result: "Using a Nanostring based mRNA chip containing 550 microglia-enriched genes, 154 of these genes were found to be significantly down-regulated and 41 genes up-regulated in microglia from n-3 deficient mice whole brains (Figure 2G-J, Table 3). Among the 40 genes reported to be unique to 155 homeostatic microglia by, 20 were significantly dysregulated, indicating that maternal n-3 PUFA deficiency alters homeostatic functions of microglia (Figure 2G)." Please extend data set to a compare six vs. six animals.

Figure 3 A and B (Line 176-181) The author report the result: "We observed that maternal n-3 PUFA deficiency significantly altered the lipid profile of both microglia and synaptosomes, with an increase in n-6 fatty acids and a decrease in n-3 fatty acids (Figure 3A-B). Moreover, DHA was decreased in microglia from n-3 deficient mice (Figure 3A), while C22:5n-6 (or DPA n-6), the n-6-derived structural equivalent of DHA, was increased as a marker of n-3 PUFA deficiency (Figure 3A)." Please discuss the possibility of desaturation of C22:6n- to C22:5n6. Please compare your results with literature. Figure 3 A and B the bar graphs are hard to compare. Please modify order of bar graphs for easier overview into the data set. Add the data for ALA (C18:3n-3) into the Synaptosome part. Please discuss the possible biosynthesis of DPA in the microglia from ALA and DHA. The differences in Fatty acid profile in microglia and Synapstosome are only small. Discuss the differences of fatty acid pattern by the biosynthesis pathway of DPA and in compare to deficient mice. Do the parent generation of these animals show the same changes in Fatty acid pattern?

Figure 4 H) (line 215-218) The authors report: "However, when the same approach was used in n-3 deficient vs n-3 sufficient mice, blocking MFG-E8 action on microglia did not prevent the reduction in PSD-95 expression in the hippocampus of n-3 deficient mice (Figure 4H)." Results are not significant in the graphic. Statement can not be made up on this data set. Please make calculation again without the highest and lowest points. These determine the average and the significances.

Figure 5 A-D Line 228-229) The authors report: "Protein expression levels of CD11b (a subunit of CR3) and C1q were significantly increased in the hippocampus CA1 and DG in n-3 deficient mice (Figure 5A-B, D-E)." No Numbers of counted CD11B C1q and C3aR are given. Please show these data to claim your statement. Differences are caused by extreme outliers (Figure D) Please calculate without these extreme points. Don't refer to non significant values. Miss leading outcome. Why is the layout changed to bar graphs without individual points. Please uniform the data presentation to visible individual points. Figure 5G) Please erase p value from graphic since no significant differences are shown.

The authors should include more analytical controls in order to demonstrate that they do not analyses some separation fatty acyl artefacts or contaminations derived from buffers or other sources. Authors also should state which precaution they took to prevent autoxidation.

The authors only analysed fatty acyls therefore it is not correct to talk about lipidomics experiment or the lipidome, please remove all phrases or include lipidomics experiments including the analysis of glycerol and glycerophospholipid to make such statements.

Minor

Line 109 Please give a reference for the Y-maze task. Or describe it in the material and method section.

Figure 1 D) Please show the fold changes also the Bar graphs for the n3 sufficient animals with individual data points and variance.

Figure 2 I) Network graphic is too small. Key interactions of proteins are not visible. Please change

size. Or leave it out of the manuscript.

Figure 2 J) Because of the different case number not valuable. Please identify the individual animals with number to see the effect of each individual.

Figure 6 and C) Please divide axis for the concentrations of low abundant components. Please change to nmol/mg or other appropriate molar concentration.

Figure 6) generally concentrations are missing in the graph and in the figure legends. Please correct this error.

Figure 7C +D) It would be nice to have the base line control to Time point 0.

Reviewer #3:

Remarks to the Author:

In this article by Madore et al., the authors investigate the impact of omega-3 fatty acids (n-3 PUFAs) deficiency during pregnancy and lactation onto the phagocytic activity of microglia in the hippocampus of the offspring and related neural circuit deficits. The authors first show that deficient diet triggers a reduction in the density of spines in CA1 pyramidal cells, memory deficits as well as an increase in microglial synaptic phagocytosis and an alteration in their transcriptomic signature. To investigate the underlying mechanism, the authors further show that diet deficiency, while modifying the lipid composition of both microglia and neuronal synaptosomes, increases the capacity of microglia to engulf synaptosomes from either controls or diet-deficient neurons. The synaptic reduction was not modified by altering phosphatidyl-serine pathway but was rescued by altering the classical complement cascade. Remarkably, blocking of the complement cascade was sufficient to rescue the memory deficit triggered by a deficient diet. Finally, the authors showed that oxylipins, bioactive mediators which are rapidly converted from PUFAs, are modified in diet-deficient microglia. Specifically, 12-HETE, enhanced microglial phagocytosis and pharmacological shunting of its production was sufficient to reduce hippocampal microglial phagocytosis.

Overall, this is an impressively large experimental study that uses a variety of techniques to dig into the relationship between diet modifications, the impact on microglia and related neuronal circuits. The results are convincing, complete, and provide an important link between nutrition and brain wiring, microglial involvement, from lipidomics to behavior. These findings should be of interest to the broad readership of Nature communications.

Still, there are a few issues that need to be addressed before the paper is suitable for publication.

Comments:

-The extent of the experimental results presented makes it difficult to link the different conclusions. In particular, while the authors nicely show that diet-deficient induced microglia have an enhanced phagocytic capacity that relies on the complement cascade, it is unclear how this relates to the modifications in oxylipin profiles and whether the two pathways are parallel/convergent/overlapping. Is C3Ra expression induced by 12-HETE? Or do C3R and 12-HETE pathways act in parallel onto the phagocytic capacity of microglia? Similarly, is C3R found in the Nanostring experiment presented in Figure 2 or only detected by immunostaining? It appears in Table 3, but is not represented in Figure 2G. Clarifying these issues would be helpful for the reader; convey the conclusion and strengthen the proposed framework.

-On a similar note, it is difficult to link the data on PS with the rest of the manuscript (Figure 4 and Figure Supplemental 5). The two figures lead to slightly different conclusions that are not fully explained. The authors should either provide a clearer conclusion for these experiments.

Alternatively, they could be moved to supplemental or removed from the paper.

-Regarding the potential link with human diseases, the authors make a strong correlation with neuropsychiatric diseases such as ASD and Schizophrenia, but mostly focus on hippocampus/memory. It would be important to selectively detail what has been linked to cognitive/memory impairment in relationship to n3-deficiency in humans. As well as to detail the potential links with malnutrition (which types of diets are comparable to the one used in the

experimental paradigm?).

-Some of the images are not as convincing as they could be. In particular, Figure 5 Panels A-C underexposed and it is difficult to see how the staining in CD11b, C1q and C3R. Similarly, in Figure 2 Panel C, it is very difficult to see the red PSD95 puncta inside the microglia. The authors should provide higher magnification, different contrasts or multiple panels for these immunostainings to strengthen their claims.

-Because of the wide arrays of approaches used throughout the study, the authors need to make an extra mile to clarify all the techniques and their presentation in the figures. Please find below some specific comments:

° Figure 1B- "spine density (relative fold change)" is unclear as how it is measured? fold change compared to what?

° Figure 2 Panel A: the labeling on the EM pictures are way to small and almost not possible to read

° Figure 2 Panel E: the first description of the pHrodo experiment (text line 130) does not explain the experimental approach, but it is presented for Figure 7 (text line 279). The authors should present the experimental approach linearly to avoid confusing the reader.

° Figure 3 B states that synaptosomes were dissected from the hippocampus to do lipidomics, but there is no clear method section for lipidomics on synaptosomes.

° Were Y maze experiments performed at P21 or in adults? These behavioral tests are usually performed in adults, experiments were all performed in P21. The authors could specify this point.

Rebuttal letter

Opening statement

We would like to thank the reviewers for their insightful comments. We were able to address the majority of them, either by adding new datasets/editing original figures and/or by elaborating on several points in the Discussion section. Specifically, we further deciphered the relationship between the 12/15-LOX/12-HETE pathway and the complement system and showed that 12/15-LOX is the master controller of microglial phagocytosis under dietary n-3 PUFA deficiency, acting upstream of the complement cascade. We initially prioritized the *in vivo* and *ex vivo* experiments to understand the effects of n-3 PUFA deficiency on microglial function. Due to the shutdown of the lab following the Covid-19 pandemics, we are unable to perform all of the experiments regarding the points raised by the reviewers on the *in vitro* aspects. The French government recently declared that the French Universities were not reopening before this summer at best, which means that we would not have enough material (animals and cultures) to run new experiments not before September, in the best-case scenario.

Altogether, with the extensive modifications to the manuscript we think that the present revised version is ready for reviewing.

All modifications appear **in bold** in the manuscript and are listed in the rebuttal letter below (*in italic*). Figures noted "Rebuttal Figure X" do not appear in the final version of the manuscript. Figures noted "Figure X" or "Supplementary Figure X" have been added to the manuscript.

Reviewers comments

Reviewer #1 (Remarks to the Author):

This manuscript presents a potential molecular mechanism for detrimental effects of low maternal omega-3 fatty acid intake on hippocampal development. The authors report that low maternal dietary intake of omega-3 fatty acids increases microglial phagocytosis of synaptic elements in developing hippocampus through the activation of 12/15-LOX/12-HETE signaling, affecting neuronal morphology and cognitive function in offspring. While it may be a novel mechanism for potential n-3 deficiency-related suboptimal neurodevelopment, there are many questions to be addressed.

We would like to thank the reviewer for their time and attention to detail as well as noting the novelty of our work.

1. Figure 4H. Why did PSD-95 expression not increase with cRGD compared to sc-cRGD for the n-3 sufficient group? Given that 'pharmacological inhibition of the interaction between MFG-E8 and its microglial vitronectin receptor using a blocking peptide (cRGD) increased PSD-95 in standard diet-fed animals (Supplemental Fig. 5)', similar effects would be anticipated, even if 'the modulation of the microglial phagocytic capacity by early-life PUFA is PS recognition-independent.'

Is there large difference in the brain fatty acid profile and gene/protein expression between n-3 sufficient and standard-diet-fed animals? The fatty acid profile and gene expression data from all three diet groups should also be clearly presented, particularly because the biological difference is supposedly driven by the lipid composition.

In the experiment originally presented in supplementary Figure 5, our goal was to assess the efficiency of cRGD in a condition in which PS recognition is known to be involved in synaptic pruning in the developing hippocampus (1–4). At the same time, we were bringing additional data to the field on the potential role of PS recognition in synaptic pruning in the hippocampus under standard chow diet.

From this positive control, we can conclude that the absence of effect of cRGD on PSD95 expression in the n-3 deficient group is not due to a lack of efficiency of the peptide but rather to the fact that PS recognition is not involved in synaptic refinement in these conditions. This suggests that when maternal n-3 PUFA intake is low, microglia switch to a PS-independent/complement-dependent recognition mechanism in the offspring.

As mentioned by the reviewer, this dataset also highlights intrinsic differences between standard and n-3 sufficient mice and questions the underlying mechanisms. In n-3 sufficient mice, neither PS recognition nor the complement cascade seem to be involved in synaptic refinement. This suggests that 1) other mechanisms of microglia-mediated synaptic refinement are in place in the hippocampus of these animals. Indeed, microglia uses a plethora of recognition mechanisms for phagocytosis (5,6)

and few studies have addressed the microglia-spines interactions in the developing hippocampus (in standard-fed animals); 2) that microglia-mediated synaptic refinement is not prominent in the hippocampus of these animals.

Synaptic pruning occurs via various mechanisms, including microglia-independent processes, though no studies have ever quantified the proportion of microglial-mediated synaptic pruning in the developing hippocampus. Among other mechanisms known to mediate synaptic running, neuronal macroautophagy may play a role, as it does in the cortex (7,8), or astrocyte-mediated mechanisms as shown in the developing brain (9). Hence, the proportion microglia-independent synaptic pruning mechanisms in the developing hippocampus of n-3 sufficient mice may well be higher than anticipated.

We added some **discussion** about these aspects in the Discussion section. It appears as follows:

Of note, neither PS recognition nor the complement cascade seem to be involved in synaptic refinement in n-3 sufficient mice. This suggests that 1) other mechanisms of microglia-mediated synaptic refinement are in place in the hippocampus of these animals or 2) microglia-mediated synaptic refinement is not prominent in the hippocampus of these animals. Microglia use a plethora of recognition mechanisms for phagocytosis of neuronal elements/dead bodies (5,6) but few studies have addressed the microglia-spine interaction in the developing hippocampus. On the other hand, several mechanisms, including microglia-independent processes such as neuronal macroautophagy or phagocytosis by astrocytes, control synaptic pruning in the developing brain, though the proportion of synaptic pruning relying on microglia in the developing hippocampus is not known (7–9). Our data suggest that synaptic pruning mechanisms in the developing hippocampus of n-3 sufficient mice may favor microglia-independent processes.

In any case, these observations reinforce the necessity to use n-3 sufficient mice as controls in the present study, and not standard-fed animals as they display very distinct molecular mechanisms. What can be the reason for such discrepancies?

We previously published the fatty acid composition of the cortex of adult mice fed either a standard chow, n-3 sufficient or n-3 deficient diet (10). Cortices from standard-fed and n-3 sufficient-fed animals are quite similar in terms of fatty acid composition (see **Rebuttal Figure 1** below). Hence, differential lipid composition at the structure level is unlikely to explain the difference between both groups.

Rebuttal Figure 1: Cortex fatty acid composition of mice fed a standard chow or a n-3 sufficient diet. Data are expressed as % of total fatty acid (Extracted from 74).

However, when looking at the fatty acid composition of the 2 diets (see data below, extracted from Table 2 of the manuscript), one can see differences between the standard chow and n-3 sufficient diet (e.g. total saturated fat, monounsaturated, PUFAs, LA/ALA ratio). The content of proteins, carbohydrates, and lipids varies as well (3.1% of lipids in the standard chow vs 5% in the n-3 sufficient diet).

Table 2: Fatty acid composition of the diets (% wt of total fatty acids)

diets	Sufficient	Standard chow (A04)
16:0	22.6	20.3
18:0	3.3	2.2
other saturated FAs	1.8	1.8
total saturated FAs	27.7	24.3

18:1n-9	57.9	19.3
18:1n-7	1.5	1.5
other monounsaturated FAs	0.4	3.9
total monounsaturated FAs	60.0	24.7
18:2n-6 (LA)	10.6	45.9
Other n-6 PUFAs	n.d.	0.3
total n-6 PUFAs	10.7	46.2
18:3n-3 (ALA)	1.6	3.3
20:5 n-3	n.d.	0.6
22:5 n-3	n.d.	0.1
22:6 n-3	n.d.	0.8
total n-3 PUFAs	1.6	4.8
total PUFAs	12.3	51.0
LA/ALA	6.6	13.9

FAs, fatty acids; PUFAs, polyunsaturated fatty acids; LA: linoleic acid; ALA, α -linolenic acid.

Hence, other aspects than brain fatty acid composition are likely to explain differences between both groups, such as differences in cell energy metabolism, differences in lipid composition at the cellular resolution or difference in the production of lipid derivatives.

That being said, because the question of the role of PS-dependent recognition mechanisms in synaptic pruning in the developing brain is not our focus, and as suggested by reviewer 3, we moved Figure 4 to supplementary material (Supplementary Figure 7) and removed the dataset presented in Supplementary Figure 5 from the manuscript. We think that elaborating on this aspect does not add relevant information on how n-3 PUFA deficiency increases microglial phagocytosis of spines.

2. The DHA content seems very low in synaptosomes. What is the total fatty acid composition in the cortex or hippocampus? Supplemental Figure 4 should be expressed as weight percentage rather than the ratio.

Supplementary Figure 4 has been **edited** accordingly and data are now expressed as weight percentage.

In our hands, DHA levels in synaptosomes are between 0.5% (n-3 deficient) and 2.5% (n-3 sufficient) of total fatty acids (Figure 3B). There are limited data available in the literature on synaptosome fatty acid composition. In 1984, Dr JM Bourre reported on the fatty acid composition of synaptosomes extracted from P15 Wistar rats fed with a Soya- or sunflower-enriched diet and found DHA levels to be higher than ours (8.5% and 1.9% of total fatty acid respectively) (11).

In the whole cortex of P21 CD1 mice, we found DHA levels ranging between 5.5 \pm 0.1% (n-3 deficient) and 13.3 \pm 0.2% (n-3 sufficient) of total fatty acids (**p<0.0001; Supplementary Figure 4). The results are quite heterogeneous in the literature; yet, our data are comparable to several studies including:

- Cao 2009, Journal of Neurochemistry: In the hippocampus of P18 C57Bl/6 mice, DHA levels were of 12.39 \pm 0.99 in the "n-3 adequate" group and of 3.33 \pm 1.50 in the n-3 deficient mice (**p<0.0001) (12).
- Hamilton L et al., 2000, Lipids: In the cortex of P21 Long Evans rats, DHA levels were of 15.3 \pm 0.6% and 2.5 \pm 0.1 in n-3 adequate and n-3 deficient animals respectively (13).
- Vandal M et al., 2014, Journal of Neurochemistry. In the cortex of 4 month-old C57Bl/6 mice, DHA levels were of about 17% under standard chow (14).

Overall, the cortical DHA levels reported in our study are in the classical range when comparing to other studies, especially those examining very young animals. Hence, the low levels of DHA measured in synaptosomes cannot be explained by low levels of DHA in the whole structure. Discrepancies may come from species/strain difference (rats vs mice, CD1 vs C57Bl/6 mice), the fatty acid composition of the diet that varies significantly from one study to another (e.g. % of DHA in the n-3 sufficient – or 'adequate'- diet of Cao 2009=0.9%, while DHA is not detectable in our n-3 sufficient diet), the age of animals (the brain of the animals is growing rapidly at that period – 'brain growth spurt'), the duration of exposure to the diet (from birth vs from first day of gestation), etc.

3. The authors showed that antagonizing CR3 by administering XVA-143 directly in the hippocampus four days before assessing protein expression at P21 prevented PSD-95 expression decrease in the hippocampus of n-3 deficient mice and restored optimal memory ability in n-3 deficient mice (Fig. 5H,I). How does the treatment compensate for the loss of synapses that already occurred prior to the drug application? Presumably, the XVA-143 treatment does not make more synapses but prevents further pruning, and therefore it is strange to see the full restoration on P21. One explanation might be that phagocytic synaptic pruning occurs primarily during that 4-day period. In such case, don't we expect to observe increased PSD-95 also with the n-3 sufficient group after XVA-143 treatment?

Microglia-mediated synaptic pruning is an exponential process that peaks at P21 in the hippocampus of mice (15). Hence, and as anticipated by the reviewer, it is very likely that synaptic pruning is greater in the 3-4 days preceding P21, which might explain the strong effect of XVA-143 on the restoration of PSD-95 levels. It may be that the effects of truly aberrant complement-mediated pruning are not detectable until the majority of pruning is occurring. The fact that we do not see a protective effect with XVA-143 in n-3 sufficient adds to the evidence that the complement system is not involved in control mice fed with proper quantities of n-3 PUFAs in the hippocampus. Our observations confirm recent publications showing that in mice fed a standard diet, complement cascade is not involved in synaptic pruning in the hippocampus (1–4). It may also be the case, as alluded to above, that pruning mechanisms that are well described in the developing visual system may not be exactly the same in the hippocampus. There are papers showing regional heterogeneity of glia that could contribute to this (16).

Here is what was specified in the original **Discussion** section (to which we added the references (1,2,4) that were missing):

“We further identified the complement system, and not PS exposure, as the molecular mechanism driving spine phagocytic activity upon n-3 PUFA deficiency. This is consistent with previous work showing that in the hippocampus, PS-dependent and complement-independent trogocytosis of presynaptic elements remodels neuronal network during normal brain development (1–4) while complement-dependent phagocytosis of synapses is promoted in models of brain diseases presenting neuronal network abnormalities, including neurodevelopmental disorders (17–22)”.

We also added discussion on the potential mechanisms in place in n-3 sufficient mice (see above).

4. Is the AA-induced increases in phagocytosis (Supplemental Figure 7) mediated through 12-/15-LOX activation?

While there are no data available in the literature on microglia, this pathway has already been described as essential in the clearance of apoptotic cells by peripheral macrophages during inflammation (23). Therefore, we predict that the 12/15-LOX enzyme is involved in AA-mediated increase in microglia phagocytosis *in vitro*. However, as mentioned in the opening statement, we focused all supplementary experiments on the comprehension of *in vivo* mechanisms. Hence, we did not have the chance to decipher further the implication of the 12/15-LOX in cultured primary microglia exposed to AA.

Here is what we added in the **Discussion** section:

“While there is no data available in the literature on microglia, this pathway has already been described as essential in the clearance of apoptotic cells by peripheral macrophages during inflammation (23). Therefore, we predict that that the 12/15-LOX enzyme is involved in AA-mediated increase in microglia phagocytosis in vitro”.

5. If global increase in complement signaling such as C3/CR3 interaction accounts for the exacerbation of microglia-mediated synaptic refinement in n-3 deficient mice primarily through alteration of the developmental expression of complement cascade genes/proteins, what might be the reason that microglial phagocytic activity was not different between n-3 sufficient and deficient synaptosomes (Fig. 3C)? Low n-3 PUFA intake was shown to alter the C3 protein in the synapse (Fig. 5G).

The review raises another very good point that was not emphasized enough in the original manuscript. According to our data, the activation of microglial cells by n-3 PUFA deficiency and overexpression of CD11b (CR3) is a stronger driving force than C3-mediated opsonization of synaptosomes for phagocytosis. We here confirm several elegant studies demonstrating that without proper activation of the phagocyte, the mere opsonization of a cell/cellular element is not sufficient to induce phagocytosis. This is confirmed by the experiments presented in Figure 3C. In fact, conformational changes of CR3 and redistribution of the receptors in the membrane of phagocytes are also required to allow phagocytosis of the opsonized element. Below is the **discussion** we added on this topic:

“We further investigated the relationship between the 12/15-LOX/12-HETE pathway and the complement system. Our data reveal that the phagocytic capacity of microglia is increased only when

the cells themselves are n-3 deficient, even though C3 is overexpressed on n-3 deficient synaptosomes. These data suggest that both opsonization of synapses and recognition of C3 by its microglial receptors are required to trigger phagocytosis. This confirms multiple reports of the complement-dependent phagocytosis in the immune system. It is already well described that there is little to no ingestion of opsonized elements without phagocyte activation (23,24). Conversely, CR3-mediated phagocytosis is dependent on receptor conformational changes, lateral diffusion and clustering, processes that require activating stimuli (24–29). Hence, binding of C3 to the complement receptors can be considered as the recognition (“find-me”) mechanism of spines to be removed by microglia, as previously shown in other experimental contexts (19,30–32), while activation of the 12/15-LOX/12-HETE pathway is the triggering signal for phagocytosis by microglia. This is coherent with the observation that the complement cascade classically interacts with inflammatory pathways to control the activity of phagocytes (33).

6. What could be the mechanism for the reduced microglial phagocytosis after DHA treatment (Fig. 7B)?

To address this question, we measured the expression level of CD11b mRNA in primary microglia treated with AA or DHA (vs ethanol, control condition). Results appear in Figure 2 below:

Rebuttal Figure 2: CD11b mRNA expression in primary microglia treated with AA or DHA for 24h.

Our data reveal that DHA does not reduce the transcription level of CD11b below control. This suggests that inhibition of the complement mRNA expression does not explain the decrease in phagocytic activity in microglia treated with DHA.

As an alternative explanation, we previously showed that DHA exerts inhibitory action on the TLR4/NFκB pathway, TLR receptors being involved in microglial phagocytosis (34–37). This was confirmed in an *in vivo* model of traumatic brain injury in which DHA exerted neuroprotective effects via repressing this pathway (38). Additionally, recent work published by the group of Michael-Titus found that DHA decreases microglia phagocytic activity of myelin debris in the spinal cord after injury through mir-124 (39), a non-coding nucleotide involved in synaptic plasticity (40). In macrophages, mir-124 has been shown to regulate the actin-related protein 2/3 (ARP2/3) which is involved in the formation of the phagocytic cup (41). This is in line with results showing that DHA reduced LPS-induced phagocytic activity of microglia, through a reduction of filopodia (42).

We added some **discussion** about this topic. It appears as follows in the Discussion section:

Conversely, DHA (but not its LOX derivative, 17S-HDOHE) decreased microglial phagocytosis. It is unlikely to rely on the inhibition of the complement cascade as we found no effect of DHA on CD11b expression in vitro (data not shown). This is coherent with previous works, including ours, showing that DHA exerts inhibitory action on the TLR4/NFκB pathway, TLR receptors being capable of activating microglial phagocytosis (34–37). Additionally, recent work found that DHA decreases microglia phagocytic activity of myelin debris in the spinal cord after injury through mir-124 (39), a non-coding nucleotide involved in synaptic plasticity (40). In macrophages, mir-124 has been shown to regulate actin-related protein 2/3 (ARP2/3) which is involved in the formation of the phagocytic cup (41). This is in line with results showing that DHA reduced LPS-induced phagocytic activity of microglia, through a reduction of filopodia (42).

7. According to recent findings, it is becoming clear that 12/15-LOX and its metabolites have both pro- and anti-inflammatory effects. What could be the mechanism by which 12-HETE regulates microglial phagocytosis?

As pointed out by the reviewer, increasing evidence highlights the complex role of 12/15-LOX activation in inflammation, as its metabolites display both pro- and anti-inflammatory properties. This can be attributed to the complex array of lipid derivatives that are formed as a result of its catalytic activity of several precursors (43).

As for the mechanisms by which the metabolite 12-HETE can promote microglial phagocytosis, the literature is sparse, not always consistent and mainly based on *in vitro* studies. What is known is that 1) Activation of the 12/15 LOX is necessary for peritoneal macrophages to phagocytose apoptotic cells (23); 2) The 12/15 LOX pathway can activate the transcription factor family PPAR; 3) these transcription factors are inducers of phagocytosis (44); 4) Some members of the PPAR family enhance the expression of opsonins, such as the complement component-1 (C1qa and C1qb), thrombospondin-1 and Mfge8, all involved in bridging apoptotic cells to the macrophage surface receptor, thus increasing apoptotic cell clearance, both in human and mouse macrophages (44).

Altogether, this suggested that the downstream activation of PPAR by the 12/15-LOX/12-HETE pathway could increase microglial phagocytic activity in n-3 PUFA deficient mice. We thus quantified the expression level of PPAR γ mRNA in AA-treated microglia exposed to synaptosomes but could not find any increase of its expression (Figure 3 below).

Rebuttal Figure 3: Gene expression of the transcription factor PPAR γ in primary microglia exposed to synaptosomes for 2h and pre-incubated with AA or its vehicle (ethanol) for 24h.

The phagocytic activity of AA-treated primary microglia is not correlated to the increase of PPAR γ expression, suggesting that other mechanisms might be in place in this context.

Overall, the literature is too sparse and the statement that PPAR might be the molecular mechanism in place is too speculative and beyond the scope of our article. We chose to concentrate on and decipher further the link between the 12/15-LOX/12-HETE pathway and the complement cascade, while we are fully aware that other mechanisms might be involved.

Is the 12-HETE receptor or further metabolism involved? What is the relationship between 12-HETE signaling and gene expression and/or complement signaling? Is 12-HETE signaling upstream of gene alteration caused by n-3 fatty acid deficiency? Is 12-HETE or a 12-HETE metabolite necessary for the phagocytic activity of microglia?

12-HETE can regulate cell activity in an autocrine manner by binding to its receptors or by modulating intracellular processes in a receptor-independent manner (45,46).

Are the 12-HETE receptors involved?

We thus generated a new dataset to address this question. We quantified the mRNA expression of the 12-HETE receptors GPR31 and BLT2 in primary microglia treated with 12-HETE or its vehicle (Supplementary Figure 9A, below). Data are expressed as percentage of variation from the controls. However, when looking at the actual CT values, while GPR31 was expressed by the cells (27.41 +/- 0.3 CT), BLT2 was below detection limits in culture (33.15 +/- 0.8 CT). Hence, we then focused on the expression of GPR31 *ex vivo*, on freshly isolated n-3 deficient and n-3 sufficient microglia, treated with vehicle or baicalein, in absence or presence of synaptosomes. We could not find any significant modification of GPR31 mRNA expression level in any of the conditions tested (**Supplementary Figure 9 of the revised version; see below**).

Supplementary Figure 9: The 12/15-LOX/12-HETE signaling pathway-induced microglial phagocytosis is independent from a modulation of the 12-HETE receptors expression both *in vitro* and *ex vivo*. **A.** mRNA expression of the two principal 12-HETE receptors BLT2 and GPR31 in primary microglia culture. Means \pm SEM; n=3-5 experiments per condition. Two-tailed unpaired Student's t-test, *blt2*: vehicle vs 12-HETE, $t=0.56$, $p=0.596$, *gpr31*: vehicle vs 12-HETE $t=0.52$, $p=0.62$. **B.** Quantification of *gpr31* mRNA expression in freshly sorted n-3 deficient and n-3 sufficient microglia, treated with baicalein or its vehicle. Means \pm SEM; n=4-6 per group. Two-way ANOVA: diet effect, $F(1,15)=1.02$, $p=0.33$; treatment effect, $F(1,15)=0.51$, $p=0.49$; interaction, $F(1,15)=0.19$, $p=0.67$. **C.** Quantification of *gpr31* mRNA expression in freshly sorted n-3 deficient and n-3 sufficient microglia, exposed to synaptosomes and treated with baicalein or its vehicle. Means \pm SEM; n=6-8 per group. Two-way ANOVA: diet effect, $F(1,25)=1.27$, $p=0.279$; treatment effect, $F(1,25)=0.63$, $**p=0.0028$; interaction, $F(1,25)=1.95$, $p=0.17$.

Are receptor-independent mechanisms involved?

12-HETE can also be further metabolized into bioactive products such as lipoxins, hepxillins, eoxins, 8(S)15SDiHETE, 5(S)DiHETE and oxoETE, yet we could not measure these metabolites in n-3 deficient microglia for technical reasons (43,46). Hence, we cannot rule out that downstream metabolites of 12-HETE are involved in this process.

12-HETE can activate signaling pathway such as PPAR. We already discussed the possible role of PPAR in the modulation of n-3 deficient microglia phagocytosis. However, we could not find any modification in the expression of PPAR mRNA in these cells. Moreover, we could not find any obvious link between the PPAR pathway and the complement cascade.

What is the link between the 12/15-LOX/12-HETE pathway and the complement cascade?

Hence, to decipher further the link between the 12/15-LOX/12-HETE pathway and the complement cascade, we exposed freshly sorted n-3 sufficient or n-3 deficient microglia to the 12/15-LOX inhibitor baicalein or its vehicle and quantified the expression level of CD11b and C3aR mRNAs, 2h later (**Figure 7A of the revised version; see below**). We show that *c3ar* gene expression was increased in n-3 deficient microglia while the expression of *cd11b* was not significantly modulated by the diet, in line with our immunostaining experiments in which CD11b was moderately increased under n-3 PUFA deficiency, only in the DG, while C3aR was robustly increased in the whole hippocampus (Figure 4B of the revised version). Baicalein significantly reduced C3aR expression in n-3 deficient microglia, while it had no effect on CD11b (**Figure 7B-C**). These results suggest that in non-stimulated n-3 deficient microglia, the 12/15-LOX/12-HETE signaling pathway is implicated in the maintenance of a high C3aR tone, while the activity of CD11b (as shown in Figure 4E-F of the revised version) is likely to depend on other mechanisms.

Once activated, the expression of most immune receptors is dampened as part of a global homeostatic mechanism (47). We thus examined the relationship between the 12/15-LOX/12-HETE signaling pathway and the complement cascade in condition of stimulation with synaptosomes. We exposed freshly sorted cells to pHrodo synaptosomes in the presence of baicalein or its vehicle and quantified the expression level of *cd11b* and *c3ar* (**Figure 7D**). The expression level of CD11b was significantly increased by baicalein in n-3 deficient microglia while inhibiting the 12/15 LOX enzyme did not affect C3aR mRNA expression (**Figure 7E-F**). This suggests that once microglia have started phagocytosing synaptosomes, the 12/15-LOX/12-HETE signaling pathway inhibits CD11b expression while allowing phagocytosis of neuronal material (Figure 6F of the revised version).

This novel dataset suggests that the 12/15-LOX/12-HETE signaling pathway is the master controller of microglial phagocytosis under n-3 PUFA deficiency, partly by regulating the expression of the C3a and C3b receptors.

Figure 7, revised version: The 12/15-LOX/12-HETE signaling pathway controls gene expression of the complement pathway.

A. Experimental setup. **B-C.** Quantification of *cd11b* and *c3ar* mRNA expression in freshly sorted n-3 deficient and n-3 sufficient microglia, treated with baicalein or its vehicle. Means \pm SEM; n=4-6 per group. Two-way ANOVA: *cd11b*: diet effect, $F(1,15)=0.57$, $p=0.46$; treatment effect, $F(1,15)=1.37$, $p=0.26$; interaction, $F(1,15)=2.48$, $p=0.14$; *c3ar*: diet effect, $F(1,18)=1.36$, $p=0.26$; treatment effect, $F(1,18)=1.52$, $p=0.23$; interaction, $F(1,18)=4.36$, $*p=0.051$. **D.** Experimental setup. **E-F.** Quantification of *cd11b* and *c3ar* mRNA expression in freshly sorted n-3 deficient and n-3 sufficient microglia, exposed to synaptosomes and treated with baicalein or its vehicle. Means \pm SEM; n=8 per group. Two-way ANOVA followed by Bonferroni *post-hoc* test: *cd11b*: diet effect, $F(1,28)=0.16$, $p=0.19$; treatment effect, $F(1,28)=0.92$, $**p=0.0028$; interaction, $F(1,28)=0.072$,

$p=0.36$; n-3 deficient+vehicle vs n-3 deficient+ baicalein, $*p=0.014$; n-3 deficient+ vehicle vs n-3 deficient+ baicalein, $*p=0.029$; *c3ar*: diet effect, $F(1,28)=0.35$, $**p=0.0034$; treatment effect, $F(1,28)=0.0043$, $p=0.72$; interaction, $F(1,28)=0.0019$, $p=0.81$.

To complete our analysis, we also performed datamining and looked for evidence of a role of the 12/15-LOX pathway in various pathophysiological context. Here is what we added in the Discussion section on this aspect:

Our data are in line with previous studies showing that the 12/15-LOX/12-HETE pathway is altered in several pathological conditions in macrophage/microglia and especially in inflammatory conditions. Human macrophages incubated with 12-HETE showed induced gene expression after LPS application (48). Indeed, 12-HETE in macrophages was shown to activate extracellular signal-regulated kinase 1/2 (ERK1/2) and nuclear factor kappa-light-chain-enhancer of activated B cells (NFkB) via GPR31 (49). In humans, it was found that the expression of 12/15-LOX was increased in CD68-positive activated microglia (50). Moreover, 12/15-LOX and Gpr31 receptor gene expression is increased in microglia along with aging (16) and in DAM/MGnD (51). However, we did not observe a change in Gpr31 gene expression after application of 12-HETE in vitro or ex vivo modulated by the diet in a phagocytic mode, suggesting that the effects of 12-HETE on microglia are likely to be receptor-independent. Downstream analysis of GPR31 activation after 12-HETE application or expression should be tested.

Inhibiting 12/15-LOX using baicalein decreased phagocytosis even for n-3 sufficient microglia (Fig. 7F). Was 12-HETE or its metabolite production inhibited in these samples? What is the basal level of 12-HETE and its metabolites? Is it possible that baicalein has off-target effects? ALOX15 KO and transgenics as well as GPR31/12-HETER KO can help address some of these questions.

While it is always hard to claim that baicalein has no off-target effects, it is considered as a rather specific inhibitor of leukocyte 12/15LOX (52). This drug is a flavonoid isolated from the root of *Scutellariae radix* that was originally published as an inhibitor of platelet-type 12-LOX (53), yet it is also highly inhibitory towards the human 15-LOX isoform, and thus has the potential to inhibit the 12/15-LOX pathway (54). Hence, while we cannot exclude the participation of other pathways in the effects of baicalein, the 12/15-LOX/12-HETE pathway is the main target of this drug. We could not measure the basal levels of 12-HETE and its metabolites in our samples. However, we were able to show that 12-HETE increases microglial phagocytic activity (Figure 6C, revised version) and the relationship between this pathway and the complement cascade (Figure 7, revised version). Finally, though a genetic approach would have been valuable, we chose to favor a pharmacological approach that may have some translational relevance as it is a relatively non-toxic drug which may have potent efficacy without compromising the health of individuals (see the clinical trial NCT03830684 as an example).

Many studies have showed that baicalein is one such potential candidate (55). Finally, the 12/15-LO^{loxP/loxP} mice are available at JAX. However, with the lock down situation, we cannot start a new colony and run experiments a new set of experiments. It would most likely require between 12 and 18 months to purchase the mice (no purchasing allowed in France yet), receive them from JAX, cross them with microglia-Cre mice, and run experiments on a fair number of animals. We think that the GPR31/12-HETER KO mice would not be relevant here as we did not find any evidence of receptor-mediated activity of the 12-HETE (see above).

8. Considering that phagocytic activity of n-3 sufficient microglia was significantly affected by baicalein, what is the optimum phagocytic synaptic pruning for healthy neurodevelopment? Does baicalein affect in vivo expression of genes including PSD-95 and behavioral outcome?

Early-life synaptic pruning has been extensively studied and was shown to be critical for proper brain development. Several publications indicate that disruption of microglia-mediated synaptic pruning leads to abnormal behavior later in life (18,56). Conversely, exacerbated synaptic pruning is also detrimental for brain function (57), as we now describe in n-3 deficient mice. The optimum pruning is found in normal post-natal brain development and evidence suggests deviation from this is detrimental. We currently do not have an objective pruning metric, though this would be interesting avenue, especially if it is maintained across species. But for now, all studies measure deviation from healthy control, therefore implying this is optimum (which we agree with).

That being said, to evaluate the behavioral outcome of baicalein in n-3 sufficient (control) mice, we ran a Y-maze experiment in which we treated animals with baicalein at a concentration of 20 mg/kg (58) at P14, P17 and 30 min prior the first trial at P21. Data are presented as the time spent exploring the novel and the familiar arms during the second trial. **See Figure below (added as Supplementary Figure 8B in the final version of the manuscript):**

Supplementary Figure 8. B. Time spent in novel vs familiar arm in the Y maze task in P21 n-3 sufficient mice treated with baicalein or its vehicle. Paired t-test: n-3 sufficient group, p=0.056; n-3 sufficient + baicalein group, *p=0.038.

Importantly, our data show that baicalein injection did not alter the memory abilities of n-3 sufficient mice in the Y-maze task (data not shown).

We ran the same experiments in n-3 deficient mice and showed that injection of baicalein persistently restored optimal memory abilities in these mice in the Y-maze task (**Figure 6G of the revised version; see below left panel**), without altering locomotor activity (no change in the total travelled distance in the Y maze, data not shown) or anxiety level (**Supplementary Figure 8; see below right panel**). These in vivo results confirm that the 12/15-LOX/12-HETE pathway is active in n-3 deficient microglia to increase their phagocytic activity towards synaptic elements, which is correlated to the

occurrence of memory deficits.

Figure 6. G. Time spent in novel vs familiar arm in the Y maze task in P21 and P30 n-3 deficient mice treated with baicalein or its vehicle. Means \pm SEM; n=10 (P21) or 5 (P30) mice per group. Paired t-test: P21: n-3 deficient group, p=0.5; n-3 deficient + baicalein group, p=0.11; P30: n-3 deficient group, p=0.57; n-3 deficient + baicalein group, *p=0.029.

Supplementary Figure 8: Inhibition of the 12/15 LOX enzyme with baicalein does not affect the behavior of mice in the open Field. Time spent in the center vs periphery of the Open-Field. Means \pm SEM; n=5 mice per group. Paired t-test: n-3 deficient group, ***p<0.0001; n-3 deficient + baicalein group, ***p<0.0001.

If increased 12/15-LOX signaling mediates the impaired microglial synapse refinement caused by n-3 deficiency, what is the proper level of 12/15-LOX metabolites in the brain for optimal neurodevelopment? In such case, how much n-3 represents deficiency or sufficiency for optimal microglial activation?

We agree with the reviewer that it would be interesting to define the exact level of 12/15-LOX activity that is required for optimal development, in relation with the cell content in n-3 PUFAs. The main thrust of our study was to elucidate the problem with n-3 deficiency and the mechanisms of how this creates problem with development. We consider our control animals to have 'optimal' development, though we recognize that there will be a defined range of 12/15-LOX metabolites.

In the context of another study, we gathered some data that do not entirely address the question raised, yet they may be of interest for the reviewer (**Rebuttal Figure 4**). We performed a time course analysis of the 12-HETE metabolite synthesis and the 12-LOX enzyme gene expression in CD1 mice fed a standard chow. We performed each measurement (12-HETE and 12-LOX quantification) in each hemibrain of the same animal. These data reveal that the production of 12-HETE and 12-LOX is age-dependent, with a peak of production for both molecules right after birth.

Rebuttal Figure 4. Time course analysis of 12-HETE production and 12-*lox* mRNA expression in the brain of CD1 mice. Means \pm SEM; n=5-7 mice per group. One-Way ANOVA; 12-HETE: F(3;16)=8.5, **p=0.0013; *Post-hoc* Tukey's multiple comparisons, P0 vs P7 **p=0.0013, P0 vs P14 *p=0.0118, P0 vs P21 **p=0.0082. 12-LOX: F(3;23)=6.85, **p=0.0018; *Post-hoc* Tukey's multiple comparisons, P0 vs P14 *p=0.0105, P0 vs P21 **p=0.0039.

To fully address the reviewer's question, this would require a tremendous amount of experiments in which we would modulate the intake of dietary n-3 PUFAs incrementally in the dams and assess microglial phagocytosis, spine density, 12/15-LOX activity and memory abilities for each increment in the offspring. We consider n-3 PUFA sufficiency as the optimal situation as it is correlated with optimal memory abilities in the offspring as previously published that n-3 sufficient mice do not display cognitive deficits (non-exhaustive list: 49,50,52,54,55).

9. Total AA level did not change but free AA increased in n-3 deficient microglia. Does the observed free AA increase represent in vivo situation? How reliable is the measurement of free AA after all those processes to prepare microglia and synaptosomes? It is well-recognized that postmortem free AA level changes unless animals are killed in a manner that enzyme activities are instantaneously stopped. Is phospholipase expression/activity different in the n-3 deficient microglia?

The reviewer raises the concern about processing and measurement of free AA and eicosanoids. While there are methods to capture these pools accurately from whole tissue in vivo, it cannot be done

at the cell resolution (at the level of the microglia in our case), therefore we are unable to say if this accurately represents in vivo levels. Nevertheless, we would like to point out that there are examples where brain phospholipid AA, unesterified AA and eicosanoids concentrations change independent of each other (64,65). Unfortunately, we do not have any data regarding the expression/activity of phospholipase in n-3 deficient vs n-3 sufficient mice.

10. The principal 12/15-LOX metabolites of AA in the brain are 12(S)-HETE and 15(S)-HETE. It appears that 15-HETE also is upregulated (Fig. 6B). Did 15-HETE produce any effects?

15-HETE is upregulated in n-3 deficient microglia but not significantly (see **Rebuttal Figure 5** below). Hence, we did not consider it as a good candidate and did not apply it on primary microglia. Moreover, while the 12/15-LOX enzyme can produce both 12-HETE and 15-HETE with arachidonic acid as substrate (43), and while baicalein can inhibit the production of both metabolites, Xu et al. previously demonstrated that only 12-HETE application was able to reverse the effects of baicalein on PPAR γ activation in a microglial cell line. 15-HETE application had no effect in this context (52). In combination with our new data, we considered 12-HETE as the best candidate.

Rebuttal Figure 5. AA-derived 15-HETE mediator is not significantly modulated by early-life n-3 PUFA deficient diet. Means \pm SEM; n=4 mice per group. Two-tailed unpaired Student's t-test; t=1.30, p=0.24

11. Line 87-88. Which data reveals that n-3 PUFA deficiency impairs neuronal circuit formation?

This sentence has been **modified** to avoid overstatement:

“Using multidisciplinary approaches, we reveal that maternal exposure to an n-3 PUFA deficient diet impairs offspring’s microglial homeostatic signature and phagocytic activity, underpinning excessive synaptic pruning and subsequent behavioral abnormalities”

Reviewer #2 (Remarks to the Author):

C. Madore et al. describe in the article “Essential omega-3 fatty acids tune microglial phagocytosis of synaptic elements in the developing brain” a molecular mechanism for detrimental effects of low maternal n-3 PUFA intake on hippocampal development. C. Madore et al. show in their results part that that maternal dietary n-3 PUFA deficiency mice show increased microglial phagocytosis of synaptic elements in the developing hippocampus, through the activation of 12/15- lipoxygenase (LOX)/12-HETE signaling. They additionally state that the neuronal morphology and cognition is affected in the postnatal offspring. The authors discuss their results under consideration of changed lipid composition in microglia and synaptosomes, immunostaining, EM and their findings in proteins of the complement system.

However, a detailed discussion of the own results with the literature is missing. Instead the authors extensively discuss their results in a bigger picture and a possible general context. Also, a detailed discussion of the own findings with reported findings in other animal models is too short and must be extended.

We agree with the reviewer and added discussion on several points that are more closely related to our dataset, including discussion about:

- The relationship between the 12/15-LOX/12-HETE pathway and the complement cascade

- The fact that opsonization of synaptosomes by C3 is not sufficient to increase microglial phagocytosis, but that it also requires n-3 PUFA deficiency-dependent modification within microglia
- The distinct roles of CR3 and C3aR in microglial phagocytosis
- The fact that synaptic refinement seems to be PS recognition- and complement-independent in n-3 sufficient mice.
- The inhibitory effect of DHA on primary microglia phagocytosis
- The biosynthesis of C22:5

The discussion of the biosynthesis of C22:5n6 is missing and should be attached to manuscript.

We fully agree with Reviewer 2. See below (two sections dedicated to this point)

Generally, data about the parent animals are missing in the manuscript. A compare of results found in parents and infants of the same group would be highly valuable. Two major aspects are missing in the manuscript:

1) The results found in the offspring mice are not compared with the parent generation. To show an impact of the n-3-PUFA deficient mice it is important to compare the fatty acid pattern in microglia and synaptosome. Here an impact of n-3-PUFA deficient could be proved over the first generation.

If we have understood the reviewers' concern, it is about the potential difference between microglia/synaptosomes lipid content in the parent vs offspring generations. In the present study, our main objective is to decipher the mechanisms by which n-3/n-6 PUFA imbalance affects CNS development. Hence, we chose to expose the dams to dietary n-3 PUFA deficiency as a way to target the neurodevelopmental phase in pups. Indeed, across our experiments, the offspring only gets access to nutrients through their mother (during gestation and lactation). The subjects are used at P21, hence, before weaning. Therefore, alterations in the offspring (microglia, neurons, lipids, etc.) are a consequence of maternal alterations.

Our protocol is as follows: Life-long standard-fed mice were exposed to either n-3 deficient or n-3 sufficient diet from mating. Consequently, the males were exposed to the diet only 4 days (the mating period) while the dams were exposed to the diet for 42 days (across gestation and lactation). Hence, males were certainly not modified in terms of body/brain lipid composition, 4 days of exposure being too short to modify lipid levels (66,67). As for the dams being exposed to the diet only for a short period of time, at adulthood (when the accretion rate of PUFAs into the brain is low), lipid modifications are rather minor (67). We here compare n-3 deficient pups to their n-3 sufficient counterparts, and not to the parents' generation, as this is not the scope of our study.

Moreover, unlike some previously published studies, we do not use multi-generational n-3 PUFA deficiency. Hence, alterations of dams' lipid composition are likely to be less profound in our model. Each mother had 2-3 litters and was put back on a standard diet for a minimum of one month between each litter. If the reviewer is asking about potential cage effect, we made sure that pups from different dams were randomized for each measurement, so that we avoid any cage effect.

Here is what we added in the **Methods** section:

After mating, CD1 females were single housed and randomly assigned to either a n-3 deficient or a n-3 sufficient diet, both containing 5% fat, in the form of sunflower oil (rich in LA, the 'n-3 deficient diet') or a mixture of different oils of which rapeseed oil (rich in ALA, the 'n-3 sufficient diet') throughout gestation and lactation(59–61,68) (Table 1 and 2). All experiments were performed on the offspring from n-3 deficient and n-3 sufficient dams at post-natal day (P) 21. Pups from different dams were used for each measurement to avoid any cage effect. Each mother had 2-3 litters and was put back on a standard diet for a minimum of one month between each litter.

2) Regrettably the alteration found in the offspring of n-3-PUFA deficient are not compared with control animals.

Here, Reviewer 2 refers to n-3 PUFA deficient animals as not being compared to control animals, i.e standard chow fed mice. We do not think that the standard chow fed animals are the right control for n-3 deficient animals. The standard chow diet is not the right control as the lipid composition is not

equivalent to the deficient diets. Thus, we must compare n-3 deficient mice to animals that are fed the same diet, expect for the PUFA composition. i.e. n-3 PUFA sufficient.

Below is the PUFA composition of the three diets. One can appreciate that the composition of the standard chow diet is closer to the n-3 deficient diet in terms of LA, total n-6 PUFA, total PUFAs, while it is more similar to the n-3 sufficient diet in terms of LA/ALA ratio and ALA. Hence, we used the n-3 sufficient diet as our control as it more consistently contains higher levels of n-3 PUFAs and lower levels of n-6 PUFAs than the n-3 deficient diet. The content of proteins, carbohydrates, and lipids varies as well (3.1% of lipids in the standard chow vs 5% in the n-3 sufficient diet).

Extracted from Table 2 of the manuscript: Fatty acid composition of the diets (% wt of total fatty acids)

diets	n-3 deficient	n-3 sufficient	Standard chow (A04)
18:2n-6 (LA)	57.4	10.6	45.9
Other n-6 PUFAs	n.d.	n.d.	0.3
total n-6 PUFAs	57.4	10.7	46.2
18:3n-3 (ALA)	0.2	1.6	3.3
20:5 n-3	n.d.	n.d.	0.6
22:5 n-3	n.d.	n.d.	0.1
22:6 n-3	n.d.	n.d.	0.8
total n-3 PUFAs	0.2	1.6	4.8
total PUFAs	57.6	12.3	51.0
LA/ALA	287	6.6	13.9

FAs, fatty acids; PUFAs, polyunsaturated fatty acids; LA: linoleic acid; ALA, α -linolenic acid.

Moreover, we did find differences between standard-fed animals and n-3 sufficient mice, esp. regarding the regulation of microglial phagocytic activity towards spines (see Reviewer 1's comment and answer). Yet, it was not the aim of our study to understand the intrinsic differences between standard chow and n-3 sufficient groups as we are concerned with a defined n-3 PUFA deficiency that is prevalent in human populations.

We added this information in the **Methods** section. It appears as follows:

"The composition of the standard chow diet is closer to the n-3 deficient diet in terms of LA, total n-6 PUFA, total PUFAs, while it is more similar to the n-3 sufficient diet in terms of LA/ALA ratio and ALA. Hence, we used the n-3 sufficient diet as our control as it more consistently contains higher levels of n-3 PUFAs and lower levels of n-6 PUFAs than the n-3 deficient diet. The content of proteins, carbohydrates, and lipids varies as well (3.1% of lipids in the standard chow vs 5% in the n-3 sufficient diet)".

Since both n-3 PUFA sufficient and deficient mice were caged with different diets, an effect of the diet independent from the parent generation cannot be shown.

Our long-term ambition is to understand how maternal nutrition can affect the offspring's neurodevelopment. We do expect that the effects observed in the pups are due to modifications in the dams, as the pups are not weaned yet when we study them. We think that this experimental setup is clinically highly relevant as Human fetuses and young infants have a limited ability to synthesize n-3 PUFAs *de novo*. They are supplied via maternal (placental transfer, breastmilk) or external (enriched formula) sources. The recent decline in n-3 PUFA consumption relative to n-6 PUFA in many Western countries has raised concern about its potential detrimental effects on the neurodevelopment of human infants (reviewed in 59).

Moreover, as stated above, if the reviewer is asking about potential cage effect, we ensured that pups from different dams were randomized for each measurement, so that we avoid any cage effect.

This appears in the **Methods** section:

"Pups from different dams were used for each measurement to avoid any cage effect. Each mother had 2-3 litters and was put back on a standard diet for a minimum of one month between each litter."

Generally, the data set is really weak and I recommend to increase the number of experiments.

If the reviewer means more new experiments, we have happily provided many of these throughout our response which significantly strengthens our conclusions. If the reviewer means increase the numbers within experiments (as alluded to later) we have conducted experiments along lines that are classically used in the literature and are based on previous/pilot experiments ran in the lab that have provided

results with statistical significance. On average, we used (after removal of the outliers, see below for the calculation of outliers): N=5 for Golgi staining experiments; N=5-8 for Western blotting; N=5-10 for immunostaining; N=10-17 for Y-maze; N=4 for lipidomics; N=7-8 for ELISA; N=3-20 for ex-vivo phagocytosis; N=4-6 for transcriptomics; N=5-14 (each N containing duplicates or triplicates for each measurement) for in vitro studies; N=3 for electron microscopy (>600 measurements over the 3 animals). We respected the 3R rules, by not unnecessarily increasing the numbers of animal used in our experiments.

A more detailed description of the performed experiments and animal caging is necessary.

Because of the word limit and the wide variety of techniques used in our study, we had to reduce the amount of details in the methods section. However, we agree with Reviewer 2 (and reviewer 3) that some information was missing. Hence, we added a more detailed description of the methods used when necessary (see bolded text in the **Material and Methods section**). We also added information on animal caging.

It appears as follows:

“After mating, CD1 females were single housed and randomly assigned to either a n-3 deficient or a n-3 sufficient diet, both containing 5% fat, in the form of sunflower oil (rich in LA, the ‘n-3 deficient diet’) or a mixture of different oils of which rapeseed oil (rich in ALA, the ‘n-3 sufficient diet’) throughout gestation and lactation (59–61,68) (Table 1 and 2) (...) All experiments were performed on the offspring from n-3 deficient and n-3 sufficient dams at post-natal day (P) 21. Pups from different dams were used for each measurement to avoid any cage effect. Each mother had 2-3 litters and was put back on a standard diet for a minimum of one month between each litter”

Unfortunately, I have to recommend the rejection of this manuscript but encourage the authors to resubmit their work with are more sound data set.

Major:

Figure 1B) The authors show that Dendritic spine density of Golgi stained CA1 pyramidal neurons was significantly decreased in n-3 deficient mice compared to n-3 sufficient mice. (line 102-104) Please explain the difference of the number of data points between the N-3 sufficient and the n3-deficient in total numbers of junction. Here is a massive difference from 15 vs. 5. This fact might lead to misleading results and wrong significance. Also the graphic for the total dendritic length is confusing. Please increase the number of data points for the n-3 deficient.

We fully agree with Reviewer 2. Below is the method section relating to this issue and that explains the difference between the groups:

As stated in the **Figure’s legend**: *“For dendritic arborization and total dendritic length, n=5 neurons from 3 mice for n-3 deficient mice and n=15 neurons from 4 mice for n-3 sufficient mice”*. So the number of animals is almost the same in both groups. And in the **methods** section: *“For analysis, we randomly selected pyramidal neurons from the CA1 region of the dorsal hippocampus that were fully penetrated by the Golgi coloration and distinguishable from other neurons (n-3 deficient mice had less usable neurons than n-3 sufficient mice)”*. The reason for the higher number of neurons counted in sufficient group was that number of well-stained neurons was higher in n-3 sufficient mice, we choose the best-stained neurons in both conditions in a blinded fashion to completely remove bias. Thus, we do not think that our statistical analysis is invalid.

In order to verify that these differences were biasing the statistical analysis or not, we randomly chose 5 animals in the n-3 sufficient group that we compare to the 5 n-3 deficient mice (measure of **dendritic length**). We replicated this experiment 4 times. Below are presented the raw data. Figures in blue and italic are excluded from the statistical analysis. P-values appear in bold at the bottom. In all cases, we reach are we are close to significance. Hence, we do not think that the decrease in dendritic length that we observed in a statistical artifact.

n-3 diet	n-3 deficient	n-3 diet	n-3 deficient	n-3 diet	n-3 deficient	n-3 diet	n-3 deficient diet
2364,993*	1805,673	2364,993	1805,673	2364,993*	1805,673	2364,993*	1805,673
1403,361*	1472,531	1403,361*	1472,531	1403,361	1472,531	1403,361*	1472,531
1580,792	1918,045	1580,792*	1918,045	1580,792*	1918,045	1580,792*	1918,045
2848,948*	1474,958	2848,948*	1474,958	2848,948	1474,958	2848,948	1474,958
2500,859*	1819,187	2500,859	1819,187	2500,859*	1819,187	2500,859*	1819,187
2555,685*		2555,685		2555,685*		2555,685	
2288,226		2288,226*		2288,226*		2288,226*	
2535,949*		2535,949*		2535,949		2535,949*	

2057,843*	2057,843	2057,843	2057,843*
1497,776*	1497,776*	1497,776*	1497,776
2070,721	2070,721	2070,721*	2070,721*
2658,921*	2658,921*	2658,921*	2658,921
2692,451*	2692,451*	2692,451	2692,451
2311,612	2311,612	2311,612*	2311,612*
2012,819	2012,819*	2012,819*	2012,819*
t-test	t-test	t-test	t-test
p=0,059	p=0,0008	p=0,059	p=0,02

Figure2B /line 119-121. The author reports the result: “At P21, i.e. at the peak of hippocampal synaptic pruning, we found significantly more contacts between Iba1-positive microglial processes and the synaptic cleft and more dendritic spine inclusions within those processes in n-3 deficient mice.” In this graphic only three data points with a wide variation are reported. Because of the wide distribution, single data points determine the average. Without the outlayer no significant results are shown. Please increase the number of data points and perform more experiments for a statistically significant result.

A N of 3 is classically used for electron microscopy experiments, considering that more than 600 measurements were made in total (n=674 for n-3 sufficient mice and n= 627 for n-3 deficient mice, i.e. 200+ measurements for each animal). This technique is tedious and time consuming (several weeks of analysis per animal), explaining why N's are generally low when using this method. Each dataset presented in the manuscript has been checked for potential outlier. Hence, each graph was made from the raw data, at the exclusion of the outliers.

Calculation of the outliers: We used the Grubbs' test, also called the ESD method (extreme studentized deviate), to determine whether the most extreme value in the list is a significant outlier from the rest. Unlike some other outlier tests, Grubbs' test only asks whether that one value is an outlier. It is not appropriate to then remove that outlier, and run the test again.

The first step is to calculate the ratio Z as the difference between the outlier and the mean divided by the SD. If Z is large, the value is far from the others. The mean and SD must be calculated from all values, including the outlier.

$$Z = \frac{|\text{mean} - \text{value}|}{\text{SD}}$$

The presence of an outlier increases the calculated SD. Since the presence of an outlier increases both the numerator (difference between the value and the mean) and denominator (SD of all values), Z cannot get as large as many expect. For example, if N=3, Z cannot be larger than 1.155 for any set of values. More generally, with a sample of N observations, Z can never get larger than $\frac{(N-1)}{\sqrt{N}}$

We added this information to the **Methods** section for clarity:

Statistical analyses

For most experiments, experimental groups were compared using Student t-test or non-parametric Mann-Whitney test (when equality of variance or normality failed). We used the Grubbs' test, also called the ESD method (extreme studentized deviate), to determine whether the most extreme value in the list is a significant outlier from the rest. Paired t-tests were used for Y-maze experiments (familiar vs novel arm for each experimental group). A Two-Way ANOVA was used for XVA-143, cRGD, and baicalein experiments (except Y-maze), ex vivo and in vitro phagocytic activity. All data were expressed as means ± standard error of the mean (SEM). A p<0.05 was considered as statistically significant.

To verify that a n=3 was not a bias in our statistical analyses, we ran new statistics on the number of microglial processes instead of the number of animals (n=600+ for both dietary groups instead of n=3).

Below are the graphs and associated statistics (**Rebuttal Figure 6**). Statistical differences are greater than with the representation method we initially used. For instance, the number of spine inclusions within microglial processes is significantly greater in n-3 deficient mice, with a p-value of 0.0071 instead of p= 0.028 when considering the number of animals. We think that keeping the number of animals for the calculation of N

instead of the number of processes is physiologically more accurate, and with this demonstration we show that we did not induce false positive. Conversely, we may be underestimated the extent of n-3 PUFA deficiency effects.

Rebuttal Figure 6. Quantification of EM data from the CA1 region of n-3 sufficient and n-3 deficient mice. Means ± SEM; n=200+ measurements per animal.

Figure 2 F. line 130-132 The author report the result: “The results revealed that the percentage of phagocytic microglia is significantly higher in n-3 deficient mice across the time points studied (Figure 2E-F).” The significant results are obtained from a compare 4 vs. 3, whereby the n3- deficient animals are (n=4). Single low value in the sufficient group and additional, fourth high value in the deficient group determine the average. Without outlier probably now significant difference. Please exclude result or perform more experiments for significant statement.

We fully agree with Reviewer 2. All outliers, if any, have been removed from the statistical analyses according to the calculation presented above. Hence, we cannot remove any other data from what is presented in the manuscript.

In a bigger picture, the Reviewer highlights some of the heterogeneity of our datasets. We should emphasize here the fact that all experiments were conducted on CD1 mice, an outbred strain. Contrary to experiments performed on C57Bl6 mice, using CD1 mice implies intrinsic inter-individual differences which we consider as an asset, considering the translational perspective of our research.

We added this information to the **Methods section** for clarity:

Animals

Animal experiments were carried out according to the Quality Reference System of INRA and approved by the local ethical committee for care and use of animals (#APAFIS 4198 and 15533). Male and female CD1 mice 8-week old were purchased from Janvier Labs (Le Gesnest St Isle, France) and were SPF-housed in temperature and humidity-controlled cages on a 12 h light/dark cycle with food and water ad libitum. The use of CD1 mice, an outbred strain, implies intrinsic inter-individual differences which we consider as an asset, considering the translational perspective of our research.

Figure 2 G) (line 151-156) The author report the result: "Using a Nanostring based mRNA chip containing 550 microglia-enriched genes, 154 of these genes were found to be significantly down-regulated and 41 genes up-regulated in microglia from n-3 deficient mice whole brains (Figure 2G-J, Table 3). Among the 40 genes reported to be unique to 155 homeostatic microglia by, 20 were significantly dysregulated, indicating that maternal n-3 PUFA deficiency alters homeostatic functions of microglia (Figure 2G)." Please extend data set to a compare six vs. six animals.

Adding two animals from one group would not be statistically sound as this would induce a strong bias. We started with 6 and 6 animals but two samples were lost because of storage issues. We think that it is more accurate to keep n=4 and 6 and make statistical analyses, rather than adding 2 animals from the same group afterwards.

Figure 3 A and B (Line 176-181) The author report the result: "We observed that maternal n-3 PUFA deficiency significantly altered the lipid profile of both microglia and synaptosomes, with an increase in n-6 fatty acids and a decrease in n-3 fatty acids (Figure 3A-B). Moreover, DHA was decreased in microglia from n-3 deficient mice (Figure 3A), while C22:5n-6 (or DPA n-6), the n-6-derived structural equivalent of DHA, was increased as a marker of n-3 PUFA deficiency (Figure 3A)." Please discuss the possibility of desaturation of C22:6n- to C22:5n6. Please compare your results with literature.

We fully agree with the comment and have added **discussion** to the manuscript. Our findings of a decrease in n-3 fatty acids and increase in n-6 fatty acids is consistent with a large body of literature (70,71). It is well established that in response to n-3 PUFA deficiency, brain DHA decreases and is replaced by DPA n-6 (70,71). We have previously shown that microglia from deficient dams have higher levels of DPA (72). While the mechanisms of this replacement at the level of the microglia are not clear, in whole brain, the uptake of DPA from the plasma free pool increases dramatically (73).

It appears as follows in the Discussion section:

"Our findings of a decrease in n-3 PUFAs and increase in n-6 PUFAs in n-3 deficient mice is consistent with a large body of literature (70,71). It is well established that in response to n-3 PUFA deficiency, brain DHA decreases and is replaced by DPA n-6 (70,71). We have previously shown that microglia from deficient dams have higher levels of DPA (72). While the mechanisms of this replacement at the level of the microglia are not clear, in whole brain, the uptake of DPA from the plasma free pool increases dramatically (73)."

Figure 3 A and B the bar graphs are hard to compare. Please modify order of bar graphs for easier overview into the data set. Add the data for ALA (C18:3n-3) into the Synaptosome part.

Reviewer 2 is right and we **edited the figure** accordingly (see **Figure 3 of the revised version**). We mentioned the same list of fatty acids for both microglia and synaptosomes and mentioned when a fatty acid was not detected (nd), as this is the case for ALA in synaptosomes for instance. We kept the color-coded presentation mode (one color per type of fatty acid). We did it for the main graphs, not for the insert, for the sake of clarity.

Please discuss the possible biosynthesis of DPA in the microglia from ALA and DHA. The differences in Fatty acid profile in microglia and Synapstosome are only small. Discuss the differences of fatty acid pattern by the biosynthesis pathway of DPA and in compare to deficient mice. Do the parent generation of these animals show the same changes in Fatty acid pattern?

It is well established that when DHA decreases in the brain that it is replaced with n-6 DPA (70,71). We have previously shown that microglia from deficient dams have higher levels of n-6 DPA (72). While it is not clear if n-6 DPA was synthesized in the microglia or transported via the blood in our study, it has been shown that upon DHA deficiency that n-6 DPA transport into the brain is increased (73). Furthermore, we have shown that addition of DHA to BV2 cells in isolation does not decrease n-6 DPA further supporting the role of transport to the microglia (74).

As for n-3 DPA, it is the intermediate between EPA and DHA, in the conversion pathway from ALA. In our hands, DPA n-3 was poorly expressed in both synaptosomes and microglia and not modulated by the dietary interventions. Hence, we did not discuss this point as it does not seem crucial.

Figure 4 H) (line 215-218) The authors report: "However, when the same approach was used in n-3 deficient vs n-3 sufficient mice, blocking MFG-E8 action on microglia did not prevent the reduction in PSD-95 expression in the hippocampus of n-3 deficient mice (Figure 4H)." Results are not significant in the graphic. Statement cannot be made up on this data set. Please make calculation again without the highest and lowest points. These determine the average and the significances.

All outliers have been removed from the statistical analyses according to the calculation presented above.

We agree with Reviewer 2 that the data distribution is heterogeneous in the n-3 sufficient group. Yet, we still observe a lower expression of the PSD-95 protein in n-3 deficient mice compared to n-3 sufficient animals and treatment with cRGD does not modify the data distribution in both groups. Data distribution is still widespread in n-3 sufficient mice and rather condensed in n-3 deficient animals. Hence, if PS recognition is involved in microglia-mediated synaptic pruning it is not detectable here.

Figure 5 A-D Line 228-229) The authors report: "Protein expression levels of CD11b (a subunit of CR3) and C1q were significantly increased in the hippocampus CA1 and DG in n-3 deficient mice (Figure 5A-B, D-E)." No Numbers of counted CD11B C1q and C3aR are given. Please show these data to claim your statement.

The quantification data appeared in panels D-F of the original version, right below the images. However, we agree with Reviewer 2 that the different lettering was bringing confusion. To avoid this, we kept the same letters for the images and for the graphs (see **Figure 4 A-C, revised version**).

Differences are caused by extreme outliers (Figure D) Please calculate without these extreme points. Don't refer to non significant values. Miss leading outcome. Why is the layout changed to bar graphs without individual points. Please uniform the data presentation to visible individual points. Figure 5G) Please erase p value from graphic since no significant differences are shown.

As above, all outliers, if any, have already been removed from the statistical analyses according to the calculation presented above.

We **added the individual points** on the graphs. We thank the reviewer for pointing out the missing information.

The authors should include more analytical controls in order to demonstrate that they do not analyses some separation fatty acyl artefacts or contaminations derived from buffers or other sources. Authors also should state which precaution they took to prevent autooxidation.

Members of the team have extensive experience with both fatty acid and oxylipin measures. Several blanks are run with each sample and authentic standards are used to identify lipid pools on TLC fatty acids or oxylipins by either GC-FID or LC/MS/MS analysis, respectively. Composite standards of lipid metabolites (natural or deuterated; Cayman Chemicals Company, Ann Arbor, MI, USA) were diluted from stock solutions in ethanol for performing an eight-point calibration curve (0.05 to 5 ng). The internal standard mixtures were prepared in ethanol and added to all composite standards and samples before extraction.

Siliconized glassware was used for extraction and sample preparations. Auto-oxidation of PUFA was minimized by extracting on ice, in reduced light conditions and with solvents containing 0.1% butylated hydroxytoluene. See (75) for more details

The authors only analysed fatty acyls therefore it is not correct to talk about lipidomics experiment or the lipidome, please remove all phrases or include lipidomics experiments including the analysis of glycerol and glycerophospholipid to make such statements.

The term "lipidomics" was mentioned once in the manuscript. We agree with the reviewer that it is overstated as we did not study all classes of lipids. Hence, the sentence has been modified as follows: "Using LC-MS-MS" instead of "Using lipidomics"

Minor

Line 109 Please give a reference for the Y-maze task. Or describe it in the material and method section.

The Y maze task is described quite extensively in the Methods section and we mention 4 references from our group describing the task. We added the age of the animals when performing the Y-maze. We also added information on our new dataset on the role of the 12/15-LOX/12-HETE pathway in memory defects observed in n-3 deficient mice. To avoid confusion, we have now entitled the section “Y maze task” instead of behavioral measurements.

Here is how it appears in the Methods section:

“Y maze task

P21-old mice were handled daily and weighed before and during behavioral experiments. Sessions were recorded with a ceiling-mounted video camera and analyzed using Smart software (Panlab, Barcelona, Spain). The Y-maze was used to assess spatial working memory as previously described (61–63,76). Each arm was 34 cm long, 8 cm wide and 14 cm high. The floor of the maze was covered with corn cob litter which was mixed between each trial to remove olfactory cues. Visual cues were placed in the testing room and kept constant during the whole test. In the first trial, one arm was closed with a guillotine door and mice were allowed to visit two arms of the maze for 5 min. After a 30-min to 3h inter-trial interval (ITI), mice were placed back in the start arm and allowed free access to the three arms for 5 min. Start and closed arms were randomly assigned for each mouse. When required, baicalein was injected intraperitoneally at a concentration of 20 mg/kg (58), at P14, P17 and 30 min prior the first trial (at P21 or P30 depending on the experiments). Data are presented as the time spent exploring the novel and the familiar arms during the second trial”.

Figure 1 D) Please show the fold changes also the bar graphs for the n3 sufficient animals with individual data points and variance.

We edited the Figure accordingly

Figure 2 I) Network graphic is too small. Key interactions of proteins are not visible. Please change size. Or leave it out of the manuscript.

We have increased the size of network graph. The utility of this graph is to show that there are indeed three distinct gene clusters, highlighting their correlation, therefore we would like to keep it in the figure.

Figure 2 J) Because of the different case number not valuable. Please identify the individual animals with number to see the effect of each individual.

We now realize that we had not adequately described the graph in the legend. Each point represents an animal within the group indicated and the value is average gene expression of all genes within that cluster. We have added text to the legend for clarity.

‘J. Mean expression profile of all transcripts within clusters 1, 2 and 3 where each point represents an animal and the average gene expression of all genes within that cluster for that animal.’

Figure 6 and C) Please divide axis for the concentrations of low abundant components. Please change to nmol/mg or other appropriate molar concentration. Figure 6) generally concentrations are missing in the graph and in the figure legends. Please correct this error.

We pooled 4 brains for n=1. Hence, “ng/sample” means ng/4 brains. We sort around 400 000 microglia from the brain of P21 CD1 mice. Hence, the numbers correspond to 1.6 million of cells. We left the legend as it was in the original version (see **Figure 5A**), yet we added the information in the Material and Methods section. It appears as follows:

Because lipid analyses requiring lots of material, we pooled microglia from 4 brains for n=1. Based on our experience, we sort 400 000 cells out of the brain of P21 mice, hence measurements were made on around 1.6 million microglia. It is noted in ng/sample in the legend.

Figure 7C +D) It would be nice to have the base line control to Time point 0.

We never measured the phagocytic activity of microglia at time point 0 for two reasons:

- 1) It takes a significant amount of time to treat the cells with the synaptosomes, take them to the FACS machine for measurement and start analyses. The earliest time point we ever studied was 30 minutes post-synaptosomes application. In all the conditions tested, the percentage of phagocytic cells was always very low and similar between all groups at this time point. Differences were observed between groups starting at 1h post-treatment.
- 2) One of our co-authors, Dr A Sierra, elegantly showed it takes on average 1.5h for microglia to engulf apoptotic cells (77). Even though synaptosomes are smaller elements, hence, likely to be engulfed more rapidly, the phagocytic process requires time. Microglia must i) be in contact with the neuronal element to be removed, ii) recognize the particle to be eaten as such, iii) activate intracellular signaling pathway/transcriptomic modifications, to iiiii) modify their cytoskeleton so they can enwrap the element to be phagocytosed and iiiiii) digest it.

Reviewer #3 (Remarks to the Author):

In this article by Madore et al., the authors investigate the impact of omega-3 fatty acids (n-3 PUFAs) deficiency during pregnancy and lactation onto the phagocytic activity of microglia in the hippocampus of the offspring and related neural circuit deficits. The authors first show that deficient diet triggers a reduction in the density of spines in CA1 pyramidal cells, memory deficits as well as an increase in microglial synaptic phagocytosis and an alteration in their transcriptomic signature. To investigate the underlying mechanism, the authors further show that diet deficiency, while modifying the lipid composition of both microglia and neuronal synaptosomes, increases the capacity of microglia to engulf synaptosomes from either controls or diet-deficient neurons. The synaptic reduction was not modified by altering phosphatidyl-serine pathway but was rescued by altering the classical complement cascade. Remarkably, blocking of the complement cascade was sufficient to rescue the memory deficit triggered by a deficient diet. Finally, the authors showed that oxylipins, bioactive mediators which are rapidly converted from PUFAs, are modified in diet-deficient microglia. Specifically, 12-HETE, enhanced microglial phagocytosis and pharmacological shunting of its production was sufficient to reduce hippocampal microglial phagocytosis.

Overall, this is an impressively large experimental study that uses a variety of techniques to dig into the relationship between diet modifications, the impact on microglia and related neuronal circuits. The results are convincing, complete, and provide an important link between nutrition and brain wiring, microglial involvement, from lipidomics to behavior. These findings should be of interest to the broad readership of Nature communications.

We thank the reviewer for this very positive feedback on our work.

Still, there are a few issues that need to be addressed before the paper is suitable for publication.

Comments:

-The extent of the experimental results presented makes it difficult to link the different conclusions. In particular, while the authors nicely show that diet-deficient induced microglia have an enhanced phagocytic capacity that relies on the complement cascade, it is unclear how this relates to the modifications in oxylipin profiles and whether the two pathways are parallel/convergent/overlapping. Is C3Ra expression induced by 12-HETE? Or do C3R and 12-HETE pathways act in parallel onto the phagocytic capacity of microglia?

We agree with the reviewer (and with reviewer 1 who also raised this issue) that the link between the 12/15-LOX/12-HETE pathway and the complement cascade required further assessment.

We have added significant experiments to address this. Below is now in the **Result section**:

To decipher further the link between the 12/15-LOX/12-HETE pathway and the complement cascade, we exposed freshly sorted n-3 sufficient or n-3 deficient microglia to the 12/15-LOX inhibitor baicalein or its vehicle and quantified the expression level of CD11b and C3aR mRNAs, 2h later (**Figure 7A of the revised version; see below**). We show that *c3ar* gene expression was increased in n-3 deficient microglia while the expression of *cd11b* was not significantly modulated by the diet. This is in line with our immunostaining experiments in which CD11b was moderately increased under n-3 PUFA deficiency, only in the DG, while C3aR was robustly increased in the whole hippocampus (Figure 4B of the revised version). Baicalein significantly reduced C3aR expression in n-3 deficient microglia, while it

had no effect on CD11b (**Figure 7B-C**). These results suggest that in non-stimulated n-3 deficient microglia, the 12/15-LOX/12-HETE signaling pathway is implicated in the maintenance of a high C3aR tone, while the activity of CD11b (as shown in Figure 4E-F of the revised version) is likely to depend on other mechanisms.

Once activated, the expression of most immune receptors is dampened as part of a global homeostatic mechanism (47). We thus examined the relationship between the 12/15-LOX/12-HETE signaling pathway and the complement cascade in condition of stimulation with synaptosomes. We exposed freshly sorted cells to pHrodo synaptosomes in the presence of baicalein or its vehicle and quantified the expression level of *cd11b* and *c3ar* (**Figure 7D**). The expression level of CD11b was significantly increased by baicalein in n-3 deficient microglia, whereas inhibiting the 12/15 LOX enzyme did not affect C3aR mRNA expression (**Figure 7E-F**). This suggests that once microglia have started phagocytosing synaptosomes, the 12/15-LOX/12-HETE signaling pathway inhibits CD11b expression while allowing phagocytosis of neuronal material (**Figure 6F of the revised version**).

Figure 7, revised version: The 12/15-LOX/12-HETE signaling pathway controls gene expression of the complement pathway. **A.** Experimental setup. **B-C.** Quantification of *cd11b* and *c3ar* mRNA expression in freshly sorted n-3 deficient and n-3 sufficient microglia, treated with baicalein or its vehicle. Means \pm SEM; n=4-6 per group. Two-way ANOVA: *cd11b*: diet effect, $F(1,15)=0.57$, $p=0.46$; treatment effect, $F(1,15)=1.37$, $p=0.26$; interaction, $F(1,15)=2.48$, $p=0.14$; *c3ar*: diet effect, $F(1,18)=1.36$, $p=0.26$; treatment effect, $F(1,18)=1.52$, $p=0.23$; interaction, $F(1,18)=4.36$, $*p=0.051$. **D.** Experimental setup. **E-F.** Quantification of *cd11b* and *c3ar* mRNA expression in freshly sorted n-3 deficient and n-3 sufficient microglia, exposed to synaptosomes and treated with baicalein or its vehicle. Means \pm SEM; n=8 per group. Two-way ANOVA followed by Bonferroni *post-hoc* test: *cd11b*: diet effect, $F(1,28)=0.16$, $p=0.19$; treatment effect, $F(1,28)=0.92$, $**p=0.0028$; interaction, $F(1,28)=0.072$, $p=0.36$; n-3 deficient+vehicle vs n-3 deficient+ baicalein, $*p=0.014$; n-3 deficient+ vehicle vs n-3 deficient+ baicalein, $*p=0.029$; *c3ar*: diet effect, $F(1,28)=0.35$, $**p=0.0034$; treatment effect, $F(1,28)=0.0043$, $p=0.72$; interaction, $F(1,28)=0.0019$, $p=0.81$.

This novel dataset suggests that the 12/15-LOX/12-HETE signaling pathway is the master controller of microglial phagocytosis under n-3 PUFA deficiency, partly by regulating the expression of the C3aR and CR3 receptors expression.

We also added some **discussion** about this topic in the Discussion section. It appears as follows:

"We further deciphered the relationship between the 12/15-LOX/12-HETE pathway and the complement system. Our data reveal that the phagocytic capacity of microglia is increased only when the cells are n-3 deficient, even though C3 is overexpressed on n-3 deficient synaptosomes. These data suggest that both opsonization of synapses and recognition of C3 by its microglial receptors are

required to trigger phagocytosis. This confirms a plethora of reports on the complement-dependent phagocytosis in the immune system. It is already well described that there is little to no ingestion of opsonized elements without phagocyte activation (78,79). Conversely, CR3-mediated phagocytosis is dependent on receptor conformational changes, lateral diffusion and clustering, processes that require activating stimuli (24–29). Hence, binding of C3 to the complement receptors can be considered as the recognition (“find-me”) mechanism of spines to be removed by microglia, as previously shown in other experimental contexts (19,30–32), while activation of the 12/15-LOX/12-HETE pathway is the triggering signal for phagocytosis by microglia. This is coherent with the observation that the complement cascade classically interacts with inflammatory pathways to control the activity of phagocytes (33).

Similarly, is C3R found in the Nanostring experiment presented in Figure 2 or only detected by immunostaining? It appears in Table 3, but is not represented in Figure 2G. Clarifying these issues would be helpful for the reader; convey the conclusion and strengthen the proposed framework.

Both CD11b (CR3) and C3aR mRNA were quantified (qPCR on the whole hippocampus and Nanostring experiments respectively). We also evaluated their protein expression and location by immunohistochemistry. In all cases, the expression of C3aR (mRNA and protein) was dramatically increased in microglia cells (**Figure 2G and 4C, revised version**). C3aR is one of the most up-regulated genes in n-3 deficient mice as compared to n-3 sufficient group (ratio n-3 deficient/n-3 sufficient=4.4). We thus highlighted this gene in Figure 2G as well. Conversely, the increase in CD11b immunostaining was rather slight and limited to the dentate gyrus of the hippocampus (**Figure 4B, revised version**). We also observed a significant increase in the expression of CD11b mRNA when assessing the whole hippocampus (of note, CD11b does not appear in the microarray setup by our collaborator as it is not considered as a marker of the homeostatic signature of microglia).

C3a, that binds to microglial C3aR, is usually considered as a chemoattractant signal, while C3b, that binds to CR3 (CD11b+CD18) opsonizes the target to be removed. As stated above, CR3-mediated phagocytosis is dependent on receptor conformational changes, lateral diffusion and clustering, rather than on the modulation of its expression (24–29). Hence, these data are in line with our observations that the 12/15-LOX pathway is a potent modulator of C3aR expression on one hand and of CR3 activity on the other hand.

We edited the **result section** accordingly to strengthen the conclusion that n-3 PUFA deficiency deeply affects the complement system on the hippocampus of mice. We also edited the **discussion part** of the manuscript. Here the added sentences:

Result section:

“The complement system acts as a recognition mechanism by which C1q, the initiating protein of the cascade, tags synapses to be removed (80). C1q-tagging leads to the cleavage of C3 and subsequent deposition of C3a and C3b, which activates CD11b and C3aR respectively on microglia and triggers synaptic elimination by microglial (19,80,81). C1q immunostaining was significantly increased in both the hippocampus CA1 and DG in n-3 deficient mice (Figure 4A). Protein expression levels of CD11b (a subunit of CR3) was slightly, yet significantly increased in the DG in n-3 deficient mice (Figure 4B). CD11b gene expression was also enhanced in the whole hippocampus in these mice (Supplementary Figure 5). Our analyses also revealed a robust increase of C3aR immunostaining in the whole hippocampus of n-3 deficient mice, corroborating microglia transcriptomic data (Table 3). (Figure 4C; Figure 2G; Table 3)”.

Discussion section:

Interestingly, C3a, that binds to microglial C3aR, is usually considered as a chemoattractant signal, attracting immune cells including microglia towards the structures to be cleared, while C3b, that binds to CR3 (CD11b+CD18) opsonizes the target to be removed (33,81–84). Our data show that the 12/15-LOX/12-HETE pathway controls the basal expression of C3aR in n-3 deficient microglia, suggesting that it may control the capacity of microglia to reach the target. However, the 12/15-LOX/12-HETE pathway does not regulate the gene expression of CR3, which is in line with the observations that CR3 activation is more dependent on conformational changes and redistribution in the cellular membrane rather than on transcriptional regulation (24–29). As for CR3 activity, our data suggest that the 12/15-LOX/12-HETE pathway is involved in the homeostatic down-regulation of CR3 expression once the opsonized elements have been recognized by microglia (83). Overall, we show that the 12/15-LOX/12-HETE pathway is activated upstream of the complement cascade, controlling the gene

expression level and activity of this latter. We also demonstrate that this lipid pathway is the master controller of microglial phagocytic activity as its inhibition blunts synaptosomes engulfment”.

On a similar note, it is difficult to link the data on PS with the rest of the manuscript (Figure 4 and Figure Supplemental 5). The two figures lead to slightly different conclusions that are not fully explained. The authors should either provide a clearer conclusion for these experiments. Alternatively, they could be moved to supplemental or removed from the paper.

We fully agree with Reviewer 3. The discrepancy between Figure 4 and Supplementary Figure 5 add confusion to the manuscript. Following Reviewer 3's advice, we have removed Supplementary Figure 5 from the manuscript and put Figure 4 as supplementary material. Reviewer 1 raised the same issue and here is what we answered:

In the experiment originally presented in supplementary Figure 5, our goal was to assess the efficiency of cRGD in a condition in which PS recognition is known to be involved in synaptic pruning in the developing hippocampus (1–4). At the same time, we were bringing additional data to the field on the potential role of PS recognition in synaptic pruning in the hippocampus under standard chow diet.

From this positive control, we can conclude that the absence of effect of cRGD on PSD95 expression in the n-3 deficient group is not due to a lack of efficiency of the peptide but rather to the fact that PS recognition is not involved in synaptic refinement in these conditions. This suggests that when maternal n-3 PUFA intake is low, microglia switch to a PS-independent/complement-dependent recognition mechanism in the offspring.

As mentioned by the reviewer, this dataset also highlights intrinsic differences between standard and n-3 sufficient mice and questions the underlying mechanisms. In n-3 sufficient mice, neither PS recognition nor the complement cascade seem to be involved in synaptic refinement. This suggests that 1) other mechanisms of microglia-mediated synaptic refinement are in place in the hippocampus of these animals. Indeed, microglia uses a plethora of recognition mechanisms for phagocytosis (5,6) and few studies have addressed the microglia-spines interactions in the developing hippocampus (in standard-fed animals); 2) that microglia-mediated synaptic refinement is not prominent in the hippocampus of these animals.

Synaptic pruning occurs via various mechanisms, including microglia-independent processes, though no studies have ever quantified the proportion of microglial-mediated synaptic pruning in the developing hippocampus. Among other mechanisms known to mediate synaptic pruning, neuronal macroautophagy may play a role, as it does in the cortex (7,8), or astrocyte-mediated mechanisms as shown in the developing brain (9). Hence, the proportion microglia-independent synaptic pruning mechanisms in the developing hippocampus of n-3 sufficient mice may well be higher than anticipated.

We added some **discussion** about these aspects in the Discussion section. It appears as follows:

Of note, neither PS recognition nor the complement cascade seem to be involved in synaptic refinement in n-3 sufficient mice. This suggests that 1) other mechanisms of microglia-mediated synaptic refinement are in place in the hippocampus of these animals or 2) microglia-mediated synaptic refinement is not prominent in the hippocampus of these animals. Microglia use a plethora of recognition mechanisms for phagocytosis of neuronal elements/dead bodies (5,6) but few studies have addressed the microglia-spine interaction in the developing hippocampus. On the other hand, several mechanisms, including microglia-independent processes such as neuronal macroautophagy or phagocytosis by astrocytes, control synaptic pruning in the developing brain, though the proportion of synaptic pruning relying on microglia in the developing hippocampus is not known (7–9). Our data suggest that synaptic pruning mechanisms in the developing hippocampus of n-3 sufficient mice may favour microglia-independent processes.

In any case, these observations reinforce the necessity to use n-3 sufficient mice as controls in the present study, and not standard-fed animals. It also reveals that PUFAs modulate the phagocytic activity of microglial and that their levels must be controlled when studying such mechanisms of microglia-spine interactions in the developing brain.

We previously published the fatty acid composition of the cortex of adult mice fed either a standard, n-3 sufficient or n-3 deficient diet (10). Cortices from standard-fed and n-3 sufficient-fed animals are quite similar in terms of fatty acid composition (see **Rebuttal Figure 1** below). Hence, differential lipid composition at the structure level is unlikely to explain the difference between both groups.

Rebuttal Figure 1: Cortex fatty acid composition of mice fed a standard chow or a n-3 sufficient diet. Data are expressed as % of total fatty acid (Extracted from (10)).

However, when looking at the fatty acid composition of the 2 diets (see data below, extracted from Table 2 of the manuscript), one can notice differences between the standard and n-3 sufficient diet (e.g. total saturated fat, monounsaturated, PUFAs, LA/ALA ratio). The content of proteins, carbohydrates, and lipids varies as well (3.1% of lipids in the standard chow vs 5% in the n-3 sufficient diet).

Table 2: Fatty acid composition of the diets (% wt of total fatty acids)

diets	Sufficient	Standard chow (A04)
16:0	22.6	20.3
18:0	3.3	2.2
other saturated FAs	1.8	1.8
total saturated FAs	27.7	24.3
18:1n-9	57.9	19.3
18:1n-7	1.5	1.5
other monounsaturated FAs	0.4	3.9
total monounsaturated FAs	60.0	24.7
18:2n-6 (LA)	10.6	45.9
Other n-6 PUFAs	n.d.	0.3
total n-6 PUFAs	10.7	46.2
18:3n-3 (ALA)	1.6	3.3
20:5 n-3	n.d.	0.6
22:5 n-3	n.d.	0.1
22:6 n-3	n.d.	0.8
total n-3 PUFAs	1.6	4.8
total PUFAs	12.3	51.0
LA/ALA	6.6	13.9

FAs, fatty acids; PUFAs, polyunsaturated fatty acids; LA: linoleic acid; ALA, α -linolenic acid.

Hence, other aspects than brain fatty acid composition are likely to explain differences between both groups, such as difference in cell energy metabolism, difference in lipid composition at the cellular resolution or difference in the production of lipid derivatives.

That being said, because the question of the role of PS-dependent recognition mechanisms in synaptic pruning in the developing brain is not our focus, and as suggested by reviewer 3, we moved Figure 4 to supplementary material (Supplementary Figure 7) and removed the dataset presented in Supplementary Figure 5 from the manuscript. We think that elaborating on this aspect does not add relevant information on how n-3 deficiency increases microglial phagocytosis of spines.

-Regarding the potential link with human diseases, the authors make a strong correlation with neuropsychiatric diseases such as ASD and Schizophrenia, but mostly focus on hippocampus/memory. It would be important to selectively detail what has been linked to cognitive/memory impairment in relationship to n3-deficiency in humans. As well as to detail the potential links with malnutrition (which types of diets are comparable to the one used in the experimental paradigm?).

We acknowledge Reviewer 3's request to have more precisions concerning the link between dietary n-3 PUFA deficiency and neurodevelopmental disorders on one hand and the focus on the hippocampus and memory on the other hand.

The Western diet, that arose in industrialized countries over the last century, is low in n-3 PUFAs (including the precursor ALA), with an estimate of 85% of individuals who do not reach dietary recommendations (86,87). The entire population is concerned by this global imbalance in n-3 /n-6 PUFA dietary intake, including pregnant women and infants (88,89). The low consumption of ALA (which lead to a high LA/ALA ratio) by pregnant women has a direct consequence on the biosynthesis of DHA from ALA and possibly its accretion in the developing brain (89). As previously mentioned, an adequate early-life dietary content in n-3 PUFA is critical for proper brain development and cognition (90), even if the exact dose and type of n-3 PUFAs that have to be provided during pregnancy and breastfeeding period is still a matter of debate. However, the precise effects of n-3 PUFA dietary unbalancing on brain development and on higher-order brain processing, including learning and memory and cognition, are still poorly understood.

Several epidemiological studies pinpoint that high dietary intake of n-3 PUFA correlates with better cognitive functions, especially in elderly (91,92), and higher brain/hippocampus volume (91,92,93). In addition, connectome-wide association approaches revealed that the n-3 PUFA status of an individual (which can be measured from red blood cells) is positively correlated to the functional connectivity within frontal and temporal lobed structures of healthy elderly (95). Overall, epidemiological and clinical studies pinpoint n-3 PUFA nutritional status as strongly associated with memory and hippocampus/prefrontal cortex functioning in Humans.

Over the last decade, dysregulated processing and altered nutritional abundance of n-3 PUFA have been associated with neurodevelopmental disorders, including autism spectrum disorders (ASD), schizophrenia and intellectual disability (17,96). The lack of n-3 PUFA during pregnancy and early post-natal life could participate in the etiology of these diseases. Of special interest is the limbic system as the hippocampus activity in Humans has been reported to be altered in subjects with low n-3 PUFA status and consistently altered in animal models of n-3 PUFA deficiency (62,69,97). Hippocampal alterations are reported in several neurodevelopmental disorders, including ASD, that could account for a neurodevelopmental compensatory mechanism to preserve cognitive domains (99,100). During development, DHA accumulates in the human brain at a high rate during the third trimester of pregnancy (98). Preterm infants display lower brain DHA concentrations as compared to term infants on the same ALA fortified diet (101). A very low birth weight is a high-risk factor for both n-3 PUFA deficiency and neurodevelopmental disorders/cognitive impairments (102). These children display improved recognition memory at 6-months of age when supplemented with LC n-3 PUFA (103) while brain structure was unchanged later in life (8 years old) (104). Whether the pathophysiology of cognitive alterations, a common feature of neurodevelopmental disorders, may be linked to inadequate bioavailability of n-3 PUFA in the hippocampus is of high interest in part due to the increasing knowledge on the role of nutrition in psychiatric disorders and in particular ASD, ADHD and schizophrenia.

Overall, this knowledge open new avenue for the use of specific dietary supplementation with n-3 PUFA, as an environmental modifiable factor. Indeed, dietary supplementation or increase intake of food rich in n-3 PUFA could counteract the adverse effect of low n-3 PUFA intake.

We added this information in the **Discussion** section, as follows:

Previous clinical and epidemiological studies revealed a relationship between dietary n-3 PUFA content, brain development and the prevalence of neurodevelopmental disorders(17). The n-3 PUFA index, which reports the level of n-3 PUFAs (EPA+DHA) in erythrocytes and that is used as a biomarker for cardiovascular diseases(105), is now considered as a new biomarker for several neurodevelopment diseases such as ASD(106), attention deficit hyperactivity disorder (ADHD)(107) and schizophrenia(108,109). The lack of n-3 PUFA during pregnancy and early post-natal life could participate in the etiology of these diseases. Since these essential fatty acids are necessary for healthy brain development, low dietary supply or impairment of their metabolism is suggested to be involved in the etiology of several psychiatric diseases, including neurodevelopmental disorders(67,110,111). Indeed, decreased DHA levels have been reported in several brain regions of patients diagnosed with such disorders(17,112–114). Low n-3 PUFA index is also associated with reduced brain size or connectivity(115,116), as well as to reduced cortical functional connectivity during an attentional task(116). Of special interest is the limbic system as the hippocampus activity in Humans has been reported to be altered in subjects with low n-3 PUFA status and consistently altered in animal models of n-3 PUFA deficiency (62,69,97). Hippocampal alterations are reported in several neurodevelopmental disorders, including ASD, that could account for a neurodevelopmental compensatory mechanism to preserve cognitive domains (99,100). Preterm infants display lower brain DHA concentrations as compared to term infants on the same ALA fortified diet (101). A very low birth weight is a high-risk factor for both n-3 PUFA deficiency and neurodevelopmental disorders/cognitive

impairments (102). These children display improved recognition memory at 6-months of age when supplemented with LC n-3 PUFA (103) while brain structure was unchanged later in life (8 years old) (104). Whether the pathophysiology of cognitive alterations, a common feature of neurodevelopmental disorders, may be linked to inadequate bioavailability of n-3 PUFA in the hippocampus is of high interest in part due to the increasing knowledge on the role of nutrition in psychiatric disorders and in particular ASD, ADHD and schizophrenia.

Overall, this knowledge open new avenue for the use of specific dietary supplementation with n-3 PUFA, as an environmental modifiable factor. Indeed, dietary supplementation or increase intake of food rich in n-3 PUFA could counteract the adverse effect of low n-3 PUFA intake. Our data shed light on an oxylipin-dependent mechanism underlying these clinical observations.

-Some of the images are not as convincing as they could be. In particular, Figure 5 Panels A-C underexposed and it is difficult to see how the staining in CD11b, C1q and C3R. Similarly, in Figure 2 Panel C, it is very difficult to see the red PSD95 puncta inside the microglia. The authors should provide higher magnification, different contrasts or multiple panels for these immunostainings to strengthen their claims.

Images have been **modified and/or contrasted** to better highlight staining and strengthen our claims.

-Because of the wide arrays of approaches used throughout the study, the authors need to make an extra mile to clarify all the techniques and their presentation in the figures. Please find below some specific comments:

° Figure 1B- "spine density (relative fold change)" is unclear as how it is measured? fold change compared to what?

We thank the reviewer for pointing out this mistake. The graph represents the number of spines per 10µm of dendrites. The graph has been edited accordingly.

We added some explanations to the **Method section**: "*Spine density in these neurons was determined by counting the number of spines on at least 3 basal and 3 apical dendritic segments of 10 µm in length. Segments from dendrites situated as far from the cell body as possible, with no overlap with other dendrites were randomly selected. Primary dendrites were never used for analysis as their thickness hampers detection of spines. Spine density was calculated per 10 µm and averaged across the different segments in the same neuron.*"

° Figure 2 Panel A: the labeling on the EM pictures are way to small and almost not possible to read
We increased the size of the labeling.

° Figure 2 Panel E: the first description of the pHrodo experiment (text line 130) does not explain the experimental approach, but it is presented for Figure 7 (text line 279). The authors should present the experimental approach linearly to avoid confusing the reader.

We **added the following text** when first describing the pHrodo experiments: *The pHrodo dye-labelled E. coli becomes fluorescent upon entering acidic phagosomes, enabling the assessment of phagocytic activity of microglia.*

° Figure 3 B states that synaptosomes were dissected from the hippocampus to do lipidomics, but there is no clear method section for lipidomics on synaptosomes.

We added a section dedicated to Synaptosomes in the **Method section**. We also add other information (see bolded text).

° Were Y maze experiments performed at P21 or in adults? These behavioral tests are usually performed in adults, experiments were all performed in P21. The authors could specify this point.

Y maze was performed at P21 as one of the few cognitive tasks that mice can perform well at that age. We agree with the Reviewer that this could be confusing and added the information in the text. It now appears as follows in the **Result section**:

"In the Y-maze task, a hippocampus-dependent test assessing spatial working memory, P21 n-3 deficient mice were unable to discriminate between the novel and familiar arms of the maze, unlike the

n=3 sufficient mice (Figure 1E).” And in the Methods: “P21-old mice were handled daily and weighed before and during behavioral experiments.”

References

1. Jay TR, Saucken VE von, Muñoz B, Codocedo JF, Atwood BK, Lamb BT, Landreth GE (2019): TREM2 is required for microglial instruction of astrocytic synaptic engulfment in neurodevelopment. *Glia* 67: 1873–1892.
2. Filippello F, Morini R, Corradini I, Zerbi V, Canzi A, Michalski B, *et al.* (2018): The Microglial Innate Immune Receptor TREM2 Is Required for Synapse Elimination and Normal Brain Connectivity. *Immunity* 48: 979–991.e8.
3. Weinhard L, di Bartolomei G, Bolasco G, Machado P, Schieber NL, Neniskyte U, *et al.* (2018): Microglia remodel synapses by presynaptic trogocytosis and spine head filopodia induction. *Nat Commun* 9: 1228.
4. Scott-Hewitt NJ, Perrucci F, Morini R, Erreni M, Mahoney M, Witkowska A, *et al.* (2020): Local externalization of phosphatidylserine mediates developmental synaptic pruning by microglia. *bioRxiv* 2020.04.24.059584.
5. Brown GC, Neher JJ (2014): Microglial phagocytosis of live neurons. *Nat Rev Neurosci* 15: 209–216.
6. Diaz-Aparicio I, Beccari S, Abiega O, Sierra A (2016): Clearing the corpses: regulatory mechanisms, novel tools, and therapeutic potential of harnessing microglial phagocytosis in the diseased brain. *Neural Regen Res* 11: 1533–1539.
7. Tang G, Gudsnuk K, Kuo S-H, Cotrina ML, Rosoklija G, Sosunov A, *et al.* (2014): Loss of mTOR-dependent macroautophagy causes autistic-like synaptic pruning deficits. *Neuron* 83: 1131–1143.
8. Lieberman OJ, McGuirt AF, Tang G, Sulzer D (2019): Roles for neuronal and microglial autophagy in synaptic pruning during development. *Neurobiol Dis* 122: 49–63.
9. Chung W-S, Clarke LE, Wang GX, Stafford BK, Sher A, Chakraborty C, *et al.* (2013): Astrocytes mediate synapse elimination through MEGF10 and MERTK pathways. *Nature* 504: 394–400.
10. Joffre C, Grégoire S, De Smedt V, Acar N, Bretillon L, Nadjar A, Layé S (2016): Modulation of brain PUFA content in different experimental models of mice. *Prostaglandins Leukot Essent Fatty Acids* 114: 1–10.
11. Bourre J-M, Pascal G, Durand G, Masson M, Dumont O, Piciotti M (1984): Alterations in the Fatty Acid Composition of Rat Brain Cells (Neurons, Astrocytes, and Oligodendrocytes) and of Subcellular Fractions (Myelin and Synaptosomes) Induced by a Diet Devoid of n-3 Fatty Acids. *Journal of Neurochemistry* 43: 342–348.
12. Cao D, Kevala K, Kim J, Moon H-S, Jun SB, Lovinger D, Kim H-Y (2009): Docosahexaenoic acid promotes hippocampal neuronal development and synaptic function. *J Neurochem* 111: 510–521.
13. Hamilton J, Greiner R, Salem N, Kim H-Y (2000): n-3 Fatty acid deficiency decreases phosphatidylserine accumulation selectively in neuronal tissues. *Lipids* 35: 863–869.
14. Vandal M, Alata W, Tremblay C, Rioux-Perreault C, Salem N, Calon F, Plourde M (2014): Reduction in DHA transport to the brain of mice expressing human APOE4 compared to APOE2. *Journal of Neurochemistry* 129: 516–526.
15. Jawaid S, Kidd GJ, Wang J, Swetlik C, Dutta R, Trapp BD (2018): Alterations in CA1 hippocampal synapses in a mouse model of fragile X syndrome. *Glia* 66: 789–800.
16. Grabert K, Michoel T, Karavolos MH, Clohisey S, Baillie JK, Stevens MP, *et al.* (2016): Microglial brain region-dependent diversity and selective regional sensitivities to aging. *Nat Neurosci* 19: 504–516.
17. Madore, Leyrolle Q, Lacabanne C, Benmamar-Badel A, Joffre C, Nadjar A, Layé S (2016): Neuroinflammation in Autism: Plausible Role of Maternal Inflammation, Dietary Omega 3, and Microbiota. *Neural Plast* 2016: 3597209.
18. Paolicelli RC, Bolasco G, Pagani F, Maggi L, Scianni M, Panzanelli P, *et al.* (2011): Synaptic pruning by microglia is necessary for normal brain development. *Science* 333: 1456–1458.
19. Schafer DP, Lehrman EK, Kautzman AG, Koyama R, Mardinly AR, Yamasaki R, *et al.* (2012): Microglia sculpt postnatal neural circuits in an activity and complement-dependent manner. *Neuron* 74: 691–705.
20. Stephan AH, Barres BA, Stevens B (2012): The complement system: an unexpected role in synaptic pruning during development and disease. *Annu Rev Neurosci* 35: 369–389.
21. Estes ML, McAllister AK (2015): Immune mediators in the brain and peripheral tissues in autism spectrum disorder. *Nat Rev Neurosci* 16: 469–486.
22. Salter MW, Stevens B (2017): Microglia emerge as central players in brain disease. *Nat Med* 23: 1018–1027.

23. Uderhardt S, Herrmann M, Oskolkova OV, Aschermann S, Bicker W, Ipseiz N, *et al.* (2012): 12/15-lipoxygenase orchestrates the clearance of apoptotic cells and maintains immunologic tolerance. *Immunity* 36: 834–846.
24. Gorgani NN, He JQ, Katschke KJ, Helmy KY, Xi H, Steffek M, *et al.* (2008): Complement receptor of the Ig superfamily enhances complement-mediated phagocytosis in a subpopulation of tissue resident macrophages. *J Immunol* 181: 7902–7908.
25. Harris ES, McIntyre TM, Prescott SM, Zimmerman GA (2000): The leukocyte integrins. *J Biol Chem* 275: 23409–23412.
26. Ross GD, Větvička V (1993): CR3 (CD11b, CD18): a phagocyte and NK cell membrane receptor with multiple ligand specificities and functions. *Clin Exp Immunol* 92: 181–184.
27. Luo B-H, Carman CV, Springer TA (2007): Structural basis of integrin regulation and signaling. *Annu Rev Immunol* 25: 619–647.
28. Aderem A, Underhill DM (1999): Mechanisms of phagocytosis in macrophages. *Annu Rev Immunol* 17: 593–623.
29. Griffin FM, Mullinax PJ (1981): Augmentation of macrophage complement receptor function in vitro. III. C3b receptors that promote phagocytosis migrate within the plane of the macrophage plasma membrane. *J Exp Med* 154: 291–305.
30. Presumey J, Bialas AR, Carroll MC (2017): Complement System in Neural Synapse Elimination in Development and Disease. *Adv Immunol* 135: 53–79.
31. Hong S, Beja-Glasser VF, Nfonoyim BM, Frouin A, Li S, Ramakrishnan S, *et al.* (2016): Complement and microglia mediate early synapse loss in Alzheimer mouse models. *Science* 352: 712–716.
32. Vasek MJ, Garber C, Dorsey D, Durrant DM, Bollman B, Soung A, *et al.* (2016): A complement-microglial axis drives synapse loss during virus-induced memory impairment. *Nature* 534: 538–543.
33. Boackle SA, Holers VM (2003): Role of complement in the development of autoimmunity. *Curr Dir Autoimmun* 6: 154–168.
34. De Smedt-Peyrusse V, Sargueil F, Moranis A, Harizi H, Mongrand S, Layé S (2008): Docosahexaenoic acid prevents lipopolysaccharide-induced cytokine production in microglial cells by inhibiting lipopolysaccharide receptor presentation but not its membrane subdomain localization. *J Neurochem* 105: 296–307.
35. Song M, Jin J, Lim J-E, Kou J, Pattanayak A, Rehman JA, *et al.* (2011): TLR4 mutation reduces microglial activation, increases A β deposits and exacerbates cognitive deficits in a mouse model of Alzheimer's disease. *J Neuroinflammation* 8: 92.
36. Fellner L, Irschick R, Schanda K, Reindl M, Klimaschewski L, Poewe W, *et al.* (2013): Toll-like receptor 4 is required for α -synuclein dependent activation of microglia and astroglia. *Glia* 61: 349–360.
37. Nadjar A (2018): Role of metabolic programming in the modulation of microglia phagocytosis by lipids. *Prostaglandins Leukot Essent Fatty Acids* 135: 63–73.
38. Tang R, Lin Y-M, Liu H-X, Wang E-S (2018): Neuroprotective effect of docosahexaenoic acid in rat traumatic brain injury model via regulation of TLR4/NF-Kappa B signaling pathway. *Int J Biochem Cell Biol* 99: 64–71.
39. Yip PK, Bowes AL, Hall JCE, Burguillos MA, Ip THR, Baskerville T, *et al.* (2019): Docosahexaenoic acid reduces microglia phagocytic activity via miR-124 and induces neuroprotection in rodent models of spinal cord contusion injury. *Hum Mol Genet* 28: 2427–2448.
40. Hou Q, Ruan H, Gilbert J, Wang G, Ma Q, Yao W-D, Man H-Y (2015): MicroRNA miR124 is required for the expression of homeostatic synaptic plasticity. *Nat Commun* 6: 10045.
41. Herdoiza Padilla E, Crauwels P, Bergner T, Wiederspohn N, Förstner S, Rinas R, *et al.* (2019): mir-124-5p Regulates Phagocytosis of Human Macrophages by Targeting the Actin Cytoskeleton via the ARP2/3 Complex. *Front Immunol* 10: 2210.
42. Tremblay M-E, Zhang I, Bisht K, Savage JC, Lecours C, Parent M, *et al.* (2016): Remodeling of lipid bodies by docosahexaenoic acid in activated microglial cells. *J Neuroinflammation* 13: 116.
43. Singh NK, Rao GN (2019): Emerging role of 12/15-Lipoxygenase (ALOX15) in human pathologies. *Prog Lipid Res* 73: 28–45.
44. Mukundan L, Odegaard JI, Morel CR, Heredia JE, Mwangi JW, Ricardo-Gonzalez RR, *et al.* (2009): PPAR-delta senses and orchestrates clearance of apoptotic cells to promote tolerance. *Nat Med* 15: 1266–1272.

45. Porro B, Songia P, Squellerio I, Tremoli E, Cavalca V (2014): Analysis, physiological and clinical significance of 12-HETE: A neglected platelet-derived 12-lipoxygenase product. *Journal of Chromatography B* 964: 26–40.
46. Powell WS, Rokach J (2015): Biosynthesis, biological effects, and receptors of hydroxyeicosatetraenoic acids (HETEs) and oxoeicosatetraenoic acids (oxo-ETEs) derived from arachidonic acid. *Biochim Biophys Acta* 1851: 340–355.
47. Zanoni I, Ostuni R, Marek LR, Barresi S, Barbalat R, Barton GM, *et al.* (2011): CD14 controls the LPS-induced endocytosis of Toll-like receptor 4. *Cell* 147: 868–880.
48. Namgaladze D, Snodgrass RG, Angioni C, Grossmann N, Dehne N, Geisslinger G, Brüne B (2015): AMP-activated protein kinase suppresses arachidonate 15-lipoxygenase expression in interleukin 4-polarized human macrophages. *J Biol Chem* 290: 24484–24494.
49. Guo Y, Zhang W, Giroux C, Cai Y, Ekambaram P, Dilly A-K, *et al.* (2011): Identification of the orphan G protein-coupled receptor GPR31 as a receptor for 12-(S)-hydroxyeicosatetraenoic acid. *J Biol Chem* 286: 33832–33840.
50. Haynes RL, van Leyen K (2013): 12/15-lipoxygenase expression is increased in oligodendrocytes and microglia of periventricular leukomalacia. *Dev Neurosci* 35: 140–154.
51. Krasemann S, Madore C, Cialic R, Baufeld C, Calcagno N, El Fatimy R, *et al.* (2017): The TREM2-APOE Pathway Drives the Transcriptional Phenotype of Dysfunctional Microglia in Neurodegenerative Diseases. *Immunity* 47: 566-581.e9.
52. Xu J, Zhang Y, Xiao Y, Ma S, Liu Q, Dang S, *et al.* (2013): Inhibition of 12/15-lipoxygenase by baicalein induces microglia PPAR β/δ : a potential therapeutic role for CNS autoimmune disease. *Cell Death Dis* 4: e569.
53. Sekiya K, Okuda H (1982): Selective inhibition of platelet lipoxygenase by baicalein. *Biochem Biophys Res Commun* 105: 1090–1095.
54. Deschamps JD, Kenyon VA, Holman TR (2006): Baicalein is a potent in vitro inhibitor against both reticulocyte 15-human and platelet 12-human lipoxygenases. *Bioorg Med Chem* 14: 4295–4301.
55. Bie B, Sun J, Guo Y, Li J, Jiang W, Yang J, *et al.* (2017): Baicalein: A review of its anti-cancer effects and mechanisms in Hepatocellular Carcinoma. *Biomed Pharmacother* 93: 1285–1291.
56. Zhan Y, Paolicelli RC, Sforazzini F, Weinhard L, Bolasco G, Pagani F, *et al.* (2014): Deficient neuron-microglia signaling results in impaired functional brain connectivity and social behavior. *Nat Neurosci* 17: 400–406.
57. Gu X-H, Xu L-J, Liu Z-Q, Wei B, Yang Y-J, Xu G-G, *et al.* (2016): The flavonoid baicalein rescues synaptic plasticity and memory deficits in a mouse model of Alzheimer’s disease. *Behavioural Brain Research* 311: 309–321.
58. Choudhary R, Malairaman U, Katyal A (2017): Inhibition of 12/15 LOX ameliorates cognitive and cholinergic dysfunction in mouse model of hypobaric hypoxia via. attenuation of oxidative/nitrosative stress. *Neuroscience* 359: 308–324.
59. Lafourcade M, Larrieu T, Mato S, Duffaud A, Sepers M, Matias I, *et al.* (2011): Nutritional omega-3 deficiency abolishes endocannabinoid-mediated neuronal functions. *Nat Neurosci* 14: 345–350.
60. Madore, Nadjar A, Delpech J-C, Sere A, Aubert A, Portal C, *et al.* (2014): Nutritional n-3 PUFAs deficiency during perinatal periods alters brain innate immune system and neuronal plasticity-associated genes. *Brain Behav Immun* 41: 22–31.
61. Moranis A, Delpech J-C, De Smedt-Peyrusse V, Aubert A, Guesnet P, Lavielle M, *et al.* (2012): Long term adequate n-3 polyunsaturated fatty acid diet protects from depressive-like behavior but not from working memory disruption and brain cytokine expression in aged mice. *Brain Behav Immun* 26: 721–731.
62. Delpech J-C, Thomazeau A, Madore C, Bosch-Bouju C, Larrieu T, Lacabanne C, *et al.* (2015): Dietary n-3 PUFAs Deficiency Increases Vulnerability to Inflammation-Induced Spatial Memory Impairment. *Neuropsychopharmacology* 40: 2774–2787.
63. Labrousse VF, Leyrolle Q, Amadiou C, Aubert A, Sere A, Coutureau E, *et al.* (2018): Dietary omega-3 deficiency exacerbates inflammation and reveals spatial memory deficits in mice exposed to lipopolysaccharide during gestation. *Brain Behav Immun* 73: 427–440.
64. Alashmali SM, Kitson AP, Lin L, Lacombe RJS, Bazinet RP (2019): Maternal dietary n-6 polyunsaturated fatty acid deprivation does not exacerbate post-weaning reductions in arachidonic acid and its mediators in the mouse hippocampus. *Nutr Neurosci* 22: 223–234.
65. Hopperton KE, Trépanier M-O, James NCE, Chouinard-Watkins R, Bazinet RP (2018): Fish oil feeding attenuates neuroinflammatory gene expression without concomitant changes in brain

- eicosanoids and docosanoids in a mouse model of Alzheimer's disease. *Brain Behav Immun* 69: 74–90.
66. Connor WE, Neuringer M, Lin DS (1990): Dietary effects on brain fatty acid composition: the reversibility of n-3 fatty acid deficiency and turnover of docosahexaenoic acid in the brain, erythrocytes, and plasma of rhesus monkeys. *J Lipid Res* 31: 237–247.
 67. Bazinet RP, Layé S (2014): Polyunsaturated fatty acids and their metabolites in brain function and disease. *Nat Rev Neurosci* 15: 771–785.
 68. Mingam R, De Smedt V, Amédée T, Bluthé R-M, Kelley KW, Dantzer R, Layé S (2008): In vitro and in vivo evidence for a role of the P2X7 receptor in the release of IL-1 beta in the murine brain. *Brain Behav Immun* 22: 234–244.
 69. Joffre C, Nadjar A, Lebbadi M, Calon F, Laye S (2014): n-3 LCPUFA improves cognition: the young, the old and the sick. *Prostaglandins Leukot Essent Fatty Acids* 91: 1–20.
 70. Stark KD, Lim S-Y, Salem N (2007): Artificial rearing with docosahexaenoic acid and n-6 docosapentaenoic acid alters rat tissue fatty acid composition. *J Lipid Res* 48: 2471–2477.
 71. Kim H-Y, Akbar M, Lau A (2003): Effects of docosapentaenoic acid on neuronal apoptosis. *Lipids* 38: 453–457.
 72. Rey C, Nadjar A, Joffre F, Amadieu C, Aubert A, Vaysse C, *et al.* (2018): Maternal n-3 polyunsaturated fatty acid dietary supply modulates microglia lipid content in the offspring. *Prostaglandins Leukot Essent Fatty Acids* 133: 1–7.
 73. Igarashi M, Kim H-W, Gao F, Chang L, Ma K, Rapoport SI (2012): Fifteen weeks of dietary n-3 polyunsaturated fatty acid deprivation increase turnover of n-6 docosapentaenoic acid in rat-brain phospholipids. *Biochim Biophys Acta* 1821: 1235–1243.
 74. Fourrier C, Remus-Borel J, Greenhalgh AD, Guichardant M, Bernoud-Hubac N, Lagarde M, *et al.* (2017): Docosahexaenoic acid-containing choline phospholipid modulates LPS-induced neuroinflammation in vivo and in microglia in vitro. *J Neuroinflammation* 14: 170.
 75. Chen CT, Trépanier M-O, Hopperton KE, Domenichiello AF, Masoodi M, Bazinet RP (2014): Inhibiting mitochondrial β -oxidation selectively reduces levels of nonenzymatic oxidative polyunsaturated fatty acid metabolites in the brain. *J Cereb Blood Flow Metab* 34: 376–379.
 76. Labrousse VF, Nadjar A, Joffre C, Costes L, Aubert A, Grégoire S, *et al.* (2012): Short-term long chain omega3 diet protects from neuroinflammatory processes and memory impairment in aged mice. *PLoS ONE* 7: e36861.
 77. Sierra A, Encinas JM, Deudero JJP, Chancey JH, Enikolopov G, Overstreet-Wadiche LS, *et al.* (2010): Microglia shape adult hippocampal neurogenesis through apoptosis-coupled phagocytosis. *Cell Stem Cell* 7: 483–495.
 78. Brown EJ (1989): The interaction of small oligomers of complement 3B (C3B) with phagocytes. High affinity binding and phorbol ester-induced internalization by polymorphonuclear leukocytes. *J Biol Chem* 264: 6196–6201.
 79. Brown EJ (1991): Complement receptors and phagocytosis. *Curr Opin Immunol* 3: 76–82.
 80. Stevens B, Allen NJ, Vazquez LE, Howell GR, Christopherson KS, Nouri N, *et al.* (2007): The classical complement cascade mediates CNS synapse elimination. *Cell* 131: 1164–1178.
 81. Lian H, Litvinchuk A, Chiang AC-A, Aithmitti N, Jankowsky JL, Zheng H (2016): Astrocyte-Microglia Cross Talk through Complement Activation Modulates Amyloid Pathology in Mouse Models of Alzheimer's Disease. *J Neurosci* 36: 577–589.
 82. Zwirner J, Werfel T, Wilken HC, Theile E, Götze O (1998): Anaphylatoxin C3a but not C3a(desArg) is a chemotaxin for the mouse macrophage cell line J774. *Eur J Immunol* 28: 1570–1577.
 83. Gutzmer R, Köther B, Zwirner J, Dijkstra D, Purwar R, Wittmann M, Werfel T (2006): Human plasmacytoid dendritic cells express receptors for anaphylatoxins C3a and C5a and are chemoattracted to C3a and C5a. *J Invest Dermatol* 126: 2422–2429.
 84. Hartmann K, Henz BM, Krüger-Krasagakes S, Köhl J, Burger R, Guhl S, *et al.* (1997): C3a and C5a stimulate chemotaxis of human mast cells. *Blood* 89: 2863–2870.
 85. Rabiet M-J, Huet E, Boulay F (2008): Complement component 5a receptor oligomerization and homologous receptor down-regulation. *J Biol Chem* 283: 31038–31046.
 86. Sioen I, van Lieshout L, Eilander A, Fleith M, Lohner S, Szommer A, *et al.* (2017): Systematic Review on N-3 and N-6 Polyunsaturated Fatty Acid Intake in European Countries in Light of the Current Recommendations - Focus on Specific Population Groups. *Ann Nutr Metab* 70: 39–50.
 87. Tressou J, Moulin P, Vergès B, Le Guillou C, Simon N, Pasteau S (2016): Fatty acid dietary intake in the general French population: are the French Agency for Food, Environmental and Occupational Health & Safety (ANSES) national recommendations met? *Br J Nutr* 116: 1966–1973.

88. Innis SM, Elias SL (2003): Intakes of essential n-6 and n-3 polyunsaturated fatty acids among pregnant Canadian women. *Am J Clin Nutr* 77: 473–478.
89. Tressou J, Buaud B, Simon N, Pasteau S, Guesnet P (2019): Very low inadequate dietary intakes of essential n-3 polyunsaturated fatty acids (PUFA) in pregnant and lactating French women: The INCA2 survey. *Prostaglandins Leukot Essent Fatty Acids* 140: 3–10.
90. Kuratko CN, Barrett EC, Nelson EB, Salem N (2013): The relationship of docosahexaenoic acid (DHA) with learning and behavior in healthy children: a review. *Nutrients* 5: 2777–2810.
91. Kalmijn S, van Boxtel MPJ, Ocké M, Verschuren WMM, Kromhout D, Launer LJ (2004): Dietary intake of fatty acids and fish in relation to cognitive performance at middle age. *Neurology* 62: 275–280.
92. Barberger-Gateau P, Raffaitin C, Letenneur L, Berr C, Tzourio C, Dartigues JF, Alpérovitch A (2007): Dietary patterns and risk of dementia: the Three-City cohort study. *Neurology* 69: 1921–1930.
93. Tan ZS, Harris WS, Beiser AS, Au R, Himali JJ, Debette S, *et al.* (2012): Red blood cell ω -3 fatty acid levels and markers of accelerated brain aging. *Neurology* 78: 658–664.
94. Conklin SM, Gianaros PJ, Brown SM, Yao JK, Hariri AR, Manuck SB, Muldoon MF (2007): Long-chain omega-3 fatty acid intake is associated positively with corticolimbic gray matter volume in healthy adults. *Neurosci Lett* 421: 209–212.
95. Talukdar T, Zamroziewicz MK, Zwilling CE, Barbey AK (2019): Nutrient biomarkers shape individual differences in functional brain connectivity: Evidence from omega-3 PUFAs. *Hum Brain Mapp* 40: 1887–1897.
96. McNamara RK, Vannest JJ, Valentine CJ (2015): Role of perinatal long-chain omega-3 fatty acids in cortical circuit maturation: Mechanisms and implications for psychopathology. *World J Psychiatry* 5: 15–34.
97. McNamara RK, Almeida DM (2019): Omega-3 Polyunsaturated Fatty Acid Deficiency and Progressive Neuropathology in Psychiatric Disorders: A Review of Translational Evidence and Candidate Mechanisms. *Harv Rev Psychiatry* 27: 94–107.
98. Clandinin MT, Chappell JE, Leong S, Heim T, Swyer PR, Chance GW (1980): Intrauterine fatty acid accretion rates in human brain: implications for fatty acid requirements. *Early Hum Dev* 4: 121–129.
99. Richards R, Greimel E, Kliemann D, Koerte IK, Schulte-Körne G, Reuter M, Wachinger C (2020): Increased hippocampal shape asymmetry and volumetric ventricular asymmetry in autism spectrum disorder. *Neuroimage Clin* 26: 102207.
100. Hogeveen J, Krug MK, Geddert RM, Ragland JD, Solomon M (2020): Compensatory Hippocampal Recruitment Supports Preserved Episodic Memory in Autism Spectrum Disorder. *Biol Psychiatry Cogn Neurosci Neuroimaging* 5: 97–109.
101. Farquharson J, Jamieson EC, Abbasi KA, Patrick WJ, Logan RW, Cockburn F (1995): Effect of diet on the fatty acid composition of the major phospholipids of infant cerebral cortex. *Arch Dis Child* 72: 198–203.
102. Johnson S, Hollis C, Kochhar P, Hennessy E, Wolke D, Marlow N (2010): Psychiatric disorders in extremely preterm children: longitudinal finding at age 11 years in the EPICure study. *J Am Acad Child Adolesc Psychiatry* 49: 453–463.e1.
103. Henriksen C, Haugholt K, Lindgren M, Aurvåg AK, Rønnestad A, Grønn M, *et al.* (2008): Improved cognitive development among preterm infants attributable to early supplementation of human milk with docosahexaenoic acid and arachidonic acid. *Pediatrics* 121: 1137–1145.
104. Almaas AN, Tamnes CK, Nakstad B, Henriksen C, Grydeland H, Walhovd KB, *et al.* (2016): Diffusion tensor imaging and behavior in premature infants at 8 years of age, a randomized controlled trial with long-chain polyunsaturated fatty acids. *Early Hum Dev* 95: 41–46.
105. Harris WS, Miller M, Tighe AP, Davidson MH, Schaefer EJ (2008): Omega-3 fatty acids and coronary heart disease risk: clinical and mechanistic perspectives. *Atherosclerosis* 197: 12–24.
106. Parellada M, Llorente C, Calvo R, Gutierrez S, Lázaro L, Graell M, *et al.* (2017): Randomized trial of omega-3 for autism spectrum disorders: Effect on cell membrane composition and behavior. *Eur Neuropsychopharmacol* 27: 1319–1330.
107. Hawkey E, Nigg JT (2014): Omega-3 fatty acid and ADHD: blood level analysis and meta-analytic extension of supplementation trials. *Clin Psychol Rev* 34: 496–505.
108. Hoen WP, Lijmer JG, Duran M, Wanders RJA, van Beveren NJM, de Haan L (2013): Red blood cell polyunsaturated fatty acids measured in red blood cells and schizophrenia: a meta-analysis. *Psychiatry Res* 207: 1–12.

109. van der Kemp WJM, Klomp DWJ, Kahn RS, Luijten PR, Hulshoff Pol HE (2012): A meta-analysis of the polyunsaturated fatty acid composition of erythrocyte membranes in schizophrenia. *Schizophr Res* 141: 153–161.
110. Messamore E, McNamara RK (2016): Detection and treatment of omega-3 fatty acid deficiency in psychiatric practice: Rationale and implementation. *Lipids Health Dis* 15: 25.
111. Horrobin DF (1998): The membrane phospholipid hypothesis as a biochemical basis for the neurodevelopmental concept of schizophrenia. *Schizophr Res* 30: 193–208.
112. Brown CM, Austin DW (2011): Autistic disorder and phospholipids: A review. *Prostaglandins Leukot Essent Fatty Acids* 84: 25–30.
113. Tesei A, Crippa A, Ceccarelli SB, Mauri M, Molteni M, Agostoni C, Nobile M (2017): The potential relevance of docosahexaenoic acid and eicosapentaenoic acid to the etiopathogenesis of childhood neuropsychiatric disorders. *Eur Child Adolesc Psychiatry* 26: 1011–1030.
114. Mitra S, Natarajan R, Ziedonis D, Fan X (2017): Antioxidant and anti-inflammatory nutrient status, supplementation, and mechanisms in patients with schizophrenia. *Prog Neuropsychopharmacol Biol Psychiatry* 78: 1–11.
115. Samieri C, Maillard P, Crivello F, Proust-Lima C, Peuchant E, Helmer C, *et al.* (2012): Plasma long-chain omega-3 fatty acids and atrophy of the medial temporal lobe. *Neurology* 79: 642–650.
116. Almeida DM, Jandacek RJ, Weber WA, McNamara RK (2017): Docosahexaenoic acid biostatus is associated with event-related functional connectivity in cortical attention networks of typically developing children. *Nutr Neurosci* 20: 246–254.

Reviewers' Comments:

Reviewer #1:

Remarks to the Author:

Unfortunately the authors' responses do not overcome the criticisms of this reviewer in many regards. The new gene expression profile data provided for c3ar and cd11b as well as the expanded discussion lead to more questions than answers, with inconsistent explanation and assertions. Moreover, 12/15-LOX involvement solely based on baicalein data is not sufficient for the authors' assertion that 12-/15-LOX is the master controller for the observed effects of n-3 deficiency.

Her are a few examples.

1. If DHA also contributes to microglial phagocytic activity by inhibiting TLR4/NFkB as suggested by the authors, what would be the consequence of reducing DHA in n-3 deficiency? If TLR4/NFkB plays a role, baicalein alone would not be expected to reverse the effect of n-3 deficiency. After all, n-3 deficiency decreases DHA and increase n-6 fatty acids including DPAn-6 and AA.

2. Even with new data for c3ar and cd11b expression (Fig.7), the relationship between 12-HETE, 12/15-LOX and phagocytic activity is still not clear at all.

Authors stated in rebuttal: 'We show that c3ar gene expression was increased in n-3 deficient microglia while the expression of cd11b was not significantly modulated by the diet, Baicalein significantly reduced C3aR expression in n-3 deficient microglia, while it had no effect on CD11b (Figure 7B-C). These results suggest that in non-stimulated n-3 deficient microglia, the 12/15-LOX/12-HETE signaling pathway is implicated in the maintenance of a high C3aR tone, while the activity of CD11b (as shown in Figure 4E-F of the revised version) is likely to depend on other mechanisms.'

The fact that cd11b expression in n-3 deficient microglia is comparable to that in sufficient microglia (showing even decreasing trend under stimulated condition) is not consistent with the author's claim that 'overexpression of CD11b is the driving force for increased phagocytosis in deficient microglia.' If 'CD11b is likely depend on other mechanisms' while 'overexpression of CD11b is the driving force for increased phagocytosis in deficient microglia' what is the importance of 12/15-LOX/12-HETE pathway?

Does the gene expression of c3ar and cd11b change within 2 h of exposure to synaptosomes? How does the gene expression in microglia with and without synaptosome compare? If all data from eight conditions are plotted together and normalized to n-3 sufficient group without baicalein, are there significant differences between microglia groups with or without synaptosomes? Is there any indication that c3ar and cd11b expression is affected by synaptosomal contact within 2 h?

The data and responses provided are very weak at best and confusing, and do not support the authors' suggestion that 'the 12/15-LOX/12-HETE signaling pathway is the master controller of microglial phagocytosis under n-3 PUFA deficiency, partly by regulating the expression of the C3a and C3b receptors'.

3. Authors showed that 12-HETE increases microglial phagocytic activity (Figure 6C, revised version). Regarding the possible mechanism on 12-HETE effects, authors responded as following. ...The effects of 12-HETE on microglial function are unlikely to be receptor- mediated as the application of 12-HETE on primary microglia did not induce the expression of BLT2 and GPR31 (the two principal 12-HETE receptors) (Supplementary Figure 9A). While GPR31 was expressed by the cells, BLT2 was almost not measurable in culture. Hence, we then focused on the expression of GPR31 ex vivo, on freshly isolated n-3 deficient and n-3 sufficient microglia, treated with vehicle or baicalein, in absence or presence of synaptosomes. We could not find any significant modification of GPR31 mRNA expression level in any of the conditions tested (Supplementary Figure 9 B-C)..... This explanation is quite misleading as no change in 12-HETE receptor expression in response to 12-HETE does not necessarily mean that 12-HETE receptors are not involved.

4. Regarding the use of KO animals, the authors' response indicated below is not acceptable. ... the 12/15- LOloxP/loxP mice are available at JAX. However, with the lock down situation, we cannot start a new colony and run experiments a new set of experiments. It would most likely require between 12 and 18 months to purchase the mice (no purchasing allowed in France yet), receive them from JAX, cross them with microglia-Cre mice, and run experiments on a fair number

of animals....

Both 12/15-LOX KO mice that are viable and fertile are commercially available as noted by the authors, and can be used readily. The existing data do not provide convincing evidence for the claim that the 12/15-LOX signaling is the master regulator for the aberrant phagocytic activity observed in n-3 deficient microglia. This is the main mechanistic conclusion, and it should be confirmed with data from 12/15-LOX KO mice.

5. Considering the fatty acid composition in the brain cortex (13.3% vs. 5.5% for sufficient vs. deficient), the DHA content in synaptosomes (2.5% vs. 0.5% for sufficient vs. deficient) seems strikingly low beyond the reasonable variation due to the factors mentioned by the authors.

In addition, this reviewer concurs with Review 2's comment on many data points with outliers that erroneously drive the average with limited n numbers.

Reviewer #2:

Remarks to the Author:

Dear authors,

Looking to the revised manuscript I must say the work improved! However open things are remaining such as that the data should be accessible as it is the policy of the journal in an open access database. I think the data should be uploaded to Metabolights (<https://www.ebi.ac.uk/metabolights/>). In addition, it is preferable that the authors talk only about fatty acyls since they only report them and not complex lipids. It is also not correct to use the term lipidomics within the manuscript since only fatty acyls are analyzed. However, lipidomics tries to describe all lipids globally. If this issues are dressed I am happy to support the publication @ncoomms.

Reviewer #3:

Remarks to the Author:

After careful reading of the revised manuscript and the point to point response, I believe that the authors have addressed most of the concerns I raised during the first round of review.

In particular they provide novel experiments that strengthen their conclusions and reinforce the claims put forward.

I thus now fully support publication.

We would like to thank the reviewers once more, for taking the time to review our work. It has been a constructive and insightful process, which has greatly improved our manuscript.

First, we would like to emphasize all original *in vivo* data that appear in our manuscript:

1) We showed, for the first time, that the fatty acid composition of microglial cells drives their phagocytic activity, going from cellular evidence to behavioral consequences (spatial working memory). Importantly, for the first time we show that early-life dietary PUFA composition is crucial for the polarization of post-natal microglia phenotype toward a phagocytic one. This is of high importance to the general population as the majority of pregnant women are deficient in n-3 PUFAs and this deficiency is a strong risk factor for developing neurodevelopmental defects (1). In addition, the role of nutrient bioavailability, including fatty acids, in the instruction of immunity is a rising field (2), with oxylipins being a key nutrient-derived lipid mediator regulating phagocytic activity (3). The role of lipids in microglia-driven neurodegenerative and neuropsychiatric diseases is also a rising field (4,5) with lipid-associated macrophages and microglia being keys for phagocytosis (6). In addition, the role of oxylipins in microglia-driven neuropathology has been emphasized (7) and our results will have a strong impact on our community as it brings new knowledge on dietary lipid nutrients, lipid derivative-driven microglia activity and synaptic pruning.

Considering the difficulty of running such experiments, we are also the first to have analyzed the fatty acid composition of brain membranes at a cellular (microglia) and a sub-cellular (synaptosomes) resolution.

2) As a second novel result, we showed that an increased expression of complement proteins supports an increase in microglial phagocytic activity. Importantly, we demonstrated this point *in vivo* and assessed, for the first time, the consequences on animals' behavior.

3) The third major novel finding highlighted in the paper is the role of the 12/15-LOX/12-HETE pathway in mediating deleterious effects of n-3 PUFA deficiency on microglia and memory, for which we demonstrated, *in vivo*, that n-3 PUFA deficient microglia over-express 12-HETE, and that pharmacological drug inhibiting the 12/15-LOX enzyme decreases microglial phagocytosis to control levels and restores proper memory in the juvenile.

4) Finally, the new results added in response to Reviewers' comments pinpoint that pharmacological inhibition of the 12/15-LOX enzyme controls the expression of the complement cascade, which is involved in post-natal phagocytosis of synapses by microglia (8). We also showed that inhibiting this pathway significantly decreases the phagocytic activity of n-3 PUFA deficient microglia. For all the reasons above, we concluded that 12/15-LOX signaling is contributing to the alteration of n-3 deficient microglial phagocytic activity.

Our manuscript contains a significant number of novel findings, as acknowledged by Reviewers 2&3. The understanding of how the 12/15-LOX/12-HETE pathway interacts with the complement system to promote spine phagocytosis by n-3 deficient microglia would require additional experiments that are not contributing to our main message, and the development of specific tools, currently unavailable in the lab.

To answer the Reviewer's comment(s), and the fact that more studies are needed to fully describe the mechanisms occurring in n-3 deficient microglia, 1) we added some discussion while not overstating, about the 12/15-LOX signaling pathway; 2) We emphasized on the other three mechanisms unraveled in our study (see above), which are strong, novel and impactful.

It appears as follows in the **Discussion** section:

"The phagocytic activity of microglia during development is crucial in shaping neuronal networks, and dysfunction of this activity contributes to neurodevelopmental disorders(9–13). Here we show, for the first time, that low maternal n-3 PUFA intake-induced modification of microglia fatty acid composition is a potent driver leading to exacerbation of spine phagocytosis in the offspring. This process is complement-dependent, a mechanism that has been repeatedly reported to be the molecular bridge of microglia phagocytosis of spines (8,14–18). Our data also reveal that n-3 PUFA deficiency alters the shaping of hippocampal neurons and spatial working memory in pups. Finally, we identified a previously unknown function of 12-HETE, a LOX metabolite of AA, in the regulation of microglial phagocytosis. This microglial 12/15-LOX/12-HETE pathway is also likely to contribute to alterations in spine density and connectivity observed in synaptopathies and neurodevelopmental disorders (19). Considering that n-3 PUFAs also determine offspring's microglia homeostatic molecular signature and fatty acid profile, our study suggests that maternal nutritional environment is essential for proper microglial activity in the developing brain. These findings further support the importance of dietary n-3 PUFAs in the neurodevelopmental trajectory, pointing to their role in microglia-spine interactions".

“Interestingly, C3a, that binds to microglial C3aR, is usually considered as a chemoattractant signal, attracting immune cells including microglia towards the structures to be cleared, while C3b, that binds to CR3 (CD11b+CD18,) opsonizes the target to be removed (20–24). Our data show that the 12/15-LOX/12-HETE pathway regulates the basal expression of C3aR in n-3 deficient microglia, suggesting that it may control the capacity of microglia to reach the target. However, the 12/15-LOX/12-HETE pathway does not regulate the basal gene expression of CD11b, which is in line with the observations that CR3 activation is more dependent on conformational changes and redistribution in the cellular membrane rather than on transcriptional regulation (25–30). As for CR3 activity, our data suggest that the 12/15-LOX/12-HETE pathway is involved in the down-regulation of CD11b expression once the opsonized elements have been recognized by microglia (83). Overall, we show that the 12/15-LOX/12-HETE pathway is activated upstream of the complement cascade, modulating the gene expression level and activity of the latter. We also demonstrate that this lipid pathway can modulate microglial phagocytic activity as its inhibition blunts synaptosomes engulfment. More experiments should be conducted to decipher how the 12/15-LOX/12-HETE and the complement pathways regulate phagocytosis in n-3 deficient microglia.

We also dampen our conclusion, by removing the notion of “master controller” at the end of the **Results** section (we also removed this terminology from the Discussion):

“Altogether, our data suggest that the 12/15-LOX/12-HETE signaling pathway modulates microglial phagocytosis under n-3 PUFA deficiency, partly by regulating the expression of the C3a and C3b receptors.”

We also modified the **Abstract** section to moderate our conclusions:

“Our results show that maternal dietary n-3 PUFA deficiency increases microglial phagocytosis of synaptic elements in the developing hippocampus, partly through the activation of 12/15- lipooxygenase (LOX)/12-HETE signaling, which alters neuronal morphology and affects cognition in the offspring.”

Our work is of high clinical relevance as health of pregnant women, once a neglected topic, is now a focus of increasing interest, reflected in several reports from national and international health agencies. Nutrition is now considered as a key determinant of pregnancy success and next generation health, including brain health.

Reviewer #1

Unfortunately, the authors' responses do not overcome the criticisms of this reviewer in many regards. The new gene expression profile data provided for c3ar and cd11b as well as the expanded discussion lead to more questions than answers, with inconsistent explanation and assertions. Moreover, 12/15-LOX involvement solely based on baicalein data is not sufficient for the authors' assertion that 12-/15-LOX is the master controller for the observed effects of n-3 deficiency. Here are a few examples.

1. If DHA also contributes to microglial phagocytic activity by inhibiting TLR4/NFκB as suggested by the authors, what would be the consequence of reducing DHA in n-3 deficiency? If TLR4/NFκB plays a role, baicalein alone would not be expected to reverse the effect of n-3 deficiency. After all, n-3 deficiency decreases DHA and increase n-6 fatty acids including DPAn-6 and AA.

We apologize that the discussion we added on the modulation of the phagocytic activity by DHA in the revised version was confusing. We originally added the following sentence “This finding is coherent with previous works, including ours, showing that DHA exerts inhibitory action on TLR4/NFκB pathway, TLR receptors being capable of activating microglial phagocytosis (32–35)”. Here, we were referring to previous work showing that DHA exerts anti-inflammatory activities through the inhibition of NFκB, and not that the modulation of microglial phagocytic activity by DHA is mediated by NFκB, which has not been shown to our knowledge.

We agree with the Reviewer that n-3 deficient microglia display lower levels of total DHA (Figure 3A), although the free DHA pool is not different (Figure 5A). Previous work reported that DHA inhibits microglia phagocytic activity of myelin both *in vivo* and *in vitro* (36) and normalizes LPS-induced microglial phagocytic activity of spines *in vitro* (37), which is in accordance with our work showing that free DHA decreases microglia phagocytic activity of synaptosome (Fig 6B). Additionally, we found less 17S-HDoHE, a DHA derivative precursor of resolvinD, in n-3 deficient microglia (Figure 5C). However,

no significant effect of this compound on microglia phagocytic activity was found (Figure 6D). Altogether our data suggest that neither the decrease in free DHA nor in its derivative play a role in the increase of phagocytic activity of synaptosome by n-3 deficient microglia. To avoid any confusion, we removed from the discussion the paragraph on DHA and NFκB. We rather pinpointed that if any involvement of the decrease of DHA on the n-3 deficient microglia phagocytic activity, it would be a lack of repressive effect on synaptosome phagocytosis by microglia and/or an increase of pro-inflammatory factors expression (Fig 2J), in accordance with previous work showing DHA anti-inflammatory activity.

In the Discussion section, we removed: *“This is coherent with previous works, including ours, showing that DHA exerts inhibitory action on the TLR4/NFκB pathway, TLR receptors being capable of activating microglial phagocytosis”*

We replaced it with:

“Previous work reported that DHA inhibits microglia phagocytic activity of myelin both in vivo and in vitro (36) and normalizes inflammation-induced microglial phagocytic activity of spines in vitro (37). Hence, it is likely that DHA dampens microglial phagocytosis via a repressive effect on synaptosome phagocytosis and/or a decrease in pro-inflammatory factors expression, in accordance with previous work showing its anti-inflammatory activity both in vitro and in vivo (32,38–53).”

Regarding the increase in DPA n-6 that accompanies the decrease in DHA, when applying DPA n-6 on primary microglia, we could not observe any effect on the phagocytosis of synaptosomes (Figure 6B). This suggests that DPA n-6 is unlikely to be involved in this process.

It is well established that in response to n-3 PUFA deficiency, brain DHA decreases and is replaced by DPA n-6 (69,70). We have previously shown that microglia from deficient dams have higher levels of DPA n-6 (71). While the mechanisms of this replacement at the level of the microglia are not clear, in whole brain, the uptake of DPA n-6 from the plasma free pool increases dramatically (72).

We discussed this point following Reviewer 2's recommendation. It appears as follows in the **Discussion** section:

“Our findings of a decrease in n-3 PUFAs and increase in n-6 PUFAs in n-3 deficient mice are consistent with a large body of literature (69,70). It is well established that in response to n-3 PUFA deficiency, brain DHA levels decrease and DHA is replaced by DPA n-6 (69,70). We have previously shown that microglia from n-3 PUFAs deficient dams have higher levels of DPA n-6 (71). While the mechanisms of this replacement at the level of the microglia are not clear, in the whole brain, the uptake of DPA n-6 from the plasma free pool increases dramatically (72)”.

2. Even with new data for c3ar and cd11b expression (Fig.7), the relationship between 12-HETE, 12/15-LOX and phagocytic activity is still not clear at all.

Authors stated in rebuttal: ‘We show that c3ar gene expression was increased in n-3 deficient microglia while the expression of cd11b was not significantly modulated by the diet, Baicalein significantly reduced C3aR expression in n-3 deficient microglia, while it had no effect on CD11b (Figure 7B-C). These results suggest that in non-stimulated n-3 deficient microglia, the 12/15-LOX/12-HETE signaling pathway is implicated in the maintenance of a high C3aR tone, while the activity of CD11b (as shown in Figure 4E-F of the revised version) is likely to depend on other mechanisms.’

The fact that cd11b expression in n-3 deficient microglia is comparable to that in sufficient microglia (showing even decreasing trend under stimulated condition) is not consistent with the author's claim that ‘overexpression of CD11b is the driving force for increased phagocytosis in deficient microglia.’ If ‘CD11b is likely depend on other mechanisms’ while ‘overexpression of CD11b is the driving force for increased phagocytosis in deficient microglia’ what is the importance of 12/15-LOX/12-HETE pathway?

The data and responses provided are very weak at best and confusing, and do not support the authors' suggestion that ‘the 12/15-LOX/12-HETE signaling pathway is the master controller of microglial phagocytosis under n-3 PUFA deficiency, partly by regulating the expression of the C3a and C3b receptors’.

We apologize if the way we wrote the new results and discussion was confusing and overstated and regret the use of the term “master controller” which was definitively referring to a too strong statement.

Reviewer 1 asks about the importance of 12/15LOX/12HETE pathway in n-3 deficient microglia: several experiments conducted in vivo bring arguments to the involvement of this pathway in microglia phagocytic activity

- 1) As shown in Figure 6F, increased phagocytosis of synaptosomes by n-3 deficient microglia in vivo isolated is inhibited by baicalein, the inhibitor of 12/15-LOX/12-HETE. This is associated with the restoration of spatial working memory abilities in n-3 deficient mice (Figure 6G).
- 2) When inhibiting the activity of the 12/15-LOX/12-HETE pathway in n-3 deficient microglia, by using baicalein, the mRNA expression level of *cd11b* and *c3ar* is significantly modulated. We also show that the effect of this pathway is dual, depending on the microglial status (stimulated or not with synaptosomes). In non-stimulated n-3 deficient microglia, the 12/15-LOX/12-HETE signaling pathway is implicated in the maintenance of a high C3aR tone, while once microglia phagocyte synaptosomes (2h-exposure to synaptosomes), the 12/15-LOX/12-HETE signaling pathway inhibits CD11b expression while promoting phagocytosis of neuronal material (Fig 6).
- 3) Figure 7 has been added to answer to the previous question of reviewer 1 and 3 about the link between the complement pathway and the 12/15 LOX pathway in microglia. To answer this question, we performed studies on isolated microglia exposed to baicalein 2H, with or without synaptosomes (Fig.7). The purpose of these experiments was to better understand whether the inhibition of the 12/15 LOX pathway by baicalein had an impact on the expression of complement receptor genes (*cd11b* and *c3ar*) in microglia exposed to the diet *in vivo*, while phagocytosing or not (exposed or not to synaptosomes). As shown in Figure 7, n-3 deficient microglia exposed to synaptosomes display changes in *cd11b* mRNA expression when treated with baicalein, suggesting that the 12/15-LOX pathway modulates the gene expression of CD11b when microglia phagocyte synaptosomes. Conversely, treating n-3 deficient microglia with baicalein impairs *c3ar* mRNA expression when microglia are not exposed to synaptosomes, suggesting that the 12/15-LOX pathway modulates the gene expression of C3aR when microglia are not phagocytosing.

Altogether, our results suggest that baicalein-mediated inhibition of the 12/15-LOX/12-HETE pathway affects the complement system with a different dynamic according to the phagocytic status of microglia. Therefore, we modified the following sentence (at the end of the **Results** section) “*the 12/15-LOX/12-HETE signaling pathway is the master controller of microglial phagocytosis (...) partly by regulating the expression of the C3a and C3b receptors*” as follows: “*the 12/15-LOX/12-HETE signaling pathway modulates microglial phagocytosis under n-3 PUFA deficiency, partly by regulating the expression of the CD11b and C3aR receptors*”.

From our experiments, we therefore conclude that both the 12/15-LOX/12-HETE and the complement pathways are involved in n-3 deficient microglia phagocytic activity of spines. In addition, the 12/15-LOX enzyme controls the gene expression of complement molecules, and the complement system controls microglial phagocytic activity. Hence, this suggests that n-3 PUFA deficiency recruits the 12/15-LOX/12-HETE pathway in microglia which, in turn, activates the complement-mediated phagocytic activity. This does not exclude other pathways, however our data reveal for the first time that these two pathways play a significant role in these processes.

Does the gene expression of *c3ar* and *cd11b* change within 2 h of exposure to synaptosomes? How does the gene expression in microglia with and without synaptosome compare? If all data from eight conditions are plotted together and normalized to n-3 sufficient group without baicalein, are there significant differences between microglia groups with or without synaptosomes? Is there any indication that *c3ar* and *cd11b* expression is affected by synaptosomal contact within 2 h?

Reviewer 1 asks about the timing of complement gene expression 2h after exposure to synaptosome. We apologize if the experimental design was not clear. All experiments using freshly isolated microglia are conducted after 2h of exposure to synaptosomes (Fig 6F; Fig 7) and this has been added in the Figure legends. A 2h-exposure of isolated microglia to synaptosomes *ex vivo* triggers phagocytosis as previously reported (54). Some preliminary work performed *in vitro*, in the context of another study, shows that there is no effect of synaptosome on the complement gene expression (see Figure below).

Rebuttal Figure 1: **Gene expression of the complement receptors in primary microglia exposed to synaptosomes for 2h (SYN+) or not exposed (SYN-).** A 2h exposure to synaptosomes does not change the gene expression level of C3aR (t-test, $p=0.93$) and CD11b (t-test, $p=0.41$).

Concerning the question of reviewer 1 asking for plotting all the data from Fig 7 together, we cannot do such a comparison as all the experiments were not conducted at the same time. These experiments are labor intensive, and 8 experimental groups cannot be run at the same time as it requires to run cell sorting, expose cell sorted microglia to drugs and synaptosomes and run the FACS analysis the same day. In addition, it is known that in the healthy developing brain, C1q promotes activation of C3, which opsonizes subsets of synapses for elimination (55). Deciphering the interplay between exposure to synaptosomes, regulation of complement system and phagocytosis is a question of high interest but requires many experimentations which are out of the scope of our study.

Indeed, our purpose in doing these experiments was not to decipher how exposure to synaptosomes affects complement gene expression in microglia. Our objective was to test whether the 12/15-LOX/12-HETE pathway was influencing the complement system gene expression in n-3 deficient microglia. To this aim, we conducted two independent series of experiments: 1) Treating n-3 sufficient and n-3 deficient microglia with baicalein *ex vivo*; 2) Treating n-3 sufficient and n-3 deficient microglia exposed for 2 hours to synaptosomes with baicalein. Our data first confirm that *in vivo* microglia express complement receptors in the developing brain and mediate synaptic pruning (Fig 4, 6 and 7) (8). Also, these results bring arguments on the link between the 12/15-LOX/12-HETE pathway and complement system in microglia, which was an initial question of Reviewers 1 and 3. To our knowledge, this is the first demonstration that baicalein impairs the expression of complement system gene expression in sorted microglia exposed or not to synaptosomes.

3. Authors showed that 12-HETE increases microglial phagocytic activity (Figure 6C, revised version). Regarding the possible mechanism on 12-HETE effects, authors responded as following.

...The effects of 12-HETE on microglial function are unlikely to be receptor-mediated as the application of 12-HETE on primary microglia did not induce the expression of BLT2 and GPR31 (the two principal 12-HETE receptors) (Supplementary Figure 9A). While GPR31 was expressed by the cells, BLT2 was almost not measurable in culture. Hence, we then focused on the expression of GPR31 *ex vivo*, on freshly isolated n-3 deficient and n-3 sufficient microglia, treated with vehicle or baicalein, in absence or presence of synaptosomes. We could not find any significant modification of GPR31 mRNA expression level in any of the conditions tested (Supplementary Figure 9 B-C).....

This explanation is quite misleading as no change in 12-HETE receptor expression in response to 12-HETE does not necessarily mean.

We fully agree with Reviewer 1 that no change in mRNA expression does not mean no role at all of the 12-HETE receptors. Hence, we modified the structure of the paragraph as follows in the **Results** section:

"We finally studied the expression levels of BLT2 and GPR31, the two principal 12-HETE receptors (Supplementary Figure 9A). While GPR31 was expressed by the cells, BLT2 gene expression was almost not measurable in microglia in vitro. Hence, we then focused on the expression of GPR31 in vivo, on freshly isolated n-3 deficient and n-3 sufficient microglia, treated with vehicle or baicalein, in absence or presence of synaptosomes. We could not find any significant modification of GPR31 mRNA expression level in any of the conditions tested (Supplementary Figure 9 B-C). Further experiments are needed to decipher whether 12 HETE effect on phagocytosis is mediated by GPR31."

4. Regarding the use of KO animals, the authors' response indicated below is not acceptable.

... the 12/15- LOloxP/loxP mice are available at JAX. However, with the lock down situation, we cannot start a new colony and run experiments a new set of experiments. It would most likely require

between 12 and 18 months to purchase the mice (no purchasing allowed in France yet), receive them from JAX, cross them with microglia-Cre mice, and run experiments on a fair number of animals....

Both 12/15-LOX KO mice that are viable and fertile are commercially available as noted by the authors, and can be used readily. The existing data do not provide convincing evidence for the claim that the 12/15-LOX signaling is the master regulator for the aberrant phagocytic activity observed in n-3 deficient microglia. This is the main mechanistic conclusion, and it should be confirmed with data from 12/15-LOX KO mice.

The reviewer's request is to strengthen the mechanistic claims on the ability of 12/15-LOX/12-HETE signaling to regulate microglia phagocytosis by using 12/15-LOX KO mice.

Using a full KO is unlikely to bring clear response to this question as the 12/15-LOX is constitutively expressed throughout the body, including in reticulocytes, eosinophils, dendritic cells, alveolar macrophages, airway epithelial cells, immature dendritic cells, vascular cells, resident peritoneal macrophages, pancreatic islets, uterus and brain (56–61).

Moreover, a paper recently published in JBC reported that the 12/15-LOX and the 12-LOX enzymes can compensate for each other, so that 12/15-LOX KO mice overexpress the 12-LOX enzyme, and conversely, the 12-LOX KO mice overexpress the 12/15-LOX enzyme (62). As a consequence of this redundant system, 12/15-LOX KO mice still produce 12-HETE, knowing that both enzymes can produce this lipid derivative (61,62). We apologize for not emphasizing this limitation in the previous rebuttal letter.

In addition, due to the previous lockdown situation and the current restart of the COVID-19 in several countries, including France, we are not allowed by our university to receive new animal strains, as the risk of novel lockdown increases. Hence considering all the points above related to the limitation of the transgenic strains' specificity, added to the current pandemic situation with a blocked situation at the university animal facility which cannot receive new mouse strains, the increment of the transgenic approach is unfortunately not manageable for us.

Here is what we added in the **Discussion** section:

“Our data show that the 12/15-LOX/12-HETE pathway controls the basal expression of C3aR in n-3 deficient microglia, suggesting that it may control the capacity of microglia to reach the target. However, the 12/15-LOX/12-HETE pathway does not regulate the gene expression of CD11b, which is in line with the observations that CD11b activation is more dependent on conformational changes and redistribution in the cellular membrane rather than on transcriptional regulation (25–30). As for CD11b activity, our data suggest that the 12/15-LOX/12-HETE pathway is involved in the homeostatic down-regulation of CD11b expression once the opsonized elements have been recognized by microglia (83). Overall, our data suggest that the 12/15-LOX/12-HETE pathway is activated upstream of the complement cascade, controlling the gene expression level and activity of this latter. We also found that 12/15-LOX/12-HETE pathway modulates microglial phagocytic activity as baicalein reduces synaptosomes engulfment. More experiments should be conducted to decipher the interplay of 12/15-LOX/12-HETE and the complement pathways in the phagocytic activity of n-3 deficient microglia.”

5. Considering the fatty acid composition in the brain cortex (13.3% vs. 5.5% for sufficient vs. deficient), the DHA content in synaptosomes (2.5% vs. 0.5% for sufficient vs. deficient) seems strikingly low beyond the reasonable variation due to the factors mentioned by the authors.

We agree with the Reviewer that the DHA content observed in the synaptosomes is quite low when compared to the previous literature and would like to thank him/her for pointing that out. We measured fatty acids on a new batch of synaptosomes. New results appear in **Figure 3B** (see below). DHA levels were of 16.78 +/-0.27 in n-3 sufficient synaptosomes and of 2.75+/- 0.06 in n-3 deficient synaptosomes.

Results and Legend sections now appear as follows:

Results: “In synaptosomes, total AA level was significantly increased while total DHA level was significantly decreased and DPA n-6 inversely increased, as observed in microglia (Figure 3B). In synaptosomes, more fatty acids were significantly modified by the diet, such as the saturated fatty acid (SFA) stearic acid (18:0) and docosanoic acid (C22:0), and the monounsaturated fatty acids (MUFA) C16:1 n-7, C16:1 n-9, C18:1 n-9, and C20:1 n-9.”

Legend: B. “Fatty acid composition of synaptosomes sorted from n-3 sufficient and n-3 deficient mice. Means \pm SEM; n=4 mice per group. Insert: higher magnification of low expressed fatty acids. Two-tailed unpaired Student’s t-test; t=3.13, *p=0.02, C18:0; t=3.64, *p=0.011, C22:0; t=3.3, *p=0.016 C16:1 n-7; t=3.93, **p=0.0077, C16:1 n-9; t=4.96, **p=0.0026, C18:1 n-9; t=5.15, **p=0.0021, C20:1 n-9; t=3.8, **p=0.0089, LNA; t=12.44, ***p<0.0001, C20:2 n-6; t=3.4, *p=0.014, AA; t=9.2, ***p<0.0001, C22:4 n-6; t=251.6, ***p<0.0001, C22:5 n-6; t=10.12, ***p<0.0001, C20:5 n-3; t=4.73, **p=0.0032, C22:5 n-3; t=101.2, ***p<0.0001, DHA; ‘nd’: not detected.

6. In addition, this reviewer concurs with Review 2’s comment on many data points with outliers that erroneously drive the average with limited n numbers.

As pointed out by Reviewer 2 during the first round of revisions, some information was missing on our statistical approach, which we added in the new version of the manuscript.

Here is the information provided:

We have conducted experiments along lines that are classically used in the literature and are based on previous/pilot experiments ran in the lab that have provided results with statistical significance. On average, we used (after removal of the outliers, see below for the calculation of outliers): N=5 for Golgi staining experiments; N=5-8 for Western blotting; N=5-10 for immunostaining; N=10-17 for Y-maze; N=4 for lipidomics; N=7-8 for ELISA; N=3-20 for ex-vivo phagocytosis; N=4-6 for transcriptomics; N=5-14 (each N containing duplicates or triplicates for each measurement) for in vitro studies; N=3 for electron microscopy (>600 measurements over the 3 animals). We respected the 3R rules, by not unnecessarily increasing the numbers of animal used in our experiments.

Calculation of the outliers: We used the Grubbs’ test, also called the ESD method (extreme studentized deviate), to determine whether the most extreme value in the list is a significant outlier from the rest. Unlike some other outlier tests, Grubbs’ test only asks whether that one value is an outlier. It is not appropriate to then remove that outlier, and run the test again.

The first step is to calculate the ratio Z as the difference between the outlier and the mean divided by the SD. If Z is large, the value is far from the others. The mean and SD must be calculated from all values, including the outlier.

$$Z = \frac{|\text{mean} - \text{value}|}{\text{SD}}$$

The presence of an outlier increases the calculated SD. Since the presence of an outlier increases both the numerator (difference between the value and the mean) and denominator (SD of all values), Z cannot get as large as many expect. For example, if N=3, Z cannot be larger than 1.155 for any set of values. More generally, with a sample of N observations, Z can never get larger than $(N-1)/\sqrt{N}$.

We added this information to the **Methods** section for clarity:

Statistical analyses

For most experiments, experimental groups were compared using Student t-test or non-parametric Mann-Whitney test (when equality of variance or normality failed). We used the Grubbs' test, also called the ESD method (extreme studentized deviate), to determine whether the most extreme value in the list is a significant outlier from the rest. Paired t-tests were used for Y-maze experiments (familiar vs novel arm for each experimental group). A Two-Way ANOVA was used for XVA-143, cRGD, and baicalein experiments (except Y-maze), ex vivo and in vitro phagocytic activity. All data were expressed as means \pm standard error of the mean (SEM). A $p < 0.05$ was considered as statistically significant.

Reviewer #2

Dear authors,

Looking to the revised manuscript I must say the work improved! However open things are remaining such as that the data should be accessible as it is the policy of the journal in an open access database. I think the data should be uploaded to Metabolights (www.ebi.ac.uk/metabolights). In addition, it is preferable that the authors talk only about fatty acyls since they only report them and not complex lipids. It is also not correct to use the term lipidomics within the manuscript since only fatty acyls are analyzed. However, lipidomics tries to describe all lipids globally. If these issues are addressed I am happy to support the publication @ncoomms.

We would like to thank the Reviewer for the positive feedback and for supporting the publication of our work in Nature Communications.

We fully agree with Reviewer 2 that talking about lipid profiling is an overstatement. Hence, we replaced the term "Lipid" by the term "Fatty acid" across the whole manuscript. Thank you for pointing that out. According to the reviewer's previous recommendation, we already replaced "Lipidomics" by terms such as "Fatty acids analysis" or "Fatty acids quantification" in the Methods section. We fully agree that the terminology "Lipidomics" was overstated.

We also uploaded our fatty acid data on Metabolights, as suggested by the Reviewer. Our study is identified as "MTBLS1930".

Reviewer #3

After careful reading of the revised manuscript and the point to point response, I believe that the authors have addressed most of the concerns I raised during the first round of review. In particular, they provide novel experiments that strengthen their conclusions and reinforce the claims put forward. I thus now fully support publication.

We would like to thank the Reviewer for highlighting the novel experiments provided and the fact that they strengthen our conclusions.

References

1. Li M, Francis E, Hinkle SN, Ajarapu AS, Zhang C (2019): Preconception and Prenatal Nutrition and Neurodevelopmental Disorders: A Systematic Review and Meta-Analysis. *Nutrients* 11. <https://doi.org/10.3390/nu11071628>
2. Buck MD, Sowell RT, Kaech SM, Pearce EL (2017): Metabolic Instruction of Immunity. *Cell* 169: 570–586.
3. Serhan CN (2014): Pro-resolving lipid mediators are leads for resolution physiology. *Nature* 510: 92–101.
4. Wang Y, Cella M, Mallinson K, Ulrich JD, Young KL, Robinette ML, *et al.* (2015): TREM2 lipid sensing sustains microglia response in an Alzheimer's disease model. *Cell* 160: 1061–1071.
5. Nugent AA, Lin K, van Lengerich B, Lianoglou S, Przybyla L, Davis SS, *et al.* (2020): TREM2 Regulates Microglial Cholesterol Metabolism upon Chronic Phagocytic Challenge. *Neuron* 105: 837–854.e9.
6. Marschallinger J, Iram T, Zardeneta M, Lee SE, Lehallier B, Haney MS, *et al.* (2020): Lipid-droplet-accumulating microglia represent a dysfunctional and proinflammatory state in the aging brain. *Nat Neurosci* 23: 194–208.
7. Chen G, Zhang Y-Q, Qadri YJ, Serhan CN, Ji R-R (2018): Microglia in Pain: Detrimental and Protective Roles in Pathogenesis and Resolution of Pain. *Neuron* 100: 1292–1311.
8. Schafer DP, Lehrman EK, Kautzman AG, Koyama R, Mardinly AR, Yamasaki R, *et al.* (2012): Microglia sculpt postnatal neural circuits in an activity and complement-dependent manner. *Neuron* 74: 691–705.
9. Tay TL, Savage JC, Hui CW, Bisht K, Tremblay M-È (2017): Microglia across the lifespan: from origin to function in brain development, plasticity and cognition. *J Physiol (Lond)* 595: 1929–1945.
10. Zhan Y, Paolicelli RC, Sforzini F, Weinhard L, Bolasco G, Pagani F, *et al.* (2014): Deficient neuron-microglia signaling results in impaired functional brain connectivity and social behavior. *Nat Neurosci* 17: 400–406.
11. Hammond TR, Robinton D, Stevens B (2018): Microglia and the Brain: Complementary Partners in Development and Disease. *Annu Rev Cell Dev Biol* 34: 523–544.
12. Bar E, Barak B (2019): Microglia roles in synaptic plasticity and myelination in homeostatic conditions and neurodevelopmental disorders. *Glia*. <https://doi.org/10.1002/glia.23637>
13. Sellgren CM, Gracias J, Watmuff B, Biag JD, Thanos JM, Whittredge PB, *et al.* (2019): Increased synapse elimination by microglia in schizophrenia patient-derived models of synaptic pruning. *Nat Neurosci* 22: 374–385.
14. Bialas AR, Stevens B (2013): TGF- β signaling regulates neuronal C1q expression and developmental synaptic refinement. *Nat Neurosci* 16: 1773–1782.
15. Hammond TR, Marsh SE, Stevens B (2019): Immune Signaling in Neurodegeneration. *Immunity* 50: 955–974.
16. Vasek MJ, Garber C, Dorsey D, Durrant DM, Bollman B, Soung A, *et al.* (2016): A complement-microglial axis drives synapse loss during virus-induced memory impairment. *Nature* 534: 538–543.
17. Hong S, Beja-Glasser VF, Nfonoyim BM, Frouin A, Li S, Ramakrishnan S, *et al.* (2016): Complement and microglia mediate early synapse loss in Alzheimer mouse models. *Science* 352: 712–716.
18. Wyatt SK, Witt T, Barbaro NM, Cohen-Gadol AA, Brewster AL (2017): Enhanced classical complement pathway activation and altered phagocytosis signaling molecules in human epilepsy. *Exp Neurol* 295: 184–193.
19. Lepeta K, Lourenco MV, Schweitzer BC, Martino Adami PV, Banerjee P, Catuara-Solarz S, *et al.* (2016): Synaptopathies: synaptic dysfunction in neurological disorders - A review from students to students. *J Neurochem* 138: 785–805.
20. Boackle SA, Holers VM (2003): Role of complement in the development of autoimmunity. *Curr Dir Autoimmun* 6: 154–168.
21. Lian H, Litvinchuk A, Chiang AC-A, Aithmitti N, Jankowsky JL, Zheng H (2016): Astrocyte-Microglia Cross Talk through Complement Activation Modulates Amyloid Pathology in Mouse Models of Alzheimer's Disease. *J Neurosci* 36: 577–589.
22. Zwirner J, Werfel T, Wilken HC, Theile E, Götze O (1998): Anaphylatoxin C3a but not C3a(desArg) is a chemotaxin for the mouse macrophage cell line J774. *Eur J Immunol* 28: 1570–1577.
23. Gutzmer R, Köther B, Zwirner J, Dijkstra D, Purwar R, Wittmann M, Werfel T (2006): Human plasmacytoid dendritic cells express receptors for anaphylatoxins C3a and C5a and are chemoattracted to C3a and C5a. *J Invest Dermatol* 126: 2422–2429.
24. Hartmann K, Henz BM, Krüger-Krasagakes S, Köhl J, Burger R, Guhl S, *et al.* (1997): C3a and C5a stimulate chemotaxis of human mast cells. *Blood* 89: 2863–2870.

25. Gorgani NN, He JQ, Katschke KJ, Helmy KY, Xi H, Steffek M, *et al.* (2008): Complement receptor of the Ig superfamily enhances complement-mediated phagocytosis in a subpopulation of tissue resident macrophages. *J Immunol* 181: 7902–7908.
26. Harris ES, McIntyre TM, Prescott SM, Zimmerman GA (2000): The leukocyte integrins. *J Biol Chem* 275: 23409–23412.
27. Ross GD, Větvicka V (1993): CR3 (CD11b, CD18): a phagocyte and NK cell membrane receptor with multiple ligand specificities and functions. *Clin Exp Immunol* 92: 181–184.
28. Luo B-H, Carman CV, Springer TA (2007): Structural basis of integrin regulation and signaling. *Annu Rev Immunol* 25: 619–647.
29. Aderem A, Underhill DM (1999): Mechanisms of phagocytosis in macrophages. *Annu Rev Immunol* 17: 593–623.
30. Griffin FM, Mullinax PJ (1981): Augmentation of macrophage complement receptor function in vitro. III. C3b receptors that promote phagocytosis migrate within the plane of the macrophage plasma membrane. *J Exp Med* 154: 291–305.
31. Rabiet M-J, Huet E, Boulay F (2008): Complement component 5a receptor oligomerization and homologous receptor down-regulation. *J Biol Chem* 283: 31038–31046.
32. De Smedt-Peyrusse V, Sargueil F, Moranis A, Harizi H, Mongrand S, Layé S (2008): Docosahexaenoic acid prevents lipopolysaccharide-induced cytokine production in microglial cells by inhibiting lipopolysaccharide receptor presentation but not its membrane subdomain localization. *J Neurochem* 105: 296–307.
33. Fellner L, Irschick R, Schanda K, Reindl M, Klimaschewski L, Poewe W, *et al.* (2013): Toll-like receptor 4 is required for α -synuclein dependent activation of microglia and astroglia. *Glia* 61: 349–360.
34. Nadjar A (2018): Role of metabolic programming in the modulation of microglia phagocytosis by lipids. *Prostaglandins Leukot Essent Fatty Acids* 135: 63–73.
35. Song M, Jin J, Lim J-E, Kou J, Pattanayak A, Rehman JA, *et al.* (2011): TLR4 mutation reduces microglial activation, increases A β deposits and exacerbates cognitive deficits in a mouse model of Alzheimer's disease. *J Neuroinflammation* 8: 92.
36. Yip PK, Bowes AL, Hall JCE, Burguillos MA, Ip THR, Baskerville T, *et al.* (2019): Docosahexaenoic acid reduces microglia phagocytic activity via miR-124 and induces neuroprotection in rodent models of spinal cord contusion injury. *Hum Mol Genet* 28: 2427–2448.
37. Chang PK-Y, Khatchadourian A, McKinney RA, Maysinger D (2015): Docosahexaenoic acid (DHA): a modulator of microglia activity and dendritic spine morphology. *J Neuroinflammation* 12: 34.
38. Ebert S, Weigelt K, Walczak Y, Drobnik W, Mauerer R, Hume DA, *et al.* (2009): Docosahexaenoic acid attenuates microglial activation and delays early retinal degeneration. *J Neurochem* 110: 1863–1875.
39. Zhang W, Hu X, Yang W, Gao Y, Chen J (2010): Omega-3 polyunsaturated fatty acid supplementation confers long-term neuroprotection against neonatal hypoxic-ischemic brain injury through anti-inflammatory actions. *Stroke* 41: 2341–2347.
40. Antonietta Ajmone-Cat M, Lavinia Salvatori M, De Simone R, Mancini M, Biagioni S, Bernardo A, *et al.* (2012): Docosahexaenoic acid modulates inflammatory and antineurogenic functions of activated microglial cells. *J Neurosci Res* 90: 575–587.
41. Pettit LK, Varsanyi C, Tados J, Vassiliou E (2013): Modulating the inflammatory properties of activated microglia with Docosahexaenoic acid and Aspirin. *Lipids Health Dis* 12: 16.
42. Hjorth E, Zhu M, Toro VC, Vedin I, Palmblad J, Cederholm T, *et al.* (2013): Omega-3 fatty acids enhance phagocytosis of Alzheimer's disease-related amyloid- β 42 by human microglia and decrease inflammatory markers. *J Alzheimers Dis* 35: 697–713.
43. Chen S, Zhang H, Pu H, Wang G, Li W, Leak RK, *et al.* (2014): n-3 PUFA supplementation benefits microglial responses to myelin pathology. *Sci Rep* 4: 7458.
44. Zendedel A, Habib P, Dang J, Lammerding L, Hoffmann S, Beyer C, Slowik A (2015): Omega-3 polyunsaturated fatty acids ameliorate neuroinflammation and mitigate ischemic stroke damage through interactions with astrocytes and microglia. *J Neuroimmunol* 278: 200–211.
45. Wang X, Hjorth E, Vedin I, Eriksdotter M, Freund-Levi Y, Wahlund L-O, *et al.* (2015): Effects of n-3 FA supplementation on the release of proresolving lipid mediators by blood mononuclear cells: the OmegAD study. *J Lipid Res* 56: 674–681.
46. Kurtys E, Eisel ULM, Verkuyl JM, Broersen LM, Dierckx R a. JO, de Vries EFJ (2016): The combination of vitamins and omega-3 fatty acids has an enhanced anti-inflammatory effect on microglia. *Neurochem Int* 99: 206–214.
47. Inoue T, Tanaka M, Masuda S, Ohue-Kitano R, Yamakage H, Muranaka K, *et al.* (2017): Omega-3 polyunsaturated fatty acids suppress the inflammatory responses of lipopolysaccharide-stimulated

- mouse microglia by activating SIRT1 pathways. *Biochim Biophys Acta Mol Cell Biol Lipids* 1862: 552–560.
48. Madore, Nadjar A, Delpech J-C, Sere A, Aubert A, Portal C, *et al.* (2014): Nutritional n-3 PUFAs deficiency during perinatal periods alters brain innate immune system and neuronal plasticity-associated genes. *Brain Behav Immun* 41: 22–31.
49. Abiega O, Beccari S, Diaz-Aparicio I, Nadjar A, Layé S, Leyrolle Q, *et al.* (2016): Neuronal Hyperactivity Disturbs ATP Microgradients, Impairs Microglial Motility, and Reduces Phagocytic Receptor Expression Triggering Apoptosis/Microglial Phagocytosis Uncoupling. *PLoS Biol* 14: e1002466.
50. Tian Y, Zhang Y, Zhang R, Qiao S, Fan J (2015): Resolvin D2 recovers neural injury by suppressing inflammatory mediators expression in lipopolysaccharide-induced Parkinson's disease rat model. *Biochem Biophys Res Commun* 460: 799–805.
51. Lynch AM, Loane DJ, Minogue AM, Clarke RM, Kilroy D, Nally RE, *et al.* (2007): Eicosapentaenoic acid confers neuroprotection in the amyloid-beta challenged aged hippocampus. *Neurobiol Aging* 28: 845–855.
52. Hopperton KE, Trépanier M-O, Giuliano V, Bazinet RP (2016): Brain omega-3 polyunsaturated fatty acids modulate microglia cell number and morphology in response to intracerebroventricular amyloid- β 1-40 in mice. *J Neuroinflammation* 13: 257.
53. Delpech J-C, Thomazeau A, Madore C, Bosch-Bouju C, Larrieu T, Lacabanne C, *et al.* (2015): Dietary n-3 PUFAs Deficiency Increases Vulnerability to Inflammation-Induced Spatial Memory Impairment. *Neuropsychopharmacology* 40: 2774–2787.
54. Filipello F, Morini R, Corradini I, Zerbi V, Canzi A, Michalski B, *et al.* (2018): The Microglial Innate Immune Receptor TREM2 Is Required for Synapse Elimination and Normal Brain Connectivity. *Immunity* 48: 979-991.e8.
55. Stevens B, Allen NJ, Vazquez LE, Howell GR, Christopherson KS, Nouri N, *et al.* (2007): The classical complement cascade mediates CNS synapse elimination. *Cell* 131: 1164–1178.
56. Kühn H, O'Donnell VB (2006): Inflammation and immune regulation by 12/15-lipoxygenases. *Prog Lipid Res* 45: 334–356.
57. Dobrian AD, Lieb DC, Cole BK, Taylor-Fishwick DA, Chakrabarti SK, Nadler JL (2011): Functional and pathological roles of the 12- and 15-lipoxygenases. *Prog Lipid Res* 50: 115–131.
58. Kroschwald P, Kroschwald A, Kühn H, Ludwig P, Thiele BJ, Höhne M, *et al.* (1989): Occurrence of the erythroid cell specific arachidonate 15-lipoxygenase in human reticulocytes. *Biochem Biophys Res Commun* 160: 954–960.
59. Rothe T, Gruber F, Uderhardt S, Ipseiz N, Rössner S, Oskolkova O, *et al.* (2015): 12/15-Lipoxygenase-mediated enzymatic lipid oxidation regulates DC maturation and function. *J Clin Invest* 125: 1944–1954.
60. Kim JA, Gu JL, Natarajan R, Berliner JA, Nadler JL (1995): A leukocyte type of 12-lipoxygenase is expressed in human vascular and mononuclear cells. Evidence for upregulation by angiotensin II. *Arterioscler Thromb Vasc Biol* 15: 942–948.
61. Singh NK, Rao GN (2019): Emerging role of 12/15-Lipoxygenase (ALOX15) in human pathologies. *Prog Lipid Res* 73: 28–45.
62. Conteh AM, Reissaus CA, Hernandez-Perez M, Nakshatri S, Anderson RM, Mirmira RG, *et al.* (2019): Platelet-type 12-lipoxygenase deletion provokes a compensatory 12/15-lipoxygenase increase that exacerbates oxidative stress in mouse islet β cells. *J Biol Chem* 294: 6612–6620.
63. Cotman C, Blank ML, Moehl A, Snyder F (1969): Lipid composition of synaptic plasma membranes isolated from rat brain by zonal centrifugation. *Biochemistry* 8: 4606–4612.
64. Breckenridge WC, Gombos G, Morgan IG (1972): The lipid composition of adult rat brain synaptosomal plasma membranes. *Biochim Biophys Acta* 266: 695–707.
65. Sun GY, Go J, Sun AY (1974): Induction of essential fatty acid deficiency in mouse brain: effects of fat deficient diet upon acyl group composition of myelin and synaptosome-rich fractions during development and maturation. *Lipids* 9: 450–454.
66. Cao D, Kevala K, Kim J, Moon H-S, Jun SB, Lovinger D, Kim H-Y (2009): Docosahexaenoic acid promotes hippocampal neuronal development and synaptic function. *J Neurochem* 111: 510–521.
67. Sidhu VK, Huang BX, Kim H-Y (2011): Effects of docosahexaenoic acid on mouse brain synaptic plasma membrane proteome analyzed by mass spectrometry and (16)O/(18)O labeling. *J Proteome Res* 10: 5472–5480.
68. Bourre J-M, Pascal G, Durand G, Masson M, Dumont O, Piciotti M (1984): Alterations in the Fatty Acid Composition of Rat Brain Cells (Neurons, Astrocytes, and Oligodendrocytes) and of Subcellular Fractions (Myelin and Synaptosomes) Induced by a Diet Devoid of n-3 Fatty Acids. *Journal of Neurochemistry* 43: 342–348.

69. Stark KD, Lim S-Y, Salem N (2007): Artificial rearing with docosahexaenoic acid and n-6 docosapentaenoic acid alters rat tissue fatty acid composition. *J Lipid Res* 48: 2471–2477.
70. Kim H-Y, Akbar M, Lau A (2003): Effects of docosapentaenoic acid on neuronal apoptosis. *Lipids* 38: 453–457.
71. Rey C, Nadjar A, Joffre F, Amadieu C, Aubert A, Vaysse C, *et al.* (2018): Maternal n-3 polyunsaturated fatty acid dietary supply modulates microglia lipid content in the offspring. *Prostaglandins Leukot Essent Fatty Acids* 133: 1–7.
72. Igarashi M, Kim H-W, Gao F, Chang L, Ma K, Rapoport SI (2012): Fifteen weeks of dietary n-3 polyunsaturated fatty acid deprivation increase turnover of n-6 docosapentaenoic acid in rat-brain phospholipids. *Biochim Biophys Acta* 1821: 1235–1243.

Reviewers' Comments:

Reviewer #1:

Remarks to the Author:

The authors provided additional data for correct information and made efforts to clarify confusing inconsistencies. This reviewer now agrees their novel findings to be published in Nat Commun.

REVIEWERS' COMMENTS

Reviewer #1 (Remarks to the Author):

The authors provided additional data for correct information and made efforts to clarify confusing inconsistencies. This reviewer now agrees their novel findings to be published in Nat Commun.

We thank the reviewer for his/her insightful comments and for supporting the publication of our work in Nature Communications.